# From Gradient Flow on Population Loss to Learning with Stochastic Gradient Descent

**Ayush Sekhari**[‡]
as3663@cornell.edu

**Satyen Kale**[*]
satyenkale@google.com

**Jason D. Lee**[†]
jasonlee@princeton.edu

**Chris De Sa**[‡]
cdesa@cs.cornell.edu

**Karthik Sridharan**[‡]
ks999@cornell.edu

## Abstract

Stochastic Gradient Descent (SGD) has been the method of choice for learning large-scale non-convex models. While a general analysis of when SGD works has been elusive, there has been a lot of recent progress in understanding the convergence of Gradient Flow (GF) on the population loss, partly due to the simplicity that a continuous-time analysis buys us. An overarching theme of our paper is providing general conditions under which SGD converges, assuming that GF on the population loss converges. Our main tool to establish this connection is a general *converse Lyapunov* like theorem, which implies the existence of a Lyapunov potential under mild assumptions on the rates of convergence of GF. In fact, using these potentials, we show a one-to-one correspondence between rates of convergence of GF and geometrical properties of the underlying objective. When these potentials further satisfy certain self-bounding properties, we show that they can be used to provide a convergence guarantee for Gradient Descent (GD) and SGD (even when the paths of GF and GD/SGD are quite far apart). It turns out that these self-bounding assumptions are in a sense also necessary for GD/SGD to work. Using our framework, we provide a unified analysis for GD/SGD not only for classical settings like convex losses, or objectives that satisfy PŁ / KŁ properties, but also for more complex problems including Phase Retrieval and Matrix sq-root, and extending the results in the recent work of Chatterjee [13].

## 1 Introduction

Stochastic Gradient Descent (SGD) has been a method of choice to train complex, large scale machine learning models. While understanding of SGD for convex objectives is comprehensive, a general understanding of when SGD works for non-convex models has been somewhat elusive. A large slew of properties like, convexity [47], one-point-convexity [35], linearizability [32], KŁ [5, 40] and PŁ [33, 49, 41] properties, and more problem specific, tailored analysis of SGD and Gradient Descent (GD) for specific problem instances like matrix square-root problem, matrix completion [31], phase retrieval [11, 16, 53] and Dictionary learning [4] have been proposed. Recent success of SGD in over-parameterized deep learning models have lead to the idea that SGD perhaps optimizes training objective with an implicit bias given by some implicit regularizer [24, 50, 29, 25, 26]. However, in [32] it is argued that there are over-parameterized models for which SGD works but no method that minimizes an implicit regularized training objective can learn successfully, thus showing that in general, the success of SGD cannot be explained by implicit regularization.

---

[*]Google Research, NY

[†]Princeton University and Google Research, Princeton

[‡]Cornell University

36th Conference on Neural Information Processing Systems (NeurIPS 2022).

The goal of our paper is to provide a unifying analysis for when SGD/GD works. More specifically, we do this via first showing that Gradient Flow (GF) works and then extending this analysis to SGD and GD. Gradient Flow (GF) can be seen as a continuous time analogue of GD. In an idealized world, if one had access to the population loss, it turns out that convergence analysis for running gradient flow on population loss is somewhat simpler due to tools from continuous time analysis and PDEs. There has been several recent works [6, 14, 17] that have provided convergence analysis for GF even on non-convex objectives. The high level theme of this paper is to show that, under some mild/appropriate assumptions of population loss/objective and on the noise of gradient estimates, "if, GF converges on population loss, then SGD that uses one fresh example per iteration is successful at learning". Notice, that GF converging on population loss is a purely deterministic optimization problem. However, the fact that SGD works is a learning result that implies a sample complexity bound.

There have been past works that have aimed at providing convergence analysis for Gradient Descent (GD) starting from Gradient Flow (GF). Typical route to obtain a convergence analysis of GD starting from GF tries to think of GD updates as approximating GF path. Even with more sophisticated discretization schemes like Euler discretization, obtaining convergence for GD, starting from GF can be quite complex. In this paper, to show that when GF converges, SGD/GD also converges, we take a different approach. A key tool for proving convergence results for GF is by constructing so called Lyapunov potentials. In the literature of Ordinary Differential Equations (ODEs), when ODEs have regular enough convergence rates, one can show, so called converse Lyapunov theorems (see [34] for a nice survey of classic results) that state that when an ODE converges to stable solutions, there has to exist a corresponding Lyapunov potential. While convergence of GF in terms of sub-optimality is quite different from convergence in the ODE sense, in this paper, we first prove a converse Lyapunov style theorem for GF. Specifically, we show that when GF converges in terms of sub-optimality to a global minimum, then there has to exist a corresponding Lyapunov potential and using such potential, the rates can be recovered. This result becomes a starting point for our analysis. We show that if this Lyapunov potential (obtained from the converse Lyapunov style theorem) satisfies certain extra self-bounding regularity conditions, then one can show that GD and SGD algorithms converge in terms of sub-optimality when appropriate step sizes are used. Such convergence for SGD/GD happens even when the GF path and GD/SGD paths can be quite different.

We summarize our main contributions below:

- We prove a converse Lyapunov style theorem that shows that if gradient flow converges with rate specified by with an appropriate rate function, then there exists a corresponding Lyapunov potential that recovers this rate.

- We provide a geometric characterization for a given rate of convergence of gradient flow (ie. GF converges at a particular rate if and only if a specific geometric condition on objective holds.)

- There are problems for which GF converges at a specific rate but GD can be arbitrarily slow to converge.

- This motivates the necessity of additional conditions to ensure GD/SGD converges even when GF converges. We provide certain self-bounding regularity conditions on the Lyapunov potential, under which we show that GD converges. We also provide conditions on gradient estimate noise under which we show that SGD using these gradient estimates also converges.

- We instantiate our results for problems such KŁ functions, matrix square-root and phase retrieval, amongst other applications.

## 2 Setup

Given a continuously differentiable function and non-negative function $F : \mathbb{R}^d \mapsto \mathbb{R}$, our goal is to minimize $F(w)$. Without any loss of generality, we assume that $\min_w F(w) = 0$. First-order algorithms are popular for such optimization tasks. In the following, we formally describe the Gradient Descent and Stochastic Gradient Descent algorithm, and their continuous time counterpart called gradient flow.

**Gradient Descent (GD).** Gradient descent is the most popular iterative algorithm to minimize differentiable functions. Starting from an initial point $w_0$, GD on the function $F(w)$ performs the

following update on every iteration:

$$w_{t+1} \leftarrow w_t - \eta \nabla F(w_t), \tag{1}$$

where $\eta$ denotes the step size. After $T$ rounds, GD algorithm returns the point $\widehat{w}_T :=$ $\mathrm{argmin}_{s \leq T} F(w_s)$.

**Stochastic Gradient Descent (SGD).** Stochastic gradient descent (SGD) has been the method of choice for optimizing complex convex and non-convex learning problems in practice. In the learning setting, $F(w)$ corresponds to the unknown population loss and can be written as $F(w) = \mathbb{E}_{z \sim \mathcal{D}}[f(w; z)]$ where the expectation is taken with respect to samples $z$ drawn from an unknown distribution $\mathcal{D}$. SGD algorithm (mini-batch size 1) is an iterative algorithm that at every round $t \geq 0$, draws a fresh sample $z_t$ from $\mathcal{D}$ to compute a stochastic unbiased estimate $\nabla f(w_i; z_i)$ of the gradient $\nabla F(w_i)$, and performs the update

$$w_{t+1} \leftarrow w_t - \eta \nabla f(w_t; z_t) \tag{2}$$

where $\eta$ is the step size and $w_0$ denotes the initial point. After $T$ rounds, SGD algorithm returns $\widehat{w}_T$ by sampling a point uniformly at random from the set $\{w_1, \ldots, w_T\}$.

**Gradient Flow (GF).** Gradient flow from a point $w_0$ is continuous time process $(w(t))_{t \geq 0}$ that starts at $w(0) = w_0$ and evolves as

$$\frac{\mathrm{d}w(t)}{\mathrm{d}t} = -\nabla F(w(t)). \tag{3}$$

GF has been thought of as a continuous time analogue of GD and is popularly used to understand behavior of gradient based optimization algorithms in the limit, primarily due to its simplicity and lack of step size.

**Additional notation.** For a vector $w \in \mathbb{R}^d$, $w[j]$ denotes its $j$-th coordinate and $\|w\|$ denotes its Euclidean norm. For any $w_1, w_2 \in \mathbb{R}^d$, $\langle w_1, w_2 \rangle$ denotes their inner product. For a matrix $W$, $\sigma_d(W)$ and $\|M\|$ denotes its minimum singular value and spectral norm respectively. We define the set $\mathbb{R}^+$ to contain all non-negative real numbers. We use $\mathbf{1}_d$ to denote a $d$-dimensional vector of all 1s, and $\mathrm{I}_d$ to denote the identity matrix in $d$-dimensions. $\mathcal{N}(0, \sigma^2 \mathrm{I}_d)$ denotes $d$-dimensional Gaussian distribution with variance $\sigma^2 \mathrm{I}_d$. $\mathrm{Ber}(p)$ denotes the Bernoulli distribution with mean $p$.

For a function $f : \mathbb{R}^d \mapsto \mathbb{R}$, we denote the $p$-th derivative at the point $w$ by $\nabla^p f(w) \in \mathbb{R}^d$. We say that a real valued $f$ is monotonically increasing if $f' \geq 0$, and monotonically decreasing if $f' \leq 0$. The function $f$ is said to be $L$-Lipschitz if $f(w_1) - f(w_2) \leq L\|w_1 - w_2\|$ for all $w_1, w_2$. For a set of initial points $\mathcal{W}$, we denote $\mathrm{clo}(\mathcal{W})$ as its *closure* under GF, i.e. $\mathrm{clo}(\mathcal{W}) = \{w' : w' \text{ is in GF path of some } w_0 \in \mathcal{W}\}$.

## 3 Gradient Flow, Potentials and Geometry

Lyapunov potentials are a popular tool for understanding convergence of GF [38, 55, 57]. At an intuitive level, a Lyapunov potential is any non-negative function $\Phi$ that satisfies $\langle \nabla \Phi(w), -\nabla F(w) \rangle \leq 0$, i.e. $\Phi$ decreases along the GF paths of $F$. This monotonicity property helps to show asymptotic convergence of GF to stable points of the underlying objective. In our work, we consider potential functions for which the rate of change (decrease) of the potential along the GF path is related to the suboptimality of the objective at that point.

**Definition 1** (Admissible potentials). *A differentiable potential function $\Phi_g : \mathbb{R}^d \mapsto \mathbb{R}^+$ is admissible w.r.t. $F$ on a set $\mathcal{W}$ if there exists a monotonically increasing function $g : \mathbb{R}^+ \mapsto \mathbb{R}^+$ with $g(0) = 0$ such that for any $w \in \mathcal{W}$,[4]*

$$\langle \nabla \Phi_g(w), \nabla F(w) \rangle \geq g(F(w)). \tag{4}$$

Existence of potential functions of the above form can be used to provide rates of convergence for GF as show in the following theorem.

---

[4]Whenever not specified, we assume that $g(z) = z$. The function $\Phi$ denote the potential function $\Phi_g$ with $g(z) = z$.

**Theorem 1** (From potentials to gradient flow). *Let $\mathcal{W}$ be a set of initial points that we want to consider, and let $\Phi_g$ be an admissible potential w.r.t. $F$ on the set $\mathrm{clo}(\mathcal{W})$. Then, for any initialization $w_0 \in \mathcal{W}$, the point $w(t)$ on the GF path with $w(0) = w_0$ satisfies for any $t \geq 0$,*

$$g(F(w(t))) \leq \frac{\Phi_g(w_0)}{t}.$$

The idea that admissible potential functions imply convergence rates for GF has appeared in various forms in the prior literature [7, 37, 55, 57]. As an example, consider the potential function $\Phi(w) = \|w - w^*\|/2$. Notice that $\Phi$ is an admissible potential for any $F$ that is convex with $g(z) = z$. This is because convexity implies that (4) is true for any $w$. Hence, using Theorem 1 we get a $\|w - w^*\|^2/2t$ rate of convergence for GF on any convex objective.

Our main result in this section is to establish a converse Lyapunov style theorem–that given a rate, finds a potential function corresponding to that rate. We start by defining admissible rate functions.

**Definition 2** (Admissible rate functions). *A function $R : \mathbb{R}^d \times \mathbb{R} \mapsto \mathbb{R}^+$ is an admissible rate function w.r.t. $F$ if for any $w \in \mathbb{R}^d$,*

($a$) $R(w, t)$ *is a non-increasing function of $t$ such that $\lim_{t \to \infty} R(w, t) = 0$.*

($b$) $R$ *satisfies the relation:* $\int_{t=0}^{\infty} \left( \frac{\partial R(w,t)}{\partial t} + \langle \nabla R(w,t), \nabla F(w) \rangle \right) \mathrm{d}t \geq 0.$

**Remark 1.** *In order to simplify the task of checking whether a given rate is admissible, note that Definition 2-(b) is satisfied whenever the condition*

$$\frac{\partial R(w,t)}{\partial t} + \langle \nabla R(w,t), \nabla F(w) \rangle \geq 0$$

*holds for every $w$ as $t \to 0$. Many rate functions, e.g. $R(w, t) = F(w)e^{-t}$ and $F$ being KŁ, in fact satisfy this condition for every $w, t \geq 0$.*

*Furthermore, also note that Definition 2-(b) is satisfied whenever the rate function is such that $R(w(s), t) \leq R(w, s + t)$ for all $s, t \geq 0$ and $w \in \mathbb{R}^d$, which may be an easier to check condition, e.g. when $R(w, t) = F(w)e^{-t}$.*

We utilize admissible rate functions to characterize behavior of GF on $F$. Before we proceed, let us motivate the two properties above. Property ($a$) is natural for any rate function and captures the fact that running GF for more time leads to better guarantees. Property ($b$), while seeming a bit mysterious, characterizes the compatibility of the rate function w.r.t. gradient flow dynamics. For interpretation consider the relaxed version given in Remark 1 which implies property-(b). Here, the condition that $R(w(s), t) \leq R(w, s+t)$ for all $s, t \geq 0$ and $w \in \mathbb{R}^d$ simply captures the fact that having additional information about the GF path should only improve the rate. Note that $R(w(0), s + t)$ corresponds to an upper bound on the sub-optimality at $w(s + t)$ and $R(w(s), t)$ corresponds to an upper bound on the same quantity but with the additional information that $w(s)$ is a point on the GF path. We remark that for any rate function $R$, it is easy to construct a new rate function $\bar{R}$ that always satisfies this condition (hence, property ($b$)) by defining $\bar{R}(w, t) = \min_{s \geq 0} R(w', t + s)$ where $w'$ is any point such that the point $w$ lies on the GF path from $w'$ at time $s$. Furthermore, the function $R(w, t) = F(w(t))$ is always an admissible rate function. All the rate functions appearing in this paper satisfy both properties ($a$) and ($b$).

Our next result shows that admissible rate functions for GF can be used to construction admissible potentials w.r.t. $F$.

**Theorem 2** (From gradient flow to potentials). *Let $\mathcal{W} \subseteq \mathbb{R}^d$ be any set of initial points that we want to consider, and $R$ be an admissible rate function w.r.t. all GF paths originating from any point in $\mathcal{W}$. Further, suppose that for any $w_0 \in \mathcal{W}$, the point $w(t)$ on the GF path satisfies $F(w(t)) \leq R(w_0, t)$, then the function $\Phi_g$ defined as*

$$\Phi_g(w) = \int_{t=0}^{\infty} g(R(w, t)) \, \mathrm{d}t \tag{5}$$

*is an admissible potential w.r.t. $F$ on the set $\mathrm{clo}(\mathcal{W})$, for any differentiable and monotonically increasing function $g : \mathbb{R}^+ \mapsto \mathbb{R}^+$ that satisfies $\int_{t=0}^{\infty} g(R(w, t)) \, \mathrm{d}t < \infty$ and $\int_{t=0}^{\infty} g'(R(w, t)) \|\nabla R(w, t)\| \, \mathrm{d}t < \infty$ for every $w \in \mathrm{clo}(\mathcal{W})$.*

As an illustration on how to apply Theorem 2, assume that for $F$ the rate for GF is $R(w,t) = F(w)e^{-t}$ For instance, we already know that such a rate holds when $F$ is PŁ. For this rate, by choosing $g(z) = z$, we get that the function $\Phi_g(w) = \int_{t=0}^{\infty} F(w)e^{-t}\,\mathrm{d}t = F(w)$ is an admissible potential w.r.t. $F$. We provide more examples in Section 5.

Theorem 1 and Theorem 2 are, in a sense, converse of each other. Theorem 1 shows that the existence of an admissible potential function implies a rate of convergence for GF. On the other hand, Theorem 2 shows how to construct admissible potentials starting from the fact that GF has a rate. One might wonder whether there always exist a Lyapunov function, more specifically a $g$ function above, such that the rate implied by the constructed potential in Theorem 1 matches the rate that we started with for Theorem 2, i.e. $R(w,t) \approx g^{-1}(\Phi_g(w)/t)$. We answer this in the positive for rate functions that are of the product form.

**Corollary 1.** *Let $\mathcal{W} \subseteq \mathbb{R}^d$ be any set of initial points that we want to consider, and $R$ be an admissible rate function w.r.t. all GF paths originating from points in $\mathcal{W}$. Additionally, suppose $R$ has the product form $R(w,t) = h(w)r(t)$ where $h$ is differentiable and $r$ is a non-increasing function that satisfies $r(t) \leq \lambda|r'(t)|\max\{1,t\}$ for any $t \in \mathbb{R}$ (where $\lambda$ is a universal constant). Furthermore, suppose that for any $w_0 \in \mathcal{W}$, the point $w(t)$ on the GF path satisfies $F(w(t)) \leq R(w_0, t)$. Then, there exists a monotonically increasing function $g : \mathbb{R}^+ \mapsto \mathbb{R}^+$ such that the potential $\Phi_g(w)$ constructed in Theorem 2 using $g$, when plugged in Theorem 1, implies that GF has the rate*

$$F(w(t)) \leq \max_{w \in \mathcal{W}} R(w, t/\log^2(t))$$

*for any initialization $w(0) \in \mathcal{W}$.*

### 3.1 Geometric Interpretation

The definition of an admissible potential comes with a geometric condition on the function $F$ given in (4). Since Theorem 2 constructs admissible potentials, when a rate $R(w,t)$ holds for GF it suggests that the geometric property in (4) holds for the objective function $F$. As an example, say GF on $F$ satisfies the rate $R(w,t) = F(w)e^{-t}$. From Theorem 2, we note that $\Phi_g(w) = F(w)$ is an admissible potential w.r.t $F$ with $g(z) = z$. This implies the geometric property

$$\langle \nabla F(w), \nabla F(w) \rangle \geq F(w) \tag{6}$$

holds for $F$ whenever GF has rate $R(w,t) = F(w)e^{-t}$. On the other, we know that whenever (6) holds the function $\Phi(w) = F(w)$ satisfies (4) and is thus an admissible potential for $F$ (with $g(z) = z$), and hence Theorem 1 implies the rate of $F(w)/t$, which is equivalent to the rate $F(w)e^{-t}$ (c.f. Lemma 6). This implies an equivalence between the rates $R(w,t) = F(w)e^{-t}$ and the geometric property (6). We formalize this in the following.

**Proposition 1.** *The following two properties are equivalent:*

(a) *For any $w(0) \in \mathbb{R}^d$ and $t \geq 0$, GF has the rate $F(w(t)) \leq F(w(0))e^{-\lambda t}$,*

(b) *$F(w)$ satisfies the Polyak-Łojasiewicz (PL) property i.e. $\lambda F(w) \leq \|\nabla F(w)\|^2$,*

*for any $\lambda \geq 0$. Theorem 2 implies $(b)$ and yields the potential function $\Phi(w) = F(w)$.*

A similar equivalence also holds for the more general class of KŁ functions. We defer this result to Proposition 3 in Section E.1. In the following, we show a correspondence between the rate $R(w,t) = \frac{\|w(0)-w^*\|^2 - \|w(t)-w^*\|^2}{2t}$, and linearizability—a condition that is weaker than convexity but is sufficient for the corresponding rate of convergence for GF.

**Proposition 2.** *The following two properties are equivalent:*

(a) *For any $w(0) \in \mathbb{R}^d$ and $t \geq 0$, GF has the admissible rate $F(w(t)) \leq \lambda\frac{\|w(0)-w^*\|^2 - \|w(t)-w^*\|^2}{2t}$,*

(b) *$F(w)$ is linearizable w.r.t. $w^*$ i.e. $F(w) \leq \lambda\langle \nabla F(w), w - w^* \rangle$,*

*for any $\lambda \geq 0$.*

More generally, the equivalence between GF rates and the corresponding geometry on $F$ can be characterized as follows.

**Remark 2.** *GF on $F$ enjoys the admissible rate $R(w,t) = g^{-1}(\Phi_g(w)/t)$ if and only if $F$ has the geometric property $\langle \nabla \Phi_g(w), \nabla F(w) \rangle \geq g(F(w))$.*

# 4 Stochastic Gradient Descent and Gradient Descent

GD can be thought of as an approximate discretization of gradient flow. Thus, for problems where GF converges with a given rate $R$, one may try to get convergence guarantees for GD from an initial point $w_0$ by bounding the distance between the GD and GF trajectories starting from $w_0$. This is exactly the approach taken in prior works [27, 39, 56, 52, 62, 21]. However, coming up with non-vacuous bounds on the distance between corresponding GF and GD iterated is often quite challenging and requires much stronger assumptions on the underlying objective. In fact there are cases where both GF and GD converge to the same global minimum but their paths can be quite far away from each other. We take a different approach for proving convergence of GD/SGD which directly relies on the properties of corresponding potential for $F$. In the following theorem, we note that further assumption on top of the premise that GF has a rate are required, to even hope that GD succeeds.

**Theorem 3.** *For any integer $T_0 > 0$, there exists a continuously differentiable convex function $F$ for which $\min_w F(w) = 0$ and $w^* = 0$ is the unique minimizer, such that:*

(a) *$\Phi(w) = \|w\|^2/2$ is an admissible potential for $F$. Thus, Theorem 1 implies that for any initial point $w_0$, the point $w(t)$ on its GF path satisfies $F(w(t)) \le \frac{\|w_0\|^2}{2t}$.*

(b) *There exists an initial point $w_0$ with $\|w_0\| \le 1$ and $F(w_0) \le 2$ such that GD fails to find an $1/10$-suboptimal solution for any step size $\eta$ within $t \le T_0$ steps.*

Before giving our exact assumptions and the convergence bounds, we provide the intuition behind how admissible potentials can be used for analyzing GD (or SGD). Let the sequence of iterates generated by GD algorithm be given by $\{w_t\}_{t \ge 0}$, $g(z) = z$ and $\Phi$ be an admissible potential w.r.t. $F$. For any time $t$, the second-order Taylor's expansion of the potential $\Phi$ implies that

$$\Phi(w_{t+1}) \le \Phi(w_t) + \langle \nabla\Phi(w_t), w_{t+1} - w_t \rangle + (w_{t+1} - w_t)^T \nabla^2\Phi(\widetilde{w_t})(w_{t+1} - w_t)$$

$$\le \Phi(w_t) - \eta\langle \nabla\Phi(w_t), \nabla F(w_t) \rangle + \eta^2(\nabla F(w_t))^T \nabla^2\Phi(\widetilde{w_t})(\nabla F(w_t)),$$

where $\widetilde{w_t} = \beta w_t + (1 - \beta)w_{t+1}$ for some $\beta \in [0, 1]$, and the second line follows by plugging the GD update $w_{t+1} = w_t - \eta\nabla F(w_t)$. Rearranging the terms, we get that

$$\langle \nabla\Phi(w_t), \nabla F(w_t) \rangle \le \frac{\Phi(w_t) - \Phi(w_{t+1})}{\eta} + \eta(\nabla F(w_t))^T \nabla^2\Phi(\widetilde{w_t})(\nabla F(w_t)), \qquad (7)$$

The key idea that enables us to get performance guarantees for GD is that the linear term in the left hand side above upper bounds the suboptimality of $F$ at the point $w_t$ since $\Phi$ is an admissible potential w.r.t. $F$. In particular, the condition (4) implies that

$$\langle \nabla\Phi(w_t), \nabla F(w_t) \rangle \le F(w_t).$$

Using the above relation in (7), telescoping $t$ from 0 to $T - 1$, and dividing by $T$, we get that

$$\frac{1}{T}\sum_{t=0}^{T-1} g(F(w_t)) \le \frac{\Phi(w_0) - \Phi(w_T)}{\eta T} + \frac{\eta}{T} \cdot \sum_{t=0}^{T-1}(\nabla F(w_t))^T \nabla^2\Phi(\widetilde{w_t})(\nabla F(w_t)). \qquad (8)$$

Thus, we can bound the expected suboptimality of the point $\widehat{w} \sim \text{Uniform}(\{w_0, \ldots, w_{T-1}\})$ returned by the GD algorithm after $T$ steps, whenever the second order term in the bound (8) is well behaved. For example, if $(\nabla F(w_t))^T \nabla^2\Phi(\widetilde{w_t})(\nabla F(w_t)) \le K$ for any $w_t, w_{t+1}$ and $\widetilde{w_t}$, we immediately get that

$$\frac{1}{T}\sum_{t=0}^{T-1} g(F(w_t)) \le \frac{\Phi(w_0) - \Phi(w_T)}{\eta T} + \eta K = O\left(\frac{1}{\sqrt{T}}\right),$$

for $\eta = O(1/\sqrt{T})$. While the above holds for a very simplified setup, the intuition can be extended to more general cases as well. Below we present two regularity conditions that are sufficient to show convergence of GD.

**Assumption 1.** *There exists a monotonically increasing function $\psi : \mathbb{R}^+ \mapsto \mathbb{R}^+$ such that $\|\nabla F(w)\|^2 \le \psi(F(w))$ for any point $w \in \mathcal{W}$.*

**Assumption 2.** *The potential function $\Phi$ is second-order differentiable, and there exists a monotonically increasing function $\rho : \mathbb{R}^+ \mapsto \mathbb{R}^+$ such that $\|\nabla^2\Phi(w)\| \le \rho(\Phi(w))$ at any point $w \in \mathcal{W}$.*

We will refer to the above conditions on $F$ and $\Phi$ as self-bounding regularity conditions. The following theorem provides convergence guarantees for GD when an admissible potential exists and the above assumptions are satisfies.

**Theorem 4** (GD convergence guarantee). *Let $\Phi_g$ be an admissible potential w.r.t. $F$. Assume that $F$ satisfies Assumption 1 and $\Phi_g$ satisfies Assumption 2. Then, for any $T \geq 0$ and setting $\eta$ appropriately, the point $\widehat{w}_T$ returned by GD algorithm has the convergence guarantee[5]*

$$g(F(\widehat{w}_T)) = O\left(\frac{1}{\sqrt{T}}\right). \tag{9}$$

*Furthermore, if the function $\lambda(z) := \frac{\psi(z)}{g(z)}$ is monotonically increasing in $z$, then for a different appropriate choice of $\eta$,*

$$g(F(\widehat{w}_T)) = O\left(\frac{1}{T}\right). \tag{10}$$

Let us consider an example. Suppose that gradient flow on $F$ achieves the admissible rate $R(w, t) = \left(\|w - w^*\|^2 - \|w(t) - w^*\|^2\right)/2t$. This implies that $F$ is linearizable (Proposition 2), and thus $\Phi_g(w) = \|w - w^*\|^2/2$ is an admissible potential for $F$ with $g(z) = z$ as it clearly satisfies (4). However, as we saw in Theorem 3 just existence of such a rate function does not imply the GD will succeed and we need to make further assumptions. Notice that in this case $\Phi_g(w)$ satisfies Assumption 2 with $\rho(z) = 1$. If we further assume that $F$ is $L$-Lipschitz, then Assumption 1 is satisfied with $\psi(z) = L^2$. Hence, applying Theorem 4 for this setting, we get that GD has convergence rate $F(\widehat{w}_T) = O(\|w - w^*\|L/\sqrt{T})$. Instead if $F$ was $H$-smooth, Assumption 1 is satisfied with $\psi(z) = 4Hz$ and $\psi(z)/g(z) = 4H$ is a monotonically increasing function and thus using (10), we get that GD has the convergence rate $F(\widehat{w}_T) = O(H\|w - w^*\|^2/T)$. Notice that both of these rates are optimal for GD under the Lipschitz/Smoothness assumptions on $F$, and the fact that $F$ is linearizable [47]. On similar lines, using the rates for GF convergence on PL/KŁ functions, we can also recover optimal convergence rates for GD under appropriate smoothness assumptions on $F$.

We next consider the convergence of SGD algorithm. Recall that at the iterate $w_t$, SGD performs the updated using $\nabla f(w_t, z_t)$, a stochastic and unbiased estimate of $\nabla F(w_t)$. Of course, unless one has some form of control over the distribution of $\nabla f(w_t, z_t)$, one cannot hope to prove any convergence guarantees of SGD. To this end, we make the following regularity assumption on the noise in $\nabla f(w, z_t)$ while estimating $\nabla F(w)$.

**Assumption 3** (Noise regularity). *There exists a monotonically increasing function $\chi : \mathbb{R}^+ \mapsto \mathbb{R}^+$ such that for any point $w$, the gradient estimate $\nabla f(w, z)$ satisfies*

$$\Pr\left(\|\nabla f(w; z) - \nabla F(w)\|^2 \geq t \cdot \chi(F(w))\right) \leq e^{-t}.$$

Assumption 3 is quite general, and can be specialized by appropriately setting the function $\chi$ to model various stochastic optimization problem settings observed in practice. For example, the classical stochastic optimization setting in which $\nabla f(w; z) = \nabla F(w) + \varepsilon_t$ where $\varepsilon_t$ is a sub-Gaussian random variable with mean $0$ and variance $\sigma^2$ is captured by the above assumption when $\chi(z) = \sigma^2$ [46]. However, it turns out that for many interesting ML problems, the noise typically scales with the function value [58, 59].

**Theorem 5** (SGD convergence guarantee). *Let $\Phi_g$ be an admissible potential w.r.t. $F$. Assume that $F$ satisfies Assumption 1, $\Phi_g$ satisfies Assumption 2 and the stochastic gradient estimates $\nabla f(w; z)$ satisfy Assumption 3. Then, for any $T \geq 0$ and setting $\eta$ appropriately, the point $\widehat{w}_T$ returned by SGD algorithm has the convergence guarantee[1]*

$$g(F(\widehat{w}_T)) = \widetilde{O}\left(\frac{1}{\sqrt{T}}\right).$$

*with probability at least $0.7$ over the randomization of the algorithm and stochastic gradients.*

**Remark 3.** *In most classic settings, one expects a $1/\sqrt{T}$ rate for SGD [10]. However, in cases where $\Phi_g$ is an admissible potential and $g(z) = o(z)$, Theorem 5 seems to suggest a $g^{-1}(1/\sqrt{T})$ rate of convergence which is faster than $1/\sqrt{T}$. This is where the self-bounding regularity conditions play an important role. As an example for PŁ style rates, one can show that $F(w)^p$ is an admissible potential with $g(z) = z^p$ for any $p$. However, the self-regularity conditions are not satisfied unless $p \geq 1$. Setting $p = 1$ recovers the $1/\sqrt{T}$ rate of SGD for PŁ functions which is optimal [1].*

---

[5]The $O(\cdot)$ notation here hides initialization and problem dependent constants fully specified in the Appendix.

# 5 Examples: From Gradient Flow to Gradient Descent

So far, we discussed classical examples like PŁ functions, convex functions, etc. At a high level, in order to show convergence of SGD for these problems, we first establish an admissible rate of convergence for GF, which implies an admissible potential that is used to show convergence of SGD. We next extend this approach for other more complex stochastic non-convex optimization problems.

## 5.1 Phase retrieval

In the phase retrieval problem [11, 16, 53], we wish to reconstruct a hidden vector $w^\star \in \mathbb{R}^d$ with $\|w^\star\| = 1$ using phaseless observations $\mathcal{S} = \{(a_j, y_j)\}_{j \leq T}$ of the form $y_j = \langle a_j, w^\star \rangle^2$ where $a_j \sim \mathcal{N}(0, \mathrm{I}_d)$. The classical approach to recover $w^\star$ is by using the per-sample loss function $f_{\mathrm{pr}}(w; (a_j, y_j)) = ((a_j^\top w)^2 - y_j)^2$ for which the corresponding population loss is given by

$$F_{\mathrm{pr}}(w) = \mathbb{E}[f_{\mathrm{pr}}(w; (a, y))] = \mathbb{E}_{a \sim \mathcal{N}(0, \mathrm{I}_d)}\Big[\big((a^\top w)^2 - (a^\top w^\star)^2\big)^2\Big]. \tag{11}$$

$F_{\mathrm{pr}}$ is non-convex, and has stationary points (and local minima) that do not correspond to the global minima. In the following, we provide convergence guarantees for GD algorithm on $F_{\mathrm{pr}}$, and SGD algorithms that computes stochastic gradient estimates using $\mathcal{S}$. We first note that $F_{\mathrm{pr}}$ satisfies self-bounding regularity conditions, and GF has a rate of convergence to global minimizers of $F_{\mathrm{pr}}$.

**Lemma 1.** $F_{\mathrm{pr}}$ *satisfies Assumption 1. Furthermore, for any initial point $w_0$, the point $w(t)$ on its gradient flow path satisfies*

$$F_{\mathrm{pr}}(w(t)) \leq \min\Big\{F_{\mathrm{pr}}(w_0), F_{\mathrm{pr}}(w_0)e^{-t + \frac{1}{\langle w_0, w^\star \rangle^2}}\Big\} =: R_{\mathrm{pr}}(w(0), t).$$

*Furthermore, the function $R_{\mathrm{pr}}$ above is an admissible rate of convergence w.r.t. $F_{\mathrm{pr}}$.*

The above rate follows from independently analyzing the parallel and perpendicular components $\langle w, w^\star \rangle$ and $\|w\|^2 - \langle w, w^\star \rangle$ respectively. Our main tool for getting the convergence guarantee for GD / SGD is to utilize Theorem 2 to get an admissible potential w.r.t. $F_{\mathrm{pr}}$, which can be plugged in Theorem 4 and 5 to get the corresponding rates.

**Theorem 6.** *Consider the phase retrieval objective $F_{\mathrm{pr}}$ given in (11). For any initial point $w_0$ and $T \geq 1$, setting $\eta$ appropriately,*

(a) *The point $\widehat{w}_T$ returned by GD starting from $w_0$ satisfies $F_{\mathrm{pr}}(\widehat{w}_T) = O\big(\min\big\{\frac{1}{T}, e^{-O(T - t_0)}\big\}\big)$ for all $T \geq t_0$, where $t_0$ is a $w_0$ dependent constant.*

(b) *The point $\widehat{w}_T$ returned by SGD starting from $w_0$ and using stochastic gradient estimates for which Assumption 3 holds, satisfies $F_{\mathrm{pr}}(\widehat{w}_T) = \widetilde{O}\Big(\frac{1}{\sqrt{T}}\Big)$ with probability at least $0.7$.*

The $O(\cdot)$ notation above hides $w_0$ dependent constants which we specify in the Appendix. Our rate for GD above matches the best known result in the literature in terms of the dependence on $T$ [16]. To the best of our knowledge, ours is also the first convergence analysis of SGD under arbitrary noise conditions satisfying Assumption 3. While this rate is optimal under certain noise conditions, e.g. when $\chi(z) = \sigma^2$, further improvements are possible when $\chi$ is favorable. For example, suppose the stochastic gradient estimates were computed using samples from $\mathcal{S}$ by taking a fresh sample for each estimate, i.e. $\nabla f(w; (a, y)) = 4((a^\top w)^2 - y)(a^\top w)w$. In this case, the stochastic gradient satisfy Assumption 3 with $\chi(z) = \min\{\sqrt{z}, c\}$ where $c$ is a universal constant (c.f. Candes et al. [11, Lemma 7.4, 7.7]). While, our framework implies that this SGD algorithm (computing estimates using samples) converges at the rate of $1/\sqrt{T}$, this rate can be improved further [16], and we defer the refined analysis for future research.

## 5.2 Initialization specific rates

In many applications, GF is only known to converge from nice enough initial points that satisfy certain properties. In this section, we extend show how to use our tools for establishing convergence of GD/SGD for such problems, and consider matrix square root as an example. We first provide the following general utility lemma that shows how to construct admissible potentials when the rate for GF from $w_0$ holds only when $w_0$ satisfies a certain property characterized by $h(w_0) \geq 0$.

**Lemma 2.** *Let $h : \mathbb{R}^d \mapsto [0,1]$ be a continuously differentiable function, and suppose that for any point $w$ for which $h(w) > 0$, GF with $w(0) = w$ has rate $F(w(t)) \leq R(w,t)$ where $R(w,\cdot)$ is a monotonically decreasing function in $t$. Furthermore, suppose that $F(w) \leq R(w,0)$, $F$ satisfies Assumption 1, $R(w, h(w)t)$ is an admissible rate function w.r.t. $F$, and for any $w$,*

(a) *the function $\Gamma(w) := \int_{t=0}^{\infty} R(w,t)\,\mathrm{d}t$ is continuously differentiable, and $\max\{\|\nabla\Gamma(w)\|, \|\nabla^2\Gamma(w)\|\} \leq \lambda(\Gamma(w))$ where $\lambda$ is a positive, monotonically increasing function.*

(b) *$\max\{\|\nabla h(w)\|, \|\nabla^2 h(w)\|\} \leq \pi(\Gamma(w))$ where $\pi$ is a positive, monotonically increasing function.*

(c) *$(h(w) - h(w^*))^2 \leq \mu(\Gamma(w))$ where $\mu$ is a positive, monotonically increasing function with the property that $k\mu(z) \leq \mu(kz)$ for any $k \geq 1$.*

*Then, the function $\Phi_g(w) = \Gamma(w)/h(w)$ is an admissible potential w.r.t. $F$ with $g(z) = z$, and satisfies the self-bounding regularity condition in Assumption 2.*

### 5.2.1 Matrix square root

In the matrix square root problem [19, 28], we are given a positive definite and symmetric matrix $M \in \mathbb{R}^{d \times d}$ with $\sigma_d(M) > 0$, and wish to find a symmetric $W \in \mathbb{R}^{d \times d}$ that minimizes the objective

$$F_{\mathrm{ms}}(W) = \|M - W^2\|_F^2. \tag{12}$$

$F_{\mathrm{ms}}$ is non-convex in $W$, and has spurious stationary points. In the following, we provide convergence guarantees for GD/SGD algorithm on $F_{\mathrm{ms}}$. We first note that $F_{\mathrm{ms}}$ satisfies self-bounding regularity conditions, and GF on $F_{\mathrm{ms}}$ converges to the global minimizer when the initial point $w_0$ satisfies additional assumptions. We capture these initial conditions using the function $h_{\mathrm{ms}}$ defined as

$$h_{\mathrm{ms}}(W) = \sigma\big(\phi(W^2) - \alpha\big), \tag{13}$$

where the function $\phi(Z) := \frac{-1}{\gamma}\log(\mathrm{tr}(e^{-\gamma Z}) + e^{-16\alpha\gamma})$, $\alpha = \sigma_{\min}(M)/1600$, $\gamma = \log(d+1)/\alpha$, and $\sigma$ denotes a smoothened version of the indicator function given by $\sigma(z) = \{0$ if $z \leq 0,\ \frac{2}{\alpha^2}z^2$ if $0 \leq z \leq \frac{\alpha}{2},\ -\frac{2}{\alpha^2}z^2 + \frac{4}{\alpha}z - 1$ if $\frac{\alpha}{2} \leq z \leq \alpha$, and $1$ if $\alpha \leq z\}$.

**Lemma 3.** *$F_{\mathrm{ms}}$ satisfies Assumption 1. Furthermore, for any initial point $W_0$ for which $h_{\mathrm{ms}}(W_0) > 0$, the point $W(t)$ on its GF path satisfies*

$$F_{\mathrm{ms}}(W(t)) \leq F_{\mathrm{ms}}(W_0)\exp(-16\alpha t) =: R_{\mathrm{ms}}(W(0), t),$$

*where $\alpha = \sigma_{\min}(M)/1600$, $\gamma = \log(d+1)/\alpha$ and the function $h_{\mathrm{ms}}$ is defined in (13).*

The above rate follows from directly solving the PDE associated with the gradient flow on the underlying objective. Lemma 3 provides conditions on $W_0$ under which the GF path converges with the rate function $R_{\mathrm{ms}}$. Our main tool for showing the convergence of GD / SGD is by using Lemma 2 to get admissible potentials. Note that the function $h_{\mathrm{ms}}$ takes values in $[0, 1]$, is continuously differentiable, and as we show in the appendix satisfies all the required self-bounding regularity conditions in Lemma 2. Thus, Lemma 2 provides an admissible potential w.r.t. $F_{\mathrm{pr}}$ which can be used to get the following rates.

**Theorem 7.** *Consider the matrix square root objective $F_{\mathrm{ms}}$ given in (12). For any $\kappa > 0$, initial point $W_0$ for which $h_{\mathrm{ms}}(W_0) > 0$ and setting $\eta$ appropriately,*

(a) *The point $\widehat{W}_T$ returned by GD starting from $W_0$ satisfies $F_{\mathrm{ms}}(\widehat{W}_T) = O\big(\min\{\frac{1}{T}, e^{-O(T-t_0)}\}\big)$ for all $T \geq t_0$, where $t_0$ is a $w_0$ dependent constant.*

(b) *The point $\widehat{W}_T$ returned by SGD starting from $W_0$ and using stochastic gradient estimates for which Assumption 3 holds, satisfies $F_{\mathrm{ms}}(\widehat{W}_T) = \widetilde{O}\big(\frac{1}{\sqrt{T}}\big)$ with probability at least $0.7$.*

The $O(\cdot)$ notation above hides $W_0$ dependent constants which we specify in the Appendix. Our rate for GD above matches the best known result in the literature in terms of the dependence on $T$

[28]. Ours is also the first convergence analysis of SGD under arbitrary noise conditions satisfying Assumption 3. Note that the classical stochastic optimization setting in which $\nabla f_{\mathrm{ms}}(w; z) = 2(W^2 - M)W + 2W(W^2 - M) + \varepsilon_t$ where $\varepsilon_t$ is a sub-Gaussian random variable with mean $0$ and variance $\sigma^2$ satisfies Assumption 3 with $\chi(z) = \sigma^2$, and as a result of Theorem 7, we get that SGD converges at the rate of $1/\sqrt{T}$. To the best of our knowledge, convergence of SGD in the stochastic optimization setting for matrix square root problem was not known before.

## 6    Conclusion

In this paper, we provide a new framework for establishing performance guarantees for SGD in stochastic non-convex optimization. We introduce admissible potentials, and use them to get finite-time convergence guarantees for SGD. We also provide a method for constructing such admissible potentials using the rate function with which gradient flow converges on the underlying non-convex objective, provided that this rate function satisfies additional admissibility conditions. Thus, informally speaking, our results suggest that whenever gradient flow has an admissible rate of convergence and additional regularity conditions hold, SGD succeeds in minimizing the underlying non-convex objective (with the rate given in Theorem 5). We discuss some extensions and open problems below:

- Contrary to the prior approaches [27, 39, 56, 52, 62, 21], our convergence proof for SGD does not proceed by showing that the corresponding paths of SGD and gradient flow dynamics are point-wise close to each other. In fact, the example in Theorem 3 suggests that this may not be true even for convex functions, since for that example, gradient flow converges to minimizers but SGD diverges away from good solution. Our key technique is to use admissible potentials, that satisfy (4) w.r.t. gradient flow dynamics, to analyze discrete time algorithms like SGD.

- Our framework is motivated by Lyapunov analysis of dynamical systems [12, 15, 18, 57]. The property (4) in fact implies that any admissible potential is a Lyapunov potential w.r.t. the gradient flow dynamics on the underlying non-convex loss. It would be interesting to explore if techniques from the Lyapunov analysis of dynamical systems can be used to improve our rates further, or to relax various regularity and admissibility assumptions that we assume for our results. In particular, it would be interesting to explore how to extend our framework for non-smooth non-convex stochastic optimization.

- While we restricted ourselves to GD in the paper, our framework can be easily extended to analyze mirror descent algorithms (to get improved dependence on the problem geometry), by modifying the admissibility condition (4) to hold w.r.t. gradient flow dynamics in the dual space (mirror space). Furthermore, we can also extend our framework to other first-order algorithms like acceleration, momentum, etc., by changing (4) to hold w.r.t. the corresponding continuous time dynamics for these algorithms [36, 52, 48].

- Theorem 2 gives a construction of admissible potentials using the rate function $R$ for gradient flow on the underlying objective. However, the convergence bound for SGD in Theorem 5 holds only when this constructed potential satisfies additional self-bounded regularity conditions in Assumption 2. In order to get an end-to-end result, it would be interesting to explore what structural conditions on the rate function $R$ implies that the obtained potential satisfies Assumption 2.

In the paper, we demonstrate the generality of our framework by considering various non-convex stochastic optimization problems including PŁ/KŁ functions, phase retrieval and matrix square root, and show that admissible rate functions and the corresponding admissible potentials can be easily obtained by explicitly solving the partial differential equation associated with gradient flow; hence getting rates of convergence for SGD for these problems. Looking forward, it would be interesting to apply our framework for other non-convex stochastic optimization problems appearing in machine learning, and in particular deep learning.

**Acknowledgements**

AS thanks Robert D. Kleinberg for useful discussions. KS acknowledges support from NSF CAREER Award 1750575. JDL acknowledges support of the ARO under MURI Award W911NF-11-1-0304, the Sloan Research Fellowship, NSF CCF 2002272, NSF IIS 2107304, NSF CIF 2212262, ONR Young Investigator Award, and NSF CAREER Award 2144994. CD acknowledges support from NSF CAREER Award 2046760.

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
