# Contents of Appendix

## A   Preliminaries

In the following, we provide some basic definitions, probabilistic inequalities, and technical results.

**Definition 3** (*L*-Lipschitz function). *A function $F : \mathbb{R}^d \mapsto \mathbb{R}$ is said to be L-Lipschitz if for any $w_1, w_2$, $|F(w_1) - F(w_2)| \le L\|w_1 - w_2\|$.*

**Definition 4** (*H*-smooth functions). *A differentiable function $F : \mathbb{R}^d \mapsto \mathbb{R}$ is said to be H-Lipschitz if for any $w_1, w_2$,*

$$F(w_2) \le F(w_1) + \langle \nabla F(w_1), w_2 - w_1 \rangle + \frac{H}{2}\|w_2 - w_1\|^2.$$

**Definition 5** ($\lambda$-Linearizable). *A function $F(w)$ is $\lambda$-Linearizable if there exists a $w^* \in \arg\min F(w)$ such that for every point $w \in \mathbb{R}^d$,*

$$F(w) - F(w^*) \le \lambda \langle \nabla F(w), w - w^* \rangle.$$

**Lemma 4** (Azuma's inequality). *Let $\{X_t\}_{t \ge 0}$ be a super-martingale sequence such that for any $t \ge 0$, $A_t \le X_{t+1} - X_t \le B_t$ where $A_t$ and $B_t$ are $\mathcal{F}_t$-measurable, and satisfy $|B_t - A_t| \le c_t$. Then, for any $\gamma > 0$,*

$$\Pr(X_t - X_0 \ge \gamma) \le \exp\left(-\frac{\gamma^2}{2\sum_{t=1}^n c_t^2}\right).$$

The next technical lemma shows that $F(w(t))$ monotonically decreases along any GF path.

**Lemma 5.** *Let $w_0$ be any initial point. Then, for any $t \ge 0$, the point $w(t)$ on the GF path with $w(0) = w_0$ satisfies $F(w(t)) \le F(w(0))$.*

**Proof.** Fix $w(0) = w_0$ and define the function $\ell(t) = F(w(t))$, where $w(t)$ is on the GF path from $w_0$ at time $t$. Using Chain rule, we note that

$$\frac{\mathrm{d}g(t)}{\mathrm{d}t} = \langle \nabla F(w(t)), \frac{\mathrm{d}w(t)}{\mathrm{d}t} \rangle = -\|\nabla F(w(t))\|^2,$$

where the last equality holds from the definition of GF in (3). The above implies that $g(t) = F(w(t))$ is monotonically increasing with $t$. $\qquad\square$

**Lemma 6.** *Suppose starting from any initial point $w(0)$ and for any $t \ge 0$, the point $w(t)$ on the GF path satisfies*

$$F(w(t)) \le \frac{F(w(0))}{\lambda t}.$$

*Then, we have that for any $w(0)$ and $t \ge 1$,*

$$F(w(t)) \le F(w(0))e^{-\lfloor \lambda t/e \rfloor}.$$

.

**Proof.** Fix any $t \ge e$ and divide $[0, t]$ into $k = \lfloor \lambda t/e \rfloor$ many chunks of size $e/\lambda$ each. Let this partition be $[0, t_1, \ldots, t_k = t]$. Clearly, we have that for any $j \le k$, the point $w(t_j)$ corresponds to the point at time $e/\lambda$ on the GF path starting from $w(t_{j-1})$. The given rate assumption thus implies that

$$F(w(t_j)) \le \frac{F(w(t_{j-1}))}{e}.$$

Recursing the above for $j$ from 1 to $k$, we get that

$$F(w(t)) = F(w(t_k)) \le e^{-k}F(w(0)) = F(w(0))e^{-\lfloor \lambda t/e \rfloor}$$

.                                                                                                          $\square$

**Lemma 7** (Lemma 2.1, [51]). *For any H smooth function $F : \mathbb{R}^d \mapsto \mathbb{R}$, for any $x \in \mathbb{R}^d$,*

$$\|\nabla F(x)\| \le \sqrt{4H(F(x) - F^*)},$$

*where $F^* := \min_x F(x)$,*

# B Proofs from Section 3

**Proof of Theorem 1.** Let $w(s)$ be the point on the GF path after time $s$ when starting from the point $w(0)$. An application of chain rule implies that

$$\frac{\mathrm{d}\Phi(w(s))}{\mathrm{d}s} = \left\langle \nabla\Phi(w(s)), \frac{\mathrm{d}w(s)}{\mathrm{d}t} \right\rangle$$
$$= \langle \nabla\Phi(w(s)), -\nabla F(w(s)) \rangle$$
$$\leq -g(F(w(s))),$$

where the equality in the second line holds by the update rule of GF, i.e. $\frac{\mathrm{d}w(s)}{\mathrm{d}s} = -\nabla F(w(s))$ and the last line follows by using Definition 1 where $g$ is a monotonically increasing function that satisfies (4). Rearranging the terms and integrating both the sides for $s$ from $0$ to $t$, we get

$$\int_{s=0}^{t} g(F(w(s)))\,\mathrm{d}s \leq -\int_{s=0}^{t} \frac{\mathrm{d}\Phi(w(s))}{\mathrm{d}s}\,\mathrm{d}s = \Phi(w(0)) - \Phi(w(s)) \leq \Phi(w(0)), \qquad (14)$$

where the last inequality in the above holds because $\Phi(\cdot) \geq 0$ by definition.

We finally conclude by noting that $F(w(t))$ is a decreasing function of $t$ since

$$\frac{\mathrm{d}F(w(t))}{\mathrm{d}t} = \left\langle \nabla F(w(t)), \frac{\mathrm{d}t(w)}{\mathrm{d}t} \right\rangle = -\langle \nabla F(w(t)), \nabla F(w(t)) \rangle \leq 0,$$

where the second equality above follows from GF update rule. Since $g$ is a monotonically increasing function, the above implies that $g(F(w(t))) \leq g(F(w(s)))$ for all $s \leq t$. Using this relation in (14) implies that

$$g(F(w(t))) \cdot t \leq \int_{s=0}^{t} g(F(w(s)))\,\mathrm{d}s \leq \Phi(w(0)).$$

Rearranging the terms gives the desired relation. $\qquad \square$

**Proof of Theorem 2.** The following proof uses the most general conditions for admissibility of $R$ stated in Definition 2. Let $w \in \mathrm{clo}(W)$ be any initial point. Since $\int_{t=0}^{\infty} g(R(w,t))\,\mathrm{d}t < \infty$ and $\int_{t=0}^{\infty} g'(R(w,t))\|\nabla R(w,t)\|\,\mathrm{d}t < \infty$ for every $w \in \mathrm{clo}(\mathcal{W})$, the function $\Phi_g$ is well defined and is differentiable along the gradient flow path at the point $w$. Additionally, in the following $w(t)$ denotes the point at time $t$ on the GF path starting from $w$.

First, note that because $F(w(t)) \leq R(w,t)$, and $g$ is positive and monotonically increasing, we have

$$g(F(w)) = g(F(w(0))) \leq g(R(w,0))$$
$$= -\int_{t=0}^{\infty} \frac{\partial g(R(w,t))}{\partial t}\,\mathrm{d}t$$
$$= -\int_{t=0}^{\infty} g'(R(w,t))\frac{\partial R(w,t)}{\partial t}\,\mathrm{d}t$$
$$\leq \int_{t=0}^{\infty} g'(R(w,t))\langle \nabla R(w,t), \nabla F(w) \rangle\,\mathrm{d}t$$

where the first equality is a tautology since $\lim_{t\to\infty} g(R(w,t)) = 0$, and the second equality follows from Chain rule. The inequality in the last line uses the property in Definition 2-(b). Next, note that

$$\int_{t=0}^{\infty} g'(R(w,t))\langle \nabla R(w,t), \nabla F(w) \rangle\,\mathrm{d}t = \lim_{s\to 0^+} \int_{t=0}^{\infty} g'(R(w(s),t))\langle \nabla R(w(s),t), \nabla F(w(s)) \rangle\,\mathrm{d}t$$
$$= -\lim_{s\to 0^+} \int_{t=0}^{\infty} \frac{\partial g(R(w(s),t))}{\partial s}\,\mathrm{d}t$$
$$= -\lim_{s\to 0^+} \frac{\partial}{\partial s} \int_{t=0}^{\infty} g(R(w(s),t))\,\mathrm{d}t,$$

where the equality in the second line above holds due to Chain rule and the last line follows from interchanging the integral and the derivative, which is permissible since we have that $\int_{t=0}^{\infty} g(R(w(s),t))\,\mathrm{d}t < \infty$ for $w(s) \in \mathrm{clo}(W)$. Finally, note that

$$\lim_{s\to 0^+} \frac{\partial}{\partial s} \int_{t=0}^{\infty} g(R(w(s),t))\,\mathrm{d}t = \lim_{s\to 0^+} \frac{\partial}{\partial s} \Phi_g(w(s)) = -\langle \nabla\Phi_g(w(0)), \nabla F(w(0)) \rangle$$

where the first equality uses the definition of $\Phi_g$ and the second equality is due to Chain rule. Combining the above chain of inequalities and plugging in $w(0) = w$ implies the desired condition,

$$\langle \nabla \Phi_g(w), \nabla F(w) \rangle \geq g(F(w)).$$

$\square$

**Proof of Corollary 1.** Define $H = \max_{w \in \mathcal{W}} h(w)$, and the function $g$ as

$$g(z) = \frac{1}{\sigma(z/H) \log^2(\sigma(z/H))},$$

where the function $\sigma$ is defined as $\sigma(x) = e + r^{-1}(x)$. Using the above $g$ in Theorem 2, we get the potential

$$\Phi(w) = \int_{t=0}^{\infty} \frac{1}{\left(\sigma\left(\frac{h(w)}{H} r(t)\right)\right) \log^2\left(\sigma\left(\frac{h(w)}{H} r(t)\right)\right)} \, dt.$$

The potential satisfies

$$\Phi(w) \leq \int_{t=0}^{\infty} \frac{1}{(\sigma(r(t))) \log^2(\sigma(r(t)))} \, dt$$

$$= \int_{t=0}^{\infty} \frac{1}{(e+t) \log^2(e+t)} \, dt = 1, \tag{15}$$

where the first inequality holds because $h(w)/H \leq 1$ and since $\sigma$ is inverse of $r$, it has to be monotonically decreasing.

In addition to the above, we also note that

$$\int_{t=0}^{\infty} g'(R(w,t)) \|\nabla R(w,t)\| \, dt \leq \int_{t=0}^{\infty} \frac{3}{\sigma\left(\frac{h(w)}{H} r(t)\right)^2 \log^2\left(\sigma\left(\frac{h(w)}{H} r(t)\right)\right)} \sigma'\left(\frac{h(w)}{H} r(t)\right) \frac{\|\nabla h(w)\|}{H} r(t) \, dt$$

$$= \int_{t=0}^{\infty} \frac{3}{\sigma\left(\frac{h(w)}{H} r(t)\right)^2 \log^2\left(\sigma\left(\frac{h(w)}{H} r(t)\right)\right)} \frac{1}{r'\left(\sigma\left(\frac{h(w)}{H} r(t)\right)\right)} \frac{\|\nabla h(w)\|}{H} r(t) \, dt$$

$$\leq \int_{t=0}^{\infty} \frac{3}{\sigma\left(\frac{h(w)}{H} r(t)\right) \log^2\left(\sigma\left(\frac{h(w)}{H} r(t)\right)\right)} \frac{c}{r\left(\sigma\left(\frac{h(w)}{H} r(t)\right)\right)} \frac{\|\nabla h(w)\|}{H} r(t) \, dt$$

$$\leq \frac{3c\|\nabla h(w)\|}{h(w)} \int_{t=0}^{\infty} \frac{1}{\sigma\left(\frac{h(w)}{H} r(t)\right) \log^2\left(\sigma\left(\frac{h(w)}{h(w)} r(t)\right)\right)} \, dt$$

$$\leq \frac{3c\|\nabla h(w)\|}{H} < \infty,$$

where the first inequality is from Chain rule and a trivial algebraic upper bound. The second inequality uses the relation that $r(t) \leq c|r'(t)|t$ for any $t \geq 0$. The third inequality uses the fact that $r$ is monotonically decreasing and that $\sigma$ is the inverse of $r$, and the last line follows similar to the bound in (15). Thus, $g$ is a valid function and $\Phi$ defined above is an admissible potential. Using Theorem 1, we get that

$$g(F(w(t))) \leq \frac{\Phi(w(0))}{t} \leq \frac{1}{t}.$$

Rearranging the terms, we get

$$\sigma\left(\frac{F(w)}{H}\right) \geq \frac{t}{\log^2(t)}.$$

Using the fact that $\sigma(x) = r^{-1}(x)$ in the above, we get that

$$F(w) \leq Hr\left(t/\log^2(t)\right).$$

$\square$

**Proof of Proposition 1.** We prove the forward and reverse direction as follows:

(a) *Proof of $(a) \Rightarrow (b)$.* First note that $R(w, t) = F(w)2^{\lambda t}$ is an admissible rate function for $F$. Clearly, it is a decreasing function of $t$ and $\lim_{t \to \infty} R(w, t) = 0$ for any $w$. Furthermore, note that for $w(t)$ on the GF path of $w(0)$, we have

$$R(w(t), 0) = F(w(t)) \le F(w(0))e^{-\lambda t} = R(w(0), t),$$

where the inequality follows from the rate assumption. Thus, $R$ satisfies all the conditions in Definition 2. Thus, invoking Theorem 2 with $g(z) = z$, we get that

$$\Phi(w) = \int_{t=0}^{\infty} R(w, t) \, dt = \int_{t=0}^{\infty} F(w)e^{-\lambda t} \, dt = \frac{F(w)}{\lambda}$$

is an admissible potential for $F$. Thus, from (4), we get that

$$\frac{\|\nabla F(w)\|^2}{\lambda} = \langle \nabla \Phi(w), \nabla F(w) \rangle \ge F(w),$$

which implies the desired PŁ property.

(b) *Proof of $(b) \Rightarrow (a)$.* This follows by directly solving the corresponding differential equation along the GF path. Consider the potential function $\Phi(w) = \frac{F(w)}{\lambda}$. Note that $\Phi$ is positive, and due to the PŁ property, satisfies (4). Thus, $\Phi$ is an admissible potential w.r.t. $F$. Let $w(0)$ be the initial point for GF, we note that at the point $w(t)$ on its GF path,

$$\begin{aligned}
\frac{d\Phi(w(t))}{d(t)} &= \left\langle \nabla \Phi(w(t)), \frac{dw(t)}{dt} \right\rangle \\
&= -\langle \nabla \Phi(w(t)), \nabla F(w(t)) \rangle \\
&= -\frac{1}{\lambda} \|\nabla F(w(t))\|^2 \\
&\le -F(w(t)),
\end{aligned}$$

where the last line follows from the PŁ property. Plugging in the definition of $\Phi$ in the above, we get

$$\frac{dF(w(t))}{F(w(t))} \le -\lambda.$$

The above differential equation in $F$ has the following solution

$$F(w(t)) \le F(w(0))e^{-\lambda t}.$$

Since the above holds for any $w(0)$, $(a)$ immediately follows.

$\square$

**Proof of Proposition 2.** We prove the forward and reverse direction as follows:

1. *Proof of $(a) \Rightarrow (b)$* Since the rate is admissible, we must have that

$$\begin{aligned}
F(w) &\le \lim_{t \to 0} R(w, t) \\
&\le \lambda \lim_{t \to 0} \frac{\|w - w^*\|^2 - \|w(t) - w^*\|^2}{t} \\
&= \lambda \langle w - w^*, \nabla F(w) \rangle.
\end{aligned}$$

2. *Proof of $(b) \Rightarrow (a)$.* Clearly, $\Phi(w) = \lambda \|w - w^*\|/2$ is an admissible potential w.r.t. F since $\Phi(w) \ge 0$ and

$$\langle \nabla \Phi(w), \nabla F(w) \rangle = \lambda \langle \nabla F(w), w - w^* \rangle \ge F(w),$$

where the last inequality holds because $F$ is Linearizable. Thus, from Theorem 1 we get that for any initialization $w(0)$, the point $w(t)$ on its GF path satisfies

$$F(w(t)) \le \frac{\Phi(w(0)) - \Phi(w(t))}{t} = \lambda \frac{\|w(0) - w^*\|^2 - \|w(t) - w^*\|^2}{2t}.$$

$\square$

## C  Proofs from Section 4

**Proof of Theorem 3.** Fix any $T_0 > 0$ and set $d = (3T_0/2)^3$. Denote the variable $u = w[1:d-1]$ and $v = w[d]$, i.e. $w = (u, v)$ and consider the function

$$F(w) = \frac{1}{2}\|u\|_{3/2}^2 + g(v),$$

where

$$\|u\|_{3/2} = \left(\sum_{i=1}^{d-1} u[i]^{3/2}\right)^{2/3} \qquad \text{and} \qquad g(v) = \begin{cases} v^2 & \text{if} \quad |v| \le 1/2 \\ |v| - \frac{1}{4} & \text{if} \quad |v| \ge 1/2 \end{cases}.$$

Note that the $\min F(w)$ is attained at the point $w = 0$ and

$$\nabla F(w)[i] = \begin{cases} \sqrt{\|u\|_{3/2} \cdot u[i]} \cdot \text{sign}\{u[i]\} & \text{for } 1 \le i \le d-1 \\ \text{sign}\{v[i]\} & \text{for } i = 1 \text{ and } |v| \ge \frac{1}{2} \\ 2v[i] & \text{for } i = 1 \text{ and } |v| \le \frac{1}{2}. \end{cases}$$

We first argue that gradient flow converges at a rate of $O(1/t)$ for any initial point $w_0$. This follows from the fact that $f(w)$ is convex in $w$ and thus $\Phi(w) = \|w\|^2/2$ is a valid potential function that satisfies for any time $t$,

$$\frac{\mathrm{d}\Phi(w(t))}{\mathrm{d}t} = \langle w(t), -\nabla F(w(t))\rangle$$
$$\le -(F(w(t)) - F^*). \qquad \text{(since } F \text{ is convex)}$$

Integrating on both the sides for $t$ from $0$ to $T$ implies that:

$$\Phi(w(T)) - \Phi(w(0)) \le -\int_{t=0}^{\infty}(F(w(t)) - F^*)\,\mathrm{d}t \le -T(F(w(T)) - F^*),$$

where the inequality in the second line holds because the function value is non-increasing along any gradient flow path. Rearranging the terms and ignoring negative terms, implies the following rate of convergence for gradient flow:

$$F(w(T)) - F^* \le \frac{\Phi(w_0)}{T} \le \frac{\|w_0\|^2}{2T}.$$

Next, we argue that gradient descent algorithm given by the recursive process $w_{k+1} \leftarrow w_k - \eta\nabla F(w_k)$ fails to find a $1/10$ suboptimal solution when starting from the initial point $w_0 = \left(\frac{1}{d^{2/3}}, \ldots, \frac{1}{d^{2/3}}, 1\right)$. We consider two cases of step size $\eta$ below:

1. Case 1: $\eta \le \frac{3}{d^{1/3}}$. Note that any $w$ for which $F(w) \le 1/10$ must satisfy that $|V| \le 1$. However, recall that at initialization, $v = 1$. Furthermore, $\frac{\partial F(w)}{\partial v} = v$ whenever $v \in [1/2, 1]$ and thus gradient descent needs to take at least $\lfloor 2d^{1/3}/3 \rfloor$ many steps to ensure that $v \le 1/2$.

2. Case 2: $\eta > \frac{3}{d^{1/3}}$. We argue that gradient descent diverges to infinity in this case. In particular, after $k$ iterations of GD, the iterate $w_k = (u_k, v_k)$ satisfies

$$u_k[i] = \frac{(1 - \eta d^{1/3})^k}{d^{2/3}} \tag{16}$$

   We prove the above via induction. The base case for $k = 0$ follows by initialization. For the induction step, note that:

$$u_{k+1}[i] = u_k[i] - \eta\nabla F(u_k[i])$$
$$= \frac{(1 - \eta d^{1/3})^k}{d^{2/3}} - \eta\,\text{sign}\{u_k[i]\} \cdot \frac{|1 - \eta d^{1/3}|^k}{d^{1/3}}$$
$$= \frac{(1 - \eta d^{1/3})^{k+1}}{d^{2/3}}.$$

   Thus the above implies that after $T$ iterations, we have that $F(w) \ge (\eta d^{1/3} - 1)^T$, and thus GD fails to find a $1/10$ suboptimal solution for any $T \ge 1$.

Combining the two cases above, we get that in order to find a $1/10$ suboptimal solution, we need $T \geq \lfloor 2d^{1/3}/3 \rfloor \geq T_0$ implying the desired lower bound. Since $T_0$ is arbitrary, the above construction can be extended to hold for any $T > 0$ (by setting $d = \infty$). Thus, there exists a function for which GF succeeds at the rate of $1/T$ but GD fails to converge.

We finally conclude by noting that for the function $F(w)$ and the potential $\Phi(w) = \|w^2\|/2$, we have that for any point $w$ and $w'$,

$$\nabla^2 \Phi(w')[\nabla F(w), \nabla F(w)] = \|\nabla F(w)\|^2 \geq \|u\|_{3/2}\|u\|_1.$$

On the gradient descent trajectory (given in (16)), the point $u_k$ satisfies $\|u_k\|_1 = d^{1/3}\|u_k\|_{3/2}$ for any $k \geq 0$. Thus, we have that on the points of GD trajectory,

$$\nabla^2 \Phi(w'_k)[\nabla F(w_k), \nabla F(w_k)] = \|\nabla F(w_k)\|^2 \geq d^{1/3}\|u\|_{3/2}^2 = 2d^{1/3}(F(w_k) - g(v)).$$

Note that the above proof holds for any arbitrarily large $T_0$. $\qquad\qquad\square$

## C.1  Supporting technical results for proofs of Theorem 4 and 5

Before delving into the proof, we first establish the following structural lemma that relates the function $F$ and a corresponding potential $\Phi$.

**Lemma 8.** *let F(w) be any function that satisfies Assumption 1, and $\Phi$ be an admissible potential for $F$ (see Definition 1). Then, there exists a monotonically increasing function $\zeta : \mathbb{R}^+ \mapsto \mathbb{R}^+$ such that for any $w$,*

$$F(w) \leq \zeta(\Phi(w)).$$

**Proof of Lemma 8.** Assumption 1 implies that for any $w$,

$$\|\nabla F(w)\|^2 \leq \psi(F(w))$$

for some monotonically increasing function $\psi$. Note that without loss of generality, we can assume that $\psi(F(w)) > 1$ as one can substitute $\psi(F(w))$ by $\psi(F(w)) + 1$ while still satisfying the above condition. The above implies that

$$g(F(w)) \cdot \frac{\|\nabla F(w)\|^2}{\psi(F(w))} \leq g(F(w)).$$

Using the relation in Definition 1, we get that the potential $\Phi$ satisfies

$$\frac{g(F(w))}{\psi(F(w))}\|\nabla F(w)\|^2 \leq g(F(w)) \leq \langle \nabla \Phi(w), \nabla F(w) \rangle. \tag{17}$$

We first set up additional notation. Define a function $\sigma(z) : \mathbb{R}^+ \mapsto \mathbb{R}^+$ such that $\sigma(0) := 0$ and for any $z$, $\sigma'(z) = g(z)/\psi(z)$, and note that $\sigma$ is non-negative and monotonically increasing. We are now ready to delve into the proof. Consider any point $w$. Integrating along the gradient flow path starting from the point $w$, we get that

$$
\begin{aligned}
\sigma(F(w)) = \sigma(F(w(0))) &= \sigma(F(w(\infty))) - \int_{t=0}^{\infty} \frac{\mathrm{d}\sigma(F(w(t)))}{\mathrm{d}t}\,\mathrm{d}t \\
&\stackrel{(i)}{=} \sigma(F(w(\infty))) + \int_{t=0}^{\infty} \sigma'(F(w(t)))\|\nabla F(w(t))\|^2\,\mathrm{d}t \\
&\stackrel{(ii)}{=} \int_{t=0}^{\infty} \sigma'(F(w(t)))\|\nabla F(w(t))\|^2\,\mathrm{d}t \\
&\stackrel{(iii)}{=} \int_{t=0}^{\infty} \frac{g(F(w(t)))}{\psi(F(w(t)))}\|\nabla F(w(t))\|^2\,\mathrm{d}t,
\end{aligned}
\tag{18}
$$

where the equality in $(i)$ follows from Chain rule and because $\frac{\mathrm{d}F(w(t))}{\mathrm{d}t} = -\|\nabla F(w(t))\|^2$, $(ii)$ holds because of our assumption that $F(w(\infty)) = 0$ since gradient flow converges to the global minimizer and because $\sigma(0) = 0$. Finally, $(iii)$ follows from the definition of $\sigma'(z)$.

Similarly, integrating along the gradient flow path, we also have that

$$\Phi(w) = \Phi(w(0)) = \Phi(w(\infty)) - \int_{t=0}^{\infty} \frac{\mathrm{d}\Phi(w(t))}{\mathrm{d}t}\,\mathrm{d}t$$

$$\stackrel{(i)}{=} \Phi(w(\infty)) + \int_{t=0}^{\infty} \langle \nabla\Phi(w(t)), \nabla F(w(t)) \rangle \, \mathrm{d}t$$

$$\stackrel{(ii)}{=} \int_{t=0}^{\infty} \langle \nabla\Phi(w(t)), \nabla F(w(t)) \rangle \, \mathrm{d}t, \tag{19}$$

where in $(i)$ we used Chain rule and the fact that $\nabla w(t) = -\nabla F(w(t))$ and $(ii)$ holds because $\Phi(w(\infty)) = \Phi(w^*) = 0$ since $g(0) = 0$.

Finally, integrating (17) along the gradient flow path, we get the relation

$$\int_{t=0}^{\infty} \langle \nabla\Phi(w(t)), \nabla F(w(t)) \rangle \, \mathrm{d}t \geq \int_{t=0}^{\infty} \frac{g(F(w))}{\psi(F(w))} \|\nabla F(w)\|^2 \, \mathrm{d}t.$$

Plugging the relations (18) and (19) in the above, we get

$$\Phi(w) \geq \sigma(F(w)),$$

which implies that

$$F(w) \leq \zeta(\Phi(w)),$$

where the $\zeta(z) = \sigma^{-1}(z)$ can be uniquely defined, is positive and monotonically increasing. $\qquad\square$

We next establish the following utility lemma which is an alternative to second-order Taylor's expansion and will be useful in developing convergence bounds for GD and SGD.

**Lemma 9.** *Let $\Phi$ be any function that satisfies Assumption 2. Define the function $\theta : \mathbb{R}^+ \mapsto \mathbb{R}^+$ such that $\theta(0) = 0$ and $\theta'(z) = 1/\rho(z)$ for any $z \geq 0$. Then, for any $u \in \mathbb{R}^d$, we have*

$$\theta(\Phi(w+u)) \leq \theta(\Phi(w)) + \frac{1}{\rho(\Phi(w))} \langle \nabla\Phi(w), u \rangle + \frac{1}{2}\|u\|^2.$$

*Furthermore, at any point $w$,*

$$\|\nabla\Phi(w)\| \leq \rho(\Phi(w))\sqrt{2\theta(\Phi(w))}.$$

**Proof of Lemma 9.** Define the function

$$\ell(\alpha) := \theta(\Phi(w + \alpha u)), \tag{20}$$

and note that

$$\ell'(\alpha) = \frac{\mathrm{d}\ell(\alpha)}{\mathrm{d}\alpha} = \theta'(\Phi(w+\alpha u))\langle \nabla\Phi(w+\alpha u), u \rangle,$$

and

$$\ell''(\alpha) = \frac{\mathrm{d}^2\ell(\alpha)}{\mathrm{d}\alpha^2} = \theta''(\Phi(w+\alpha u))\langle u, \nabla\Phi(w+\alpha u) \rangle^2 + \theta'(\Phi(w+\alpha u))\langle \nabla^2\Phi(w+\alpha u)u, u \rangle$$

$$\stackrel{(i)}{\leq} \theta'(\Phi(w+\alpha u))\langle \nabla^2\Phi(w+\alpha u)u, u \rangle$$

$$\stackrel{(ii)}{\leq} \theta'(\Phi(w+\alpha u))\|\nabla^2\Phi(w+\alpha u)\|\|u\|^2$$

$$\stackrel{(iii)}{\leq} \theta'(\Phi(w+\alpha u))\rho(\Phi(w+\alpha u))\|u\|^2$$

$$\stackrel{(iii)}{\leq} \|u\|^2,$$

where $(i)$ holds because $\theta''(z) = \frac{-\rho'(z)}{\rho'(z)^2} \leq 0$ as $\rho'(z) \geq 0$ since $\rho$ is a monotonically increasing function, $(ii)$ follows from Hölder's inequality, $(iii)$ is due to Assumption 2 and finally $(iv)$ is from the definition of the function $\theta$.

Using Taylor expansion of $\ell(1)$ at the point $\alpha = 0$, we get that

$$\ell(1) \leq \ell(0) + \ell'(0) + \frac{1}{2}\ell''(\alpha'),$$

where $\alpha' \in [0, 1]$. Plugging in the values of $\ell(0)$, $\ell(1)$, $\ell'(0)$ and $\ell''(\alpha')$ from the above, we get

$$\theta(\Phi(w+u)) \leq \theta(\Phi(w)) + \theta'(\Phi(w))\langle \nabla\Phi(w), u \rangle + \frac{1}{2}\|u\|^2$$

$$= \theta(\Phi(w)) + \frac{1}{\rho(\Phi(w))}\langle \nabla\Phi(w), u \rangle + \frac{1}{2}\|u\|^2, \tag{21}$$

where the last line follows by using that fact that $\theta'(z) = 1/\rho(z)$. This proves the first relation.

We next prove the bound on $\|\nabla\Phi(w)\|$. Starting from (21), we have that for any $u \in \mathbb{R}^d$,

$$\theta(\Phi(w+u)) \leq \theta(\Phi(w)) + \frac{1}{\rho(\Phi(w))}\langle \nabla\Phi(w), u \rangle + \frac{1}{2}\|u\|^2.$$

Plugging in $u = -\frac{\nabla\Phi(w)}{\rho(\Phi(w))}$, we get

$$\theta\left(\Phi\left(w - \frac{\nabla\Phi(w)}{\rho(w)}\right)\right) \leq \theta(\Phi(w)) - \frac{1}{2\rho(\Phi(w))^2}\|\nabla\Phi(w)\|^2.$$

Rearranging the terms, we get

$$\|\nabla\Phi(w)\|^2 \leq 2\rho(\Phi(w))^2\left(\theta(\Phi(w)) - \theta\left(\Phi\left(w - \frac{\nabla\Phi(w)}{\rho(w)}\right)\right)\right)$$

$$\leq 2\rho(\Phi(w))^2\theta(\Phi(w)),$$

where the inequality in the second line holds because $\theta(z) > 0$. This proves the second relation. $\qquad\square$

## C.2    Proof of Theorem 4

We are now ready to prove the convergence guarantee for GD. We first state the full version of Theorem 4 that shows all the problem dependent constants hidden in the main body. While the following bound for GD looks complex at the first sight, this is the price we pay for the generality of our framework. Various invocations of this result are presented in Section 5.

**Theorem** (Theorem 4 restated with problem dependent constants)**.** *Let $\Phi_g$ be an admissible potential w.r.t. $F$. Assume that $F$ satisfies Assumption 1 with the bound given by the function $\psi$, and $\Phi_g$ satisfies Assumption 2 with the bound given by the function $\rho$. Then, for any initial point $w_0$,*

- *For any $T \geq 1$ and $\eta > 0$, the point $\widehat{w}_T$ returned by GD algorithm has the convergence guarantee*

$$g(F(\widehat{w}_T)) \leq \frac{2\theta(\Phi_g(w_0))\rho(\Phi_g(w_0))}{\eta T} + 2\eta\rho(\Phi_g(w_0))\psi(\zeta(\Phi_g(w_0))), \tag{22}$$

*Setting $\eta = \sqrt{\frac{\theta(\Phi_g(w_0))}{\psi(\zeta(\Phi_g(w_0)))} \cdot \frac{1}{T}}$ in the above implies the rate*

$$g(F(\widehat{w}_T)) \leq 4\rho(\Phi_g(w_0))\sqrt{\theta(\Phi_g(w_0))\psi(\zeta(\Phi_g(w_0)))} \cdot \frac{1}{\sqrt{T}}. \tag{23}$$

- *Furthermore, if the function $\frac{\psi}{g}$ is monotonically increasing, then for any $T \geq 1$ and $\eta \leq \frac{g(\zeta(\Phi_g(w_0)))}{\psi(\zeta(\Phi_g(w_0)))\cdot\rho(\Phi_g(w_0))}$, the point $\widehat{w}_T$ has the convergence guarantee*

$$g(F(\widehat{w}_T)) \leq \frac{2\theta(\Phi_g(w_0))\rho(\Phi_g(w_0))}{\eta T}. \tag{24}$$

*Setting $\eta = \frac{g(\zeta(\Phi_g(w_0)))}{\psi(\zeta(\Phi_g(w_0)))\cdot\rho(\Phi_g(w_0))}$ in the above implies the rate*

$$g(F(\widehat{w}_T)) \leq \frac{2\theta(\Phi_g(w_0))\psi(\zeta(\Phi_g(w_0)))\rho^2(\Phi_g(w_0))}{g(\zeta(\Phi_g(w_0)))} \cdot \frac{1}{T}. \tag{25}$$

- *Finally, if $\Phi_g = F$, then the above bounds hold with all occurrence of the term $\zeta(\Phi_g(w_0))$ replaced with $F(w_0)$.*

In the above, the function $\theta(z) := \int_{y=0}^{z} \frac{1}{\rho(y)} \, \mathrm{d}y$ and the function $\zeta$ is defined such that $\zeta^{-1}(z) = \int_{y=0}^{z} \frac{g(y)}{\psi(y)} \, \mathrm{d}y$.

**Proof of Theorem 4.** For the ease of notation, we remove the subscript $g$ from the potential $\Phi_g$ throughout the proof. Fix any $T > 0$ and let $\{w_t\}_{t=0}^{T}$ be the sequence of iterates generated by GD on $F(w)$ when starting from the point $w_0$ at $t = 0$. First note that for any $t \geq 0$, invoking Lemma 9 with $w = w_t$ and $u = -\eta \nabla F(w_t)$, and using Definition 1, we get

$$\theta(\Phi(w_{t+1})) \leq \theta(\Phi(w_t)) - \frac{\eta}{\rho(\Phi(w_t))} g(F(w_t)) + \frac{\eta^2}{2} \|\nabla F(w_t)\|^2$$

$$\leq \theta(\Phi(w_t)) - \frac{\eta}{\rho(\Phi(w_t))} g(F(w_t)) + \frac{\eta^2}{2} \psi(F(w_t)), \tag{26}$$

where $\theta$ is a monotonically increasing function and the second last line follows from Assumption 1. We now proceed with the proof of convergence for GD. Assume that for every $t \leq T$

$$g(F(w_t)) \geq \eta \rho(\Phi(w_0)) \psi(\zeta(\Phi(w_0))). \tag{27}$$

If the case above condition is violated, we immediately have that

$$\min_{t \leq T} g(F(w_t)) \leq \eta \rho(\Phi(w_0)) \psi(\zeta(\Phi(w_0))). \tag{28}$$

Thus, moving forward we assume that (27) holds. Fix any $t \leq T$. Starting from (33), we get

$$\theta(\Phi(w_{t+1})) \leq \theta(\Phi(w_t)) - \frac{\eta}{\rho(\Phi(w_t))} g(F(w_t)) + \frac{\eta^2}{2} \psi(F(w_t))$$

$$\leq \theta(\Phi(w_t)) - \frac{\eta}{\rho(\Phi(w_t))} g(F(w_t)) + \frac{\eta^2}{2} \psi(\zeta(\Phi(w_t))), \tag{29}$$

where the last inequality is due to Lemma 8 and because $\Psi$ is a monotonically increasing function. Before we delve into the proof of convergence of GD, we will first establish a useful property that $\Phi(w_t) \leq \Phi(w_0)$ for all $t \leq T$. We prove this via induction. For the base case ($t = 0$), starting from (29), we have

$$\theta(\Phi(w_1)) \leq \theta(\Phi(w_0)) - \frac{\eta}{\rho(\Phi(w_0))} g(F(w_0)) + \frac{\eta^2}{2} \psi(\zeta(\Phi(w_0)))$$

$$\leq \theta(\Phi(w_0)) - \frac{\eta}{2\rho(\Phi(w_0))} g(F(w_0))$$

$$\leq \theta(\Phi(w_0)),$$

where the inequality in the second line above holds due to (27). Since $\theta$ is a monotonically increasing function, the above implies that $\Phi(w_1) \leq \Phi(w_0)$. We next prove the induction step. Assume that $\Phi(w_\tau) \leq \Phi(w_0)$ for any $\tau \leq t$. Again, using (29), we have

$$\theta(\Phi(w_{t+1})) \leq \theta(\Phi(w_t)) - \frac{\eta}{\rho(\Phi(w_t))} g(F(w_t)) + \frac{\eta^2}{2} \psi(\zeta(\Phi(w_t)))$$

$$\overset{(i)}{\leq} \theta(\Phi(w_t)) - \frac{\eta}{\rho(\Phi(w_0))} g(F(w_t)) + \frac{\eta^2}{2} \psi(\zeta(\Phi(w_0)))$$

$$\overset{(ii)}{\leq} \theta(\Phi(w_t)) - \frac{\eta}{2\rho(\Phi(w_0))} g(F(w_t)) \tag{30}$$

$$\leq \theta(\Phi(w_t)),$$

where $(i)$ holds because $\Phi(w_t) \leq \Phi(w_0)$ via the induction hypothesis and because $\rho$, $\zeta$ and $\psi$ are monotonically increasing and non-negative functions and $F(w_t) \geq 0$. $(ii)$ is due to the relation in (27). Since $\theta$ is monotonic, this implies that $\Phi(w_{t+1}) \leq \Phi(w_t)$, completing the induction step and proving that $\Phi(w_t) \leq \Phi(w_0)$ for all $t \leq T$.

Since $\Phi(w_t) \leq \Phi(w)$ for all $t \leq T$, starting from (29) and replicating the steps till (35), we get that for any $t \leq T$,

$$\theta(\Phi(w_{t+1})) \leq \theta(\Phi(w_t)) - \frac{\eta}{2\rho(\Phi(w_0))} g(F(w_t)).$$

Telescoping the above for $t$ from $0$ to $T-1$ and rearranging the terms, we get that

$$\frac{\eta}{2T\rho(\Phi(w_0))} \sum_{t=1}^{T} g(F(w_t)) \leq \frac{\theta(\Phi(w_0)) - \theta(\Phi(w_{T+1}))}{T}.$$

Ignoring negative terms on the right hand side, we get

$$\frac{1}{T} \sum_{t=1}^{T} g(F(w_t)) \leq \frac{2\theta(\Phi(w_0))\rho(\Phi(w_0))}{\eta T},$$

and thus

$$\min_{t \leq T} g(F(w_t)) \leq \frac{2\theta(\Phi(w_0))\rho(\Phi(w_0))}{\eta T}. \tag{31}$$

The above analysis shows that at least one of the bound in (28) or (31) holds. Thus, taking both of them together, we get that

$$\min_{t \leq T} g(F(w_t)) \leq \frac{2\theta(\Phi(w_0))\rho(\Phi(w_0))}{\eta T} + \eta\rho(\Phi(w_0))\psi(\zeta(\Phi(w_0))).$$

**Improved bound when $\frac{\psi(z)}{g(z)}$ is a monotonically increasing function of $z$.** In this case, (33) implies that for any $t \geq 0$,

$$\theta(\Phi(w_{t+1})) \leq \theta(\Phi(w_t)) - \eta g(F(w_t)) \left( \frac{1}{\rho(\Phi(w_t))} - \frac{\eta}{2} \cdot \frac{\psi(F(w_t))}{g(F(w_t))} \right)$$

$$\leq \theta(\Phi(w_t)) - \eta g(F(w_t)) \left( \frac{1}{\rho(\Phi(w_t))} - \frac{\eta}{2} \cdot \frac{\psi(\zeta(\Phi(w_t)))}{g(\zeta(\Phi(w_t)))} \right), \tag{32}$$

where the last inequality follows from the fact that $\psi(z)/g(z)$ is an increasing function of $z$ and from Lemma 8. In the following, we will provide a convergence guarantee for GD whenever

$$\eta \leq \frac{g(\zeta(\Phi(w_0)))}{\psi(\zeta(\Phi(w_0))) \cdot \rho(\Phi(w_0))}. \tag{33}$$

We first show that for such an $\eta$, the iterates produced by GD satisfy $\Phi(w_t) \leq \Phi(w_0)$ for all $t \leq T$. The proof follows by induction. For the base case ($t = 0$), starting from relation (34), we have

$$\theta(\Phi(w_1)) \leq \theta(\Phi(w_0)) - \eta g(F(w_0)) \left( \frac{1}{\rho(\Phi(w_0))} - \frac{\eta}{2} \cdot \frac{\psi(\zeta(\Phi(w_0)))}{g(\zeta(\Phi(w_0)))} \right)$$

$$\leq \theta(\Phi(w_0)) - \frac{\eta}{2\rho(\Phi(w_0))} g(F(w_0))$$

$$\leq \theta(\Phi(w_0)), \tag{34}$$

where the inequality in the second line follows by plugging the bound on $\eta$ from (33), and the last inequality holds since $g(F(w_0)) \geq 0$. Since $\theta$ is a monotonically increasing function, the above implies that $\Phi(w_1) \leq \Phi(w_0)$ thus proving the base case.

We next prove the induction step. Assume that $\Phi(w_\tau) \leq \Phi(w_0)$ for any $\tau \leq t$. Again, starting from relation (34), we have

$$\theta(\Phi(w_{t+1})) \leq \theta(\Phi(w_t)) - \eta g(F(w_t)) \left( \frac{1}{\rho(\Phi(w_t))} - \frac{\eta}{2} \cdot \frac{\psi(F(w_t))}{g(F(w_t))} \right)$$

$$\leq \theta(\Phi(w_t)) - \eta g(F(w_t)) \left( \frac{1}{\rho(\Phi(w_0))} - \frac{\eta}{2} \cdot \frac{\psi(\zeta(\Phi(w_0)))}{g(\zeta(\Phi((w_0))))} \right)$$

$$\leq \theta(\Phi(w_t)) - \frac{\eta}{2\rho(\Phi(w_0))} g(F(w_t)) \tag{35}$$

$$\leq \theta(\Phi(w_t)),$$

where the second line holds because $F(w_t) \leq \zeta(\Phi(w_t)) \leq \zeta(\Phi(w_0))$ and $\psi(z)/g(z)$ is a monotonically increasing function of $z$, the third line holds by plugging the bound on $\eta$ from (33), and

the last inequality holds since $F(w_0) \geq 0$. Since $\theta$ is monotonically increasing, this implies that $\theta(w_{t+1}) \leq \theta(w_t)$, completing the induction step and proving that $\Phi(w_t) \leq \Phi(w_0)$ for all $t \leq T$.

We are now ready to complete the proof of convergence of GD. Since $\Phi(w_t) \leq \Phi(w)$ for all $t \leq T$, starting from (34) and replicating the steps till (35), we get that for any $t \leq T$,

$$\theta(\Phi(w_{t+1})) \leq \theta(\Phi(w_t)) - \frac{\eta}{2\rho(\Phi(w_0))} g(F(w_t)). \tag{36}$$

Telescoping the above for $t$ from $0$ to $T$ and rearranging the terms, we get that

$$\frac{\eta}{2T\rho(\Phi(w_0))} \sum_{t=1}^{T} g(F(w_t)) \leq \frac{\theta(\Phi(w_0)) - \theta(\Phi(w_{T+1}))}{T}.$$

Ignoring negative items on the right hand side, we get

$$\frac{1}{T} \sum_{t=1}^{T} g(F(w_t)) \leq \frac{2\theta(\Phi(w_0))\rho(\Phi(w_0))}{\eta T},$$

and thus

$$\min_{t \leq T} g(F(w_t)) \leq \frac{2\theta(\Phi(w_0))\rho(\Phi(w_0))}{\eta T}.$$

**Improved analysis when $\Phi_g = F$.** The proof follows identically, with the only major change being that Lemma 8 now holds with the function $\zeta(z) = z$ since $F(w) = \Phi_g(w)$. $\qquad\square$

### C.3   Proof of Theorem 5

We first note the following high probability and in-expectation bounds on the norm of the stochastic gradient estimate.

**Lemma 10.** *Let $\{w_t\}_{t \leq T}$ be the sequence of iterates generated by SGD algorithm on $F$ using stochastic estimates based on $\{z_t\}_{t \leq T}$. Then, with probability at least $1 - \delta$, for any time $t \leq T$,*

$$\|\nabla f(w; z)\|^2 \leq \Lambda(F(w)) \log(T/\delta)$$

*and for any $w > 0$,*

$$\mathbb{E}\big[\|\nabla f(w; z)\|^2\big] \leq \Lambda(F(w)),$$

*where the function $\Lambda(z) := 2\psi(z) + 2\chi(z)$, and the functions $\psi$ and $\chi$ given in Assumption 1 and 3 respectively.*

**Proof of Lemma 10.** Note that for any $0 \leq t \leq T$, with probability at least $1 - \frac{\delta}{T}$,

$$\|\nabla f(w; z) - \nabla F(w)\|^2 \leq \chi(F(w)) \cdot \log\left(\frac{T}{\delta}\right),$$

which implies that

$$\begin{aligned}
\|\nabla f(w; z)\|^2 &\leq 2\|\nabla F(w)\|^2 + 2\|\nabla f(w; z) - \nabla F(w)\|^2 \\
&\leq 2\psi(F(w)) + 2\|\nabla f(w; z) - \nabla F(w)\|^2 \\
&\leq (2\psi(F(w)) + 2\chi(F(w))) \cdot \log\left(\frac{T}{\delta}\right) \\
&= \Lambda(F(w)) \log\left(\frac{T}{\delta}\right),
\end{aligned}$$

where the inequality in the second to last line follows from Assumption 3 and the last line is from the definition of $\Lambda$. The desired bounds follows with probability at least $1 - \delta$ by taking a union bound w.r.t. $t$.

For the in-expectation bound, since for any random variable $X$, $\mathbb{E}[X] = \int_{t=0}^{\infty} \Pr(X \geq t)\, \mathrm{d}t$, we have

$$\frac{\mathbb{E}\big[\|\nabla f(w; z) - \nabla F(w)\|^2\big]}{\chi(F(w))} = \int_{t=0}^{\infty} \Pr\left(\frac{\mathbb{E}\big[\|\nabla f(w; z) - \nabla F(w)\|^2\big]}{\chi(F(w))} \geq t\right) \mathrm{d}t$$

$$\leq \int_{t=0}^{\infty} e^{-t}\, \mathrm{d}t = 1. \tag{37}$$

Thus,

$$\mathbb{E}\big[\|\nabla f(w; z)\|^2\big] \leq 2\|\nabla F(w)\|^2 + 2\,\mathbb{E}\big[\|\nabla f(w; z) - \nabla F(w)\|^2\big]$$
$$\leq 2\|\nabla F(w)\|^2 + 2\chi(F(w))$$
$$\leq 2\psi(F(w)) + 2\chi(F(w)) =: \Lambda(F(w)),$$

where the inequality in the second line above follows (37) and the last line is due to Assumption 3. $\quad\square$

We are now ready to prove the convergence guarantee for SGD. We first state the full version of Theorem 5 that shows all the problem dependent constants hidden in the main body, but keeps $\kappa$ as a free variable. Then, we provide an easier to understand result in Remark 4 by setting $\kappa$ appropriately. Various invocations of this result are presented in Section 5.

**Theorem** (Theorem 5 restated with problem dependent constants)**.** *Let $\Phi_g$ be an admissible potential w.r.t. $F$. Assume that $F$ satisfies Assumption 1 with the bound given by the function $\psi$, $\Phi_g$ satisfies Assumption 2 with the bound given by the function $\rho$, and the stochastic gradient estimates $\nabla f(w; z)$ satisfy Assumption 3 with the bound given by the function $\chi$. Then, for any $T \geq 1$, $\kappa > 1$, initial point $w_0$, setting*

$$\eta \leq \frac{M - \theta(\Phi_g(w_0))}{20 \log^2(20T)\sqrt{MBT}},$$

*we get that with probability at least $0.7$, the point $\widehat{w}_T$ returned by SGD algorithm satisfies*

$$g(F(\widehat{w}_T)) \leq \kappa\rho(\Phi(w_0))\Big(\frac{100M}{\eta T} + 50\eta B \log^2(20T)\Big).$$

*where the function $\theta(z) := \int_{y=0}^{z} \frac{1}{\rho(y)}\, \mathrm{d}y$, the function $\zeta$ is defined such that $\zeta^{-1}(z) = \int_{y=0}^{z} \frac{g(y)}{\psi(y)}\, \mathrm{d}y$ and the function $\Lambda(z) := 2\psi(z) + 2\chi(z)$. Furthermore, the constant $B = \Lambda(\zeta(\rho^{-1}(\kappa\rho(\Phi_g(w_0)))))$ and $M = \theta(\rho^{-1}(\kappa\rho(\Phi_g(w_0))))$.*

**Remark 4.** *Fix any initial point $w_0$ and let $\bar{w}$ be any point such that $\Phi_g(\bar{w}) > \Phi_g(w_0)$. Then, setting $\kappa = \frac{\rho(\Phi_g(\bar{w}))}{\rho(\Phi_g(w_0))}$ in Theorem 5 (above) implies that $B = \Lambda(\zeta(\Phi_g(\bar{w})))$ and $M = \theta(\Phi_g(\bar{w}))$. Thus, for any $T \geq 1$, setting*

$$\eta \leq \frac{\theta(\Phi_g(\bar{w})) - \theta(\Phi_g(w_0))}{20 \log^2(20T)\sqrt{\Lambda(\zeta(\Phi_g(\bar{w})))\theta(\Phi_g(\bar{w})) \cdot T}},$$

*we get that with probability at least $0.7$, the point $\widehat{w}_T$ returned by SGD algorithm satisfies*

$$g(F(\widehat{w}_T)) \leq \widetilde{O}\Big(\rho(\Phi_g(\bar{w})) \cdot \frac{\theta(\Phi_g(\bar{w}))}{\theta(\Phi_g(\bar{w})) - \theta(\Phi_g(w_0))} \cdot \sqrt{\Lambda(\zeta(\Phi_g(\bar{w}))\theta(\Phi_g(\bar{w})))} \cdot \frac{1}{\sqrt{T}}\Big).$$

**Proof of Theorem 5.** Let $\{w_t\}_{t\leq T}$ be the sequence of iterates generated by SGD algorithm in the first $T$ times steps using the random samples $\{z_t\}_{t\leq T}$ sampled i.i.d. from an unknown distribution. Let $\mathcal{F}_t$ be the natural filtration at time $t$ such that $\{w_j, z_j\}_{j\leq t}$ are $\mathcal{F}_t$-measurable, and let $\bar{\eta} = \frac{M - \theta(\Phi(w_0))}{20 \log^2(20T)\sqrt{MBT}}$.

**Part 1: Setup.** For any $0 \leq t \leq T$, an application of Lemma 9 with $w = w_t$ and $u = -\eta\nabla f(w_t; z_t)$ implies that

$$\theta(\Phi(w_{t+1})) = \theta(\Phi(w_t - \eta\nabla f(w_t; z_t)))$$
$$\leq \theta(\Phi(w_t)) - \frac{\eta}{\rho(\Phi(w_t))}\langle\nabla f(w_t; z_t), \nabla\Phi(w_t)\rangle + \frac{\eta^2}{2}\|\nabla f(w_t; z_t)\|^2. \tag{38}$$

Taking expectation on both the sides with respect to $z_t$, we get

$$\mathbb{E}_t[\theta(\Phi(w_{t+1}))] \leq \theta(\Phi(w_t)) - \frac{\eta}{\rho(\Phi(w_t))}\,\mathbb{E}_t[\langle\nabla f(w_t; z_t), \nabla\Phi(w_t)\rangle] + \frac{\eta^2}{2}\,\mathbb{E}_t\big[\|\nabla f(w_t; z_t)\|^2\big]$$
$$\leq \theta(\Phi(w_t)) - \frac{\eta}{\rho(\Phi(w_t))}g(F(w_t)) + \frac{\eta^2}{2}\,\mathbb{E}_t\big[\|\nabla f(w_t; z_t)\|^2\big]$$

$$\leq \theta(\Phi(w_t)) - \frac{\eta}{\rho(\Phi(w_t))} g(F(w_t)) + \frac{\eta^2}{2} \Lambda(F(w_t)), \tag{39}$$

where the inequality in the second line holds because $\mathbb{E}[\langle \nabla f(w_t; z_t), \nabla \Phi(w_t) \rangle] = \langle \nabla F(w_t), \nabla \Phi(w_t) \rangle \geq g(F(w_t))$ since $w_t$ is independent of $z_t$, and the last line follows from Lemma 10. Rearranging the terms and summing for $t$ from 0 to $T-1$, we get that

$$\sum_{t=0}^{T-1} \frac{\eta}{\rho(\Phi(w_t))} g(F(w_t)) \leq \sum_{t=0}^{T-1} \left( \theta(\Phi(w_t)) - \mathbb{E}_t[\theta(\Phi(w_{t+1}))] + \frac{\eta^2}{2} \Lambda(F(w_t)) \right). \tag{40}$$

Our focus in Part-2 below will be to control the term on the left hand size above.

**Part 2: Lower bound on $\rho(\Phi(w_t))$.** We first set up additional notation and derive some supporting results. Consider the stochastic process $\{Y_t\}_{t \leq T}$ defined as

$$Y_t = \begin{cases} \theta(\Phi(w_t)) + \sum_{j=0}^{t-1} \left( \frac{\eta}{\rho(\Phi(w_j))} g(F(w_j)) - \frac{\eta^2 \Lambda(F(w_j))}{2} \right) & \text{if } t \leq \tau \\ Y_\tau & \text{if } t > \tau \end{cases}, \tag{41}$$

where $\tau$ is defined as the first time smaller than or equal to $T$ at which $\rho(\Phi(w_t)) > \kappa \rho(\Phi(w_0))$ i.e.,

$$\tau := \inf\{t \leq T \mid \rho(\Phi(w_t)) > \kappa \rho(\Phi(w_0))\}, \tag{42}$$

where $\kappa > 1$ and will be set later. If there is no such $\tau$ for which (42) holds, we set $\tau = T$. Essentially, $\{Y_t\}_{t \leq T}$ is a stochastic process where $Y_t$ depends on the random variable $w_t$, and is stopped as soon as $\rho(\Phi(w_t)) > \kappa \rho(\Phi(w_0))$. To keep the current proof concise, we show in Lemma 11 (below) that the process $\{Y_t\}_{t \geq 0}$ is a super-martingale with respect to the filtration $\mathcal{F}_t$, and that with probability at least 0.95, for all $t \leq T$,

$$Y_t - Y_0 \leq \sqrt{\frac{1}{2} \sum_{j=0}^{t-1} \left( 5\eta \sqrt{M} \cdot \|\nabla f(w_j; z_j)\| + 4\eta^2 \|\nabla f(w_j; z_j)\|^2 \right)^2 \log(20T)}. \tag{$\mathscr{E}_1$}$$

where $M = \theta(\rho^{-1}(\kappa \rho(\Phi(w_0))))$. We additionally also note that from Lemma 10, with probability at least 0.95, for all $t \leq T$,

$$\|\nabla f(w_t; z_t)\|^2 \leq \Lambda(F(w_t)) \log(20T) \tag{$\mathscr{E}_2$}$$

Taking a union bound over the events $\mathscr{E}_1$ and $\mathscr{E}_2$ above, we get that for any $t \leq T$,

$$Y_t - Y_0 \leq \sqrt{\frac{1}{2} \sum_{j=0}^{t-1} \left( 5\eta \sqrt{M \cdot \Lambda(F(w_j))} + 4\eta^2 \Lambda(F(w_j)) \right)^2 \log^3(20T)}. \tag{$\mathscr{E}_3$}$$

In the following, we show that under the event $\mathscr{E}_3$, the condition in (42) never occurs. Suppose the contrary is true and that (42) occurs for some $\tau \leq T$. Then, we have that

$$Y_\tau - Y_0 \leq \sqrt{\frac{1}{2} \sum_{j=0}^{\tau-1} \left( 5\eta \sqrt{M \cdot \Lambda(F(w_j))} + 4\eta^2 \Lambda(F(w_j)) \right) \log^3(20T)}$$

$$\leq \sqrt{\frac{\tau}{2} \left( 5\eta \sqrt{MB} + 4\eta^2 B \right)^2 \log^3(20T)}$$

$$\leq 9\eta \sqrt{MB} \log^2(20T) \cdot \sqrt{\tau} \tag{43}$$

where the last line holds because $\eta \leq \bar{\eta} \leq \sqrt{M/B}$ and in the second to last line, we used the fact that

$$\Lambda(F(w_j)) \overset{(i)}{\leq} \Lambda(\zeta(\Phi(w_j))) \overset{(ii)}{\leq} \Lambda(\zeta(\rho^{-1}(\kappa \rho(\Phi(w_0))))) =: B, \tag{44}$$

where $(i)$ holds due to Lemma 8 and $(ii)$ follows from the fact that $\rho(\Phi(w_j)) \leq \kappa \rho(\Phi(w_0))$ for all $j < \tau$. However, from the definition of $Y_t$, we also have that

$$Y_\tau - Y_0 = \theta(\Phi(w_\tau)) - \theta(\Phi(w_0)) + \sum_{j=0}^{\tau-1} \left( \frac{\eta}{\rho(\Phi(w_j))} g(F(w_j)) - \frac{\eta^2 \Lambda(F(w_j))}{2} \right)$$

$$\geq \theta(\Phi(w_\tau)) - \theta(\Phi(w_0)) - \sum_{j=0}^{\tau-1} \frac{\eta^2 \Lambda(F(w_j))}{2}$$

$$\overset{(i)}{\geq} M - \theta(\Phi(w_0)) - \sum_{j=0}^{\tau-1} \frac{\eta^2 \Lambda(F(w_j))}{2}$$

$$\overset{(ii)}{\geq} M - \theta(\Phi(w_0)) - \frac{\eta^2 \tau B}{2}$$

$$\geq \frac{M - \theta(\Phi(w_0))}{2} \tag{45}$$

where in $(i)$, we used the fact that $\Phi(w_\tau) > \rho^{-1}(\kappa\rho(\Phi(w_0))) = M$, $(ii)$ follows by noting the bound in (44) for any $j < \tau$. The last line follows from the fact that $\eta \leq \bar{\eta} \leq \sqrt{(M - \theta(\Phi(w_0))/BT}$. However, note that this leads to a contradiction as both (43) and (45) can not be simultaneously true when when $\eta \leq \bar{\eta} = \frac{M - \theta(\Phi(w_0))}{20 \log^2(20T)\sqrt{MBT}}$. Thus, we must have that with probability at least 0.9, for any $t \leq T$,

$$\rho(\Phi(w_t)) \leq \kappa\rho(\Phi(w_0)) \tag{46}$$

In the following, we condition on the fact that (46) holds.

**Part 3: Convergence guarantee.** The following proof conditions on the events $\mathscr{E}_1$, $\mathscr{E}_2$ , $\mathscr{E}_3$. First note that, telescoping (38) from $t = 0$ to $T - 1$ and ignoring negative terms in the right hand side, we get that

$$\eta \sum_{t=0}^{T-1} \frac{\langle \nabla f(w_t; z_t), \nabla\Phi(w_t) \rangle}{\rho(\Phi(w_t))} \leq \theta(\Phi(w_0)) + \frac{\eta^2}{2} \sum_{t=0}^{T-1} \|\nabla f(w_t; z_t)\|^2. \tag{47}$$

The left hand side above can be controlled using Azuma-Hoeffding's inequality (Lemma 4), which implies that with probability at least 0.95,

$$\sum_{t=0}^{T-1} \frac{\langle \nabla f(w_t; z_t), \nabla\Phi(w_t) \rangle}{\rho(\Phi(w_t))} \geq \sum_{t=0}^{T-1} \mathbb{E}\left[ \frac{\langle \nabla f(w_t; z_t), \nabla\Phi(w_t) \rangle}{\rho(\Phi(w_t))} \right] - 2 \max_{t<T} \frac{\langle \nabla f(w_t; z_t), \nabla\Phi(w_t) \rangle}{\rho(\Phi(w_t))} \sqrt{T \log(20)}$$

$$\overset{(i)}{=} \sum_{t=0}^{T-1} \mathbb{E}\left[ \frac{\langle \nabla F(w_t), \nabla\Phi(w_t) \rangle}{\rho(\Phi(w_t))} \right] - 2 \max_{t<T} \frac{\|\nabla f(w_t; z_t)\| \|\nabla\Phi(w_t)\|}{\rho(\Phi(w_t))} \sqrt{T \log(20)}$$

$$\overset{(ii)}{\geq} \sum_{t=0}^{T-1} \mathbb{E}\left[ \frac{g(F(w_t))}{\rho(\Phi(w_t))} \right] - 2 \max_{t<T} \sqrt{2\theta(\Phi(w_t))} \|\nabla f(w_t; z_t)\| \sqrt{T \log(20)}$$

where $(i)$ above holds due to linearity of expectation w.r.t. $z_t$ and the inner product, and using Cauchy-Schwarz inequality. The inequality in $(ii)$ holds because of the relation (4) and Lemma 9. Plugging the above bound in (47) and rearranging the terms, we get

$$\eta \sum_{t=0}^{T-1} \mathbb{E}\left[ \frac{g(F(w_t))}{\rho(\Phi(w_t))} \right] \leq \theta(\Phi(w_0)) + \frac{\eta^2}{2} \sum_{t=0}^{T-1} \|\nabla f(w_t; z_t)\|^2 + 2\eta \max_{t<T} \sqrt{2\theta(\Phi(w_t))} \|\nabla f(w_t; z_t)\| \sqrt{T \log(20)}.$$

An application of Markov's inequality in the above implies that with probability at least 0.9,

$$\eta \sum_{t=0}^{T-1} \frac{g(F(w_t))}{\rho(\Phi(w_t))} \leq 10\eta \sum_{t=0}^{T-1} \mathbb{E}\left[ \frac{g(F(w_t))}{\rho(\Phi(w_t))} \right]$$

$$\leq 10\theta(\Phi(w_0)) + 5\eta^2 \sum_{t=0}^{T-1} \|\nabla f(w_t; z_t)\|^2 + 20\eta \max_{t<T} \sqrt{2\theta(\Phi(w_t))} \|\nabla f(w_t; z_t)\| \sqrt{T \log(20)}. \tag{$\mathscr{E}_4$}$$

Conditioning on the event $\mathscr{E}_2$ and plugging in the corresponding bound on $\|\nabla f(w_t; z_t)\|^2$, and dividing both the sides by $\eta$, we get that

$$\sum_{t=0}^{T-1} \frac{g(F(w_t))}{\rho(\Phi(w_t))} \leq \frac{10\theta(\Phi(w_0))}{\eta} + 5\eta \sum_{t=0}^{T-1} \Lambda(F(w_t)) \log(20T) + 20 \max_{t<T} \sqrt{2\theta(\Phi(w_t))\Lambda(F(w_t))T \log^2(20T)}$$

$$\leq \frac{10\theta(\Phi(w_0))}{\eta} + 5\eta \sum_{t=0}^{T-1} \Lambda(\zeta(\Phi(w_t))) \log(20T)$$

$$+ 20 \max_{t<T} \sqrt{2\theta(\Phi(w_t))\Lambda(\zeta(\Phi((w_t))))T\log^2(20T)}$$

$$\leq \frac{10M}{\eta} + 5\eta BT \log(20T) + 20\sqrt{2MBT\log^2(20T)},$$

where the second line above holds because of Lemma 8 and because $\Lambda$ is monotonically increasing. The inequality in the last line follows from plugging in the bound (46) which implies that $\Lambda(\zeta(\Phi(w_t))) \leq \Lambda(\zeta(\rho^{-1}(\kappa\rho(\Phi(w_0))))) = B$, and $\theta(\Phi(w_t)) \leq \theta(\rho^{-1}(\kappa\rho(\Phi(w_0)))) = M$ since both $\Lambda$ and $\zeta$ are monotonically increasing functions. Using (46) in the LHS above, rearranging the terms and dividing both the sides by $T$, we get that

$$\frac{1}{T}\sum_{t=0}^{T-1} g(F(w_t)) \leq \kappa\rho(\Phi(w_0))\left(\frac{10M}{\eta T} + 5\eta B\log(20T) + 20\sqrt{\frac{2MB\log^2(20T)}{T}}\right)$$

$$\leq \kappa\rho(\Phi(w_0))\left(\frac{100M}{\eta T} + 50\eta B\log^2(20T)\right),$$

where the last line is by applying AM-GM inequality on the last term.

Accounting for the union bounds for events $\mathscr{E}_1$, $\mathscr{E}_2$, $\mathscr{E}_3$ and $\mathscr{E}_4$, we get that the above bound on the rate of convergence of GD holds with probability at least $0.7$. $\qquad\square$

The following technical result is used in the proof of Theorem 5.

**Lemma 11.** *Suppose the premise of Theorem 5 holds, and let $\{w_t\}_{t\leq T}$ be the sequence of iterates generated by SGD algorithm on $F$ using stochastic estimates based on $\{z_t\}_{t\leq T}$. Let the process $\{Y_t\}_{t\geq 0}$ be defined as*

$$Y_t = \begin{cases} \theta(\Phi(w_t)) + \sum_{j=0}^{t-1}\left(\frac{\eta}{\rho(\Phi(w_j))}g(F(w_j)) - \frac{\eta^2\Lambda(F(w_j))}{2}\right) & \text{if } t \leq \tau \\ Y_\tau & \text{if } t > \tau \end{cases}, \qquad (48)$$

*where $\tau = \min\{T, \inf\{t \mid \rho(\Phi(w_t)) > \kappa\rho(\Phi(w_0))\}\}$ and $\Lambda(z) = 2\psi(F(w)) + 2\chi(F(w))$ where the function $\psi$ and $\chi$ given in Assumption 1 and 3 respectively. Then, $\{Y_t\}_{t\geq 0}$ is a super-martingale. Furthermore, with probability at-least $0.95$, for all $t \leq T$,*

$$Y_t - Y_0 \leq \sqrt{\frac{1}{2}\sum_{j=0}^{t-1}\left(5\eta\sqrt{M}\cdot\|\nabla f(w_j; z_j)\| + 4\eta^2\|\nabla f(w_j; z_j)\|^2\right)\log(20T)},$$

*where $M = \theta(\rho^{-1}(\kappa\rho(\Phi(w_0))))$.*

**Proof of Lemma 11.** Let $\mathcal{F}_t$ be the natural filtration at time $t$ such that $\{w_j, z_j\}_{j\leq t}$ are $\mathcal{F}_t$-measurable. For any $t \geq 0$, repeating the steps till (39) in the proof of Theorem 5 above we get that

$$\mathbb{E}_t[\theta(\Phi(w_{t+1}))] \leq \theta(\Phi(w_t)) - \frac{\eta}{\rho(\Phi(w_t))}g(F(w_t)) + \frac{\eta^2\Lambda(F(w_t))}{2}, \qquad (49)$$

where $\mathbb{E}_t$ denotes expectation w.r.t. the random variable $z_t$, and conditioning on $\mathcal{F}_{t-1}$. We first show that the process $\{Y_t\}_{t\geq 0}$ is a super-martingale. Note that for any time $t \leq \tau$,

$$\mathbb{E}_t[Y_{t+1}] = \mathbb{E}_t[\theta(\Phi(w_{t+1}))] + \sum_{j=0}^{t}\left(\frac{\eta}{\rho(\Phi(w_j))}g(F(w_j)) - \frac{\eta^2\Lambda(F(w_j))}{2}\right)$$

$$\leq \theta(\Phi(w_t)) + \sum_{j=0}^{t-1}\left(\frac{\eta}{\rho(\Phi(w_j))}g(F(w_j)) - \frac{\eta^2\Lambda(F(w_j))}{2}\right) = Y_t,$$

where the inequality in the second line above follows from (49). When $t > \tau$, by definition we have that $\mathbb{E}_t[Y_{t+1}] = Y_t$. Hence, the process $\{Y_t\}_{t\geq 0}$ is a super-martingale.

**Bound on the difference sequence.** There are two cases, either (a) $t > \tau$, or (b) $t \le \tau$. In the first case, $|Y_{t+1} - Y_t| = 0$. In the following, we provide a bound on the difference sequence for $t \le \tau$. First note that

$$Y_{t+1} - Y_t = \theta(\Phi(w_{t+1})) - \theta(\Phi(w_t)) + \underbrace{\frac{\eta}{\rho(\Phi(w_t))} g(F(w_t)) - \frac{\eta^2 \Lambda(F(w_t))}{2}}_{:=C_t} . \qquad (50)$$

Note that the term $C_t$ is $\mathcal{F}_t$-predictable. Thus, we just need to find $\mathcal{F}_t$-measurable processes $A'_t$ and $B'_t$ such that

$$A'_t \le \theta(\Phi(w_{t+1})) - \theta(\Phi(w_t)) \le B'_t.$$

Recall that an application of Lemma 9 with $w = w_t$ and $u = -\eta \nabla f(w_t; z_t)$ implies that

$$
\begin{aligned}
\theta(\Phi(w_{t+1})) - \theta(\Phi(w_t)) &\le -\frac{\eta}{\rho(\Phi(w_t))} \langle \nabla f(w_t; z_t), \nabla \Phi(w_t) \rangle + \frac{\eta^2}{2} \|\nabla f(w_t; z_t)\|^2 \\
&\overset{(i)}{\le} \frac{\eta}{\rho(\Phi(w_t))} \|\nabla f(w_t; z_t)\| \|\nabla \Phi(w_t)\| + \frac{\eta^2}{2} \|\nabla f(w_t; z_t)\|^2 \\
&\overset{(ii)}{\le} \underbrace{\eta \sqrt{2\theta(\Phi(w_t))} \|\nabla f(w_t; z_t)\| + \frac{\eta^2}{2} \|\nabla f(w_t; z_t)\|^2}_{=:B'_t}, \qquad (51)
\end{aligned}
$$

where $(i)$ above follows from Cauchy-Schwarz inequality, and $(ii)$ holds due to Lemma 9. Note that $B'_t$ defined to be the terms on the RHS above is $\mathcal{F}_t$-measurable.

We next consider the lower bound on $\theta(\Phi(w_{t+1})) - \theta(\Phi(w_t))$. Plugging in $w = w_{t+1}$ and $u = \eta \nabla f(w_t; z_t)$ in Lemma 9, we get that

$$
\begin{aligned}
\theta(\Phi(w_t)) &= \theta(\Phi(w_{t+1} + \eta \nabla f(w_t; z_t))) \\
&\le \theta(\Phi(w_{t+1})) + \frac{\eta}{\rho(\phi(w_{t+1}))} \langle \nabla \Phi(w_{t+1}), \nabla f(w_t; z_t) \rangle + \frac{\eta^2}{2} \|\nabla f(w_t; z_t)\|^2,
\end{aligned}
$$

rearranging the terms gives us

$$
\begin{aligned}
\theta(\Phi(w_{t+1})) - \theta(\Phi(w_t)) &\ge -\frac{\eta}{\rho(\phi(w_{t+1}))} \langle \nabla \Phi(w_{t+1}), \nabla f(w_t; z_t) \rangle - \frac{\eta^2}{2} \|\nabla f(w_t; z_t)\|^2 \\
&\overset{(i)}{\ge} -\frac{\eta}{\rho(\phi(w_{t+1}))} \|\nabla \Phi(w_{t+1})\| \|\nabla f(w_t; z_t)\| - \frac{\eta^2}{2} \|\nabla f(w_t; z_t)\|^2 \\
&\overset{(ii)}{\ge} -\eta \sqrt{2\theta(\Phi(w_{t+1}))} \cdot \|\nabla f(w_t; z_t)\| - \frac{\eta^2}{2} \|\nabla f(w_t; z_t)\|^2 \\
&= -\eta \sqrt{2(\theta(\Phi(w_{t+1})) - \theta(\Phi(w_t)) + \theta(\Phi(w_t)))} \cdot \|\nabla f(w_t; z_t)\| - \frac{\eta^2}{2} \|\nabla f(w_t; z_t)\|^2 \\
&\overset{(iii)}{\ge} -\eta \sqrt{2|\theta(\Phi(w_{t+1})) - \theta(\Phi(w_t))|} \cdot \|\nabla f(w_t; z_t)\| - \eta \sqrt{2\theta(\Phi(w_t))} \cdot \|\nabla f(w_t; z_t)\| \\
&\qquad - \frac{\eta^2}{2} \|\nabla f(w_t; z_t)\|^2 \\
&\overset{(iv)}{\ge} -\frac{|\theta(\Phi(w_{t+1})) - \theta(\Phi(w_t))|}{2} - \eta \sqrt{2\theta(\Phi(w_t))} \cdot \|\nabla f(w_t; z_t)\| - \frac{3\eta^2}{2} \|\nabla f(w_t; z_t)\|^2
\end{aligned}
$$

where $(i)$ follows from Cauchy-Schwarz inequality, and $(ii)$ holds due to Lemma 9. Inequality $(iii)$ follows from subadditivity of sq-root. Finally, $(iv)$ follows from an application of AM-GM inequality. Rearranging the terms, we get

$$\theta(\Phi(w_{t+1})) - \theta(\Phi(w_t)) \ge \underbrace{-2\eta \sqrt{2\theta(\Phi(w_t))} \cdot \|\nabla f(w_t; z_t)\| - 3\eta^2 \|\nabla f(w_t; z_t)\|^2}_{=:A'_t} . \qquad (52)$$

Note that $A'_t$, defined to be the terms on the RHS above, is $\mathcal{F}_t$-measurable.

The bounds in (51) and (52) imply that the processes $\{A'_t\}_{t \geq 0}$ and $\{B'_t\}_{t \geq 0}$ are $\mathcal{F}_t$-measurable and satisfy

$$A'_t \leq \theta(\Phi(w_{t+1})) - \theta(\Phi(w_t)) \leq B'_t$$

for any $t \geq 0$. Plugging this in (50), we get

$$A_t := A'_t + C_t \leq Y_{t+1} - Y_t \leq B'_t + C_t =: B_t.$$

Clearly both $A_t$ and $B_t$ are $\mathcal{F}_t$-measurable and satisfy

$$|B_t - A_t| \leq 5\eta\sqrt{\theta(\Phi(w_t))} \cdot \|\nabla f(w_t; z_t)\| + 4\eta^2 \|\nabla f(w_t; z_t)\|^2$$
$$\leq 5\eta\sqrt{M} \cdot \|\nabla f(w_t; z_t)\| + 4\eta^2 \|\nabla f(w_t; z_t)\|^2,$$

where the last line follow from the fact that $t \leq \tau$ and thus $\Phi(w_t) \leq \rho^{-1}(\kappa\rho(\Phi(w_0)))$ which implies that $\theta(\Phi(w_t)) \leq \theta(\rho^{-1}(\kappa\rho(\Phi(w_0)))) =: M$ since $\theta$ is a monotonically increasing function.

**High probability bound.** An application of Azuma's inequality (Lemma 4) implies that for any $t \geq 0$, with probability at least $1 - 1/20T$,

$$Y_t - Y_0 \leq \sqrt{\frac{1}{2}\sum_{j=0}^{t-1}\left(5\eta\sqrt{M} \cdot \|\nabla f(w_j; z_j)\| + 4\eta^2 \|\nabla f(w_j; z_j)\|^2\right)^2 \log(20T)}.$$

The desired statement follows by taking a union bound in the above for $t$ from 0 to $T - 1$. $\qquad\square$

# D    Proofs from Section 5

## D.1    Phase retrieval

For any $w \in \mathbb{R}^d$, the population loss for phase retrieval is given by

$$F(w) = \mathbb{E}_{a \sim \mathcal{N}(0, \mathrm{I}_d)}\left[\left((a^\top w)^2 - (a^\top w^*)^2\right)^2\right]. \tag{53}$$

Throughout this section, we will assume that the optimal parameter $w^*$ satisfies $\|w^*\| = 1$. The following technical lemma establishes some useful properties of $F$.

**Lemma 12.** *Suppose $\|w^*\| = 1$. Then, the function $F$ given in (53) satisfies for any $w \in \mathbb{R}^d$,*

(a) $F(w) = w^\top(\mathrm{I} - (w^*)(w^*)^\top)w + \frac{3}{4}\left(\|w\|^2 - 1\right)^2$.

(b) $\langle w^*, \nabla F(w)\rangle = 3(\|w\|^2 - 1)\langle w, w^*\rangle$.

(c) $\|\nabla F(w)\|^2 = 12\|w\|^2 F(w) - 8\left(\|w\|^2 - \langle w, w^*\rangle^2\right)$.

(d) $F(w) \geq (\|w\|^2 - 1)^2$.

(e) *if $F(w) \leq 1/4$, then $w$ must satisfy $\langle w, w^*\rangle^2 \geq 1/4$.*

**Proof of Lemma 12.** We prove each part separately below:

(a) The proof is straightforward. We refer the reader to Section 2.3 of Candes et al. [11] for the proof.

(b) Note that

$$\nabla F(w) = 2w - 2\langle w, w^*\rangle w^* + 3(\|w\|^2 - 1)w.$$

Thus,

$$\langle w^*, \nabla F(w)\rangle = 2\langle w, w^*\rangle - 2\langle w, w^*\rangle\|w^*\|^2 + 3(\|w\|^2 - 1)\langle w, w^*\rangle$$
$$= 2\langle w, w^*\rangle - 2\langle w, w^*\rangle + 3(\|w\|^2 - 1)\langle w, w^*\rangle$$
$$= 3(\|w\|^2 - 1)\langle w, w^*\rangle,$$

where the second line above holds because $\|w^*\|^2 = 1$.

$(c)$ We have

$$
\begin{aligned}
\|\nabla F(w)\|^2 &= \|2w - 2\langle w, w^*\rangle w^* + 3(\|w\|^2 - 1)w\|^2 \\
&= 4\|w\|^2 + 4\langle w, w^*\rangle^2 \|w^*\|^2 + 9(\|w\|^2 - 1)^2 \|w\|^2 - 8\langle w, w^*\rangle^2 \\
&\quad - 12(\|w\|^2 - 1)\langle w, w^*\rangle^2 + 12(\|w\|^2 - 1)\|w\|^2 \\
&= -12\|w\|^2\langle w, w^*\rangle^2 + 12\|w\|^4 + 9(\|w\|^2 - 1)^2\|w\|^2 + 8\langle w, w^*\rangle^2 - 8\|w\|^2 \\
&= 12\|w\|^2\big(\|w\|^2 - \langle w, w^*\rangle^2 + \frac{3}{4}(\|w\|^2 - 1)^2\big) - 8\big(\|w\|^2 - \langle w, w^*\rangle^2\big) \\
&= 12\|w\|^2 F(w) - 8\big(\|w\|^2 - \langle w, w^*\rangle^2\big),
\end{aligned}
$$

where the equality in the third line holds because $\|w^*\|^2 = 1$ and the last line follows from the definition of the function $F(w)$ in part-(a) of this lemma.

$(d)$ An application of Jensen's inequality implies that

$$
\begin{aligned}
F(w) &= \mathbb{E}_a\big[((a^\top w)^2 - (a^\top w^*)^2)^2\big] \\
&\geq \big(\mathbb{E}_a[(a^\top w)^2 - (a^\top w^*)^2]\big)^2 \\
&= (\|w\|^2 - \|w^*\|^2)^2,
\end{aligned}
$$

where the last line follow from the fact that for any $w$, we have $\mathbb{E}_{a\sim\mathcal{N}(0,\mathrm{I})}\big[(a^\top w)^2\big] = \|w\|^2$. The desired statement follows since $\|w^*\| = 1$.

$(e)$ An application of Lemma 12-(d) implies that

$$
\big(\|w\|^2 - 1\big)^2 \leq F(w) \leq \frac{1}{4},
$$

which implies that $1/2 \leq \|w\|^2 \leq 3/2$. Next, using Lemma 12-(a), we note that

$$
F(w) = w^\top(\mathrm{I} - (w^*)(w^*)^\top)w + \frac{3}{4}\big(\|w\|^2 - 1\big)^2 \geq \|w\|^2 - \langle w, w^*\rangle^2 \geq \frac{1}{2} - \langle w, w^*\rangle^2,
$$

where the last line uses the above derived bound on $\|w\|^2$. Rearranging the terms and using the fact that $F(w) \leq 1/4$ implies that $\langle w, w^*\rangle^2 \geq 1/4$.

$\square$

### D.1.1 Rate of convergence for gradient flow

The next lemma provides a rate of convergence for the phase retrieval population objective.

**Lemma 13.** *Consider the objective function $F$ given in* (53). *Then, for any initial point point $w(0) = w_0$, the point $w(t)$ on its gradient flow path satisfies*

$$
F(w(t)) \leq \min\{F(w_0), F(w_0)e^{-t+\frac{1}{\langle w_0, w^*\rangle^2}}\},
$$

**Proof of Lemma 13.** Let $w(t)$ be the point on the GF path with starting point $w(0) = w_0$. For the ease of notation, define $\alpha(t) = \langle w(t), w^*\rangle^2$ and $\beta(t) = \|w\|^2 - \alpha(t)$. A closer look at the gradient flow dynamics $w'(t) = -\nabla F(w(t))$ reveals that:

$$
\begin{aligned}
\alpha'(t) &= 6\big(\alpha(t) - \alpha(t)^2 - \alpha(t)\beta(t)\big), \\
\beta'(t) &= 2\big(\beta(t) - 3\alpha(t)\beta(t) - 3\beta(t)^2\big). \tag{54}
\end{aligned}
$$

Define the variable $\gamma(t) = \alpha(t)/\beta(t)$ and note that

$$
\begin{aligned}
\gamma'(t) &= \frac{1}{\beta(t)^2}(\beta(t)\alpha'(t) - \alpha(t)\beta'(t)) \\
&= \frac{2}{\beta(t)^2}(\beta(t)\alpha'(t) - \alpha(t)\beta'(t)) \\
&\overset{(i)}{\leq} \frac{4\alpha(t)}{\beta(t)} = 4\gamma(t),
\end{aligned}
$$

where $(i)$ follows from plugging in the relations in (54). Solving the above differential equation implies that $\gamma(t) = \gamma(0)e^{4t}$, which on plugging in the form of $\gamma$ implies that

$$\beta(t) = \alpha(t)\frac{\beta(0)}{\alpha(0)}e^{-4t}. \tag{55}$$

Plugging the above relation in (54) gives us the differential equation

$$\alpha'(t) = 6\left(\alpha(t) - \alpha(t)^2 - 3\alpha(t)^2\frac{\beta(0)}{\alpha(0)}e^{-4t}\right), \tag{56}$$

solving which implies that

$$\alpha(t) = \frac{\alpha(0)e^{6t}}{1 + 3\beta(0)(e^{2t} - 1) + \alpha(0)(e^{6t} - 1)}. \tag{57}$$

Plugging the above form of $\alpha(t)$ in (55) further implies that

$$\beta(t) = \frac{\beta(0)e^{2t}}{1 + 3\beta(0)(e^{2t} - 1) + \alpha(0)(e^{6t} - 1)}. \tag{58}$$

In the rest of the proof, we will show that

$$F(w(t)) \le \min\{F(w(0)), F(w(0))e^{-t+\frac{1}{\alpha(0)}}\}. \tag{59}$$

For the ease of notation, we will use $\alpha$ and $\beta$ to denote $\alpha(0)$ and $\beta(0)$ respectively. There are two natural cases for the above, (a) when $t \le 1/\alpha(0)$ and (b) when $t > 1/\alpha(0)$. In the former case, recalling that the function value is non-increasing along any gradient flow path (Lemma 5) we get that

$$F(w(t)) \le F(w) \le \min\{F(w(0)), F(w(0))e^{-t+\frac{1}{\alpha(0)}}\}.$$

We next show that (59) continues to holds when $t > 1/\alpha(0)$. Note that, from the form of $F$ in Lemma 12-(a), we have

$$\begin{aligned}
F(w(t)) &= \|w(t)\|^2 - \langle w(t), w^*\rangle^2 + \frac{3}{4}\big(\|w\|^2 - 1\big)^2 \\
&= \beta(t) + \frac{3}{4}(\alpha(t) + \beta(t) - 1)^2 \\
&\stackrel{(i)}{=} \frac{\beta e^{2t}}{1 + 3\beta(e^{2t} - 1) + \alpha(e^{6t} - 1)} + \frac{3}{4}\left(\frac{\alpha e^{6t} + \beta e^{2t}}{1 + 3\beta(e^{2t} - 1) + \alpha(e^{6t} - 1)} - 1\right)^2 \\
&= \frac{\beta e^{2t}}{1 + 3\beta(e^{2t} - 1) + \alpha(e^{6t} - 1)} + \frac{3}{4}\left(\frac{-2\beta(e^{2t} - 1) - (1 - \alpha - \beta)}{1 + 3\beta(e^{2t} - 1) + \alpha(e^{6t} - 1)}\right)^2 \\
&\stackrel{(ii)}{\le} \frac{\beta e^{2t}}{1 + 3\beta(e^{2t} - 1) + \alpha(e^{6t} - 1)} + 6\beta \cdot \frac{\beta(e^{2t} - 1)^2}{(1 + 3\beta(e^{2t} - 1) + \alpha(e^{6t} - 1))^2} \\
&\quad + \frac{3}{2}\frac{(1 - \alpha - \beta)^2}{(1 + 3\beta(e^{2t} - 1) + \alpha(e^{6t} - 1))^2}
\end{aligned} \tag{60}$$

where $(i)$ follows by plugging in the relations (57) and (58), and $(ii)$ holds because $(a+b)^2 \le 2a^2 + 2b^2$ for any $a, b \ge 0$. In the following, we bound the three terms on the right hand side of (60) separately for $t \ge 1/\alpha$.

1. *Term I:* Ignoring the positive term $3\beta(e^{2t} - 1)$ in the denominator, we get that

$$\frac{\beta e^{2t}}{1 + 3\beta(e^{2t} - 1) + \alpha(e^{6t} - 1)} \le \frac{\beta e^{2t}}{1 + \alpha(e^{6t} - 1)} \le \frac{1}{3}\beta e^{-t+1/\alpha},$$

   where the last inequality follows from using Lemma 14 (given below).

2. *Term II:* For the second term, we note that

$$\frac{\beta(e^{2t} - 1)^2}{(1 + 3\beta(e^{2t} - 1) + \alpha(e^{6t} - 1))^2} \le \max_{\beta > 0} \frac{\beta(e^{2t} - 1)^2}{(1 + 3\beta(e^{2t} - 1) + \alpha(e^{6t} - 1))^2}$$

$$\overset{(i)}{=} (e^{2t} - 1)\frac{\left(1 + \alpha(e^{6t} - 1)\right)}{3} \cdot \frac{1}{4(1 + \alpha(e^{6t} - 1))^2}$$

$$= \frac{1}{12} \cdot \frac{(e^{2t} - 1)}{1 + \alpha(e^{6t} - 1)}$$

$$\leq \frac{1}{12} \cdot \frac{e^{2t}}{1 + \alpha(e^{6t} - 1)}$$

$$\overset{(ii)}{\leq} \frac{1}{36} \cdot e^{-t + \frac{1}{\alpha}},$$

where $(i)$ holds because the term on the right hand side in the equation above is maximized at $\beta = (1 + \alpha(e^{6t} - 1))/3(e^{2t} - 1)$, and $(ii)$ follow from an application of Lemma 14 (given below).

3. *Term III:* Since the term on the denominator is larger than 1, we have that

$$\frac{(1 - \alpha - \beta)^2}{(1 + 3\beta(e^{2t} - 1) + \alpha(e^{6t} - 1))^2} \leq \frac{(1 - \alpha - \beta)^2}{(1 + 3\beta(e^{2t} - 1) + \alpha(e^{6t} - 1))}$$

$$\leq (1 - \alpha - \beta)^2 \cdot \frac{e^{2t}}{(1 + 3\beta(e^{2t} - 1) + \alpha(e^{6t} - 1))}$$

$$\leq \frac{1}{3} \cdot (1 - \alpha - \beta)^2 \cdot e^{-t + \frac{1}{\alpha}},$$

where the inequality in the second last line holds for any $t \geq 0$ and the last line is due to Lemma 14 (given below).

Plugging the above three bounds in (60), we get that for any $t \geq \frac{1}{\alpha}$,

$$F(w) \leq \frac{1}{3}\beta e^{-t + 1/\alpha} + \frac{1}{6}\beta e^{-t + \frac{1}{\alpha}} + \frac{1}{2} \cdot (1 - \alpha - \beta)^2 \cdot e^{-t + \frac{1}{\alpha}}$$

$$\leq \left(\beta + \frac{3}{4}(1 - \alpha - \beta)^2\right)e^{-t + \frac{1}{\alpha}}$$

$$= F(w(0))e^{-t + \frac{1}{\langle w(0), w^* \rangle^2}},$$

where in the last line we used the form of $F$ from Lemma 12-(a) and the fact that $\alpha = \alpha(0) = \langle w(0), w^* \rangle^2$ and $\beta = \beta(0) = \|w(0)\|^2 - \langle w(0), w^* \rangle^2$. Finally, using Lemma 5, we note that $F(w(t)) \leq F(w)$. Combining these two bounds gives us the relation in (59) for any $t \geq 1/\alpha(0)$. $\quad\square$

**Lemma 14.** *For any $\alpha > 0$ and $t \geq 1/\alpha$,*

$$\frac{e^{2t}}{1 + \alpha(e^{6t} - 1)} \leq \frac{e^{-t + \frac{1}{\alpha}}}{3}.$$

**Proof of Lemma 14.** We consider two cases when $\alpha \geq 1$ and when $\alpha < 1$ separately below:

1. *Case 1: $\alpha \geq 1$:* Define $g(t) = e^{3t}/(1 + \alpha(e^{6t} - 1))$ and note that $g$ is a non-increasing function of $t$ for $\alpha \geq 1$. Thus, for any $t \geq 1/\alpha$,

$$g(t) \leq g(\tfrac{1}{\alpha}) = \frac{e^{\frac{3}{\alpha}}}{1 + \alpha(e^{\frac{6}{\alpha}} - 1)} \leq \frac{1}{3}e^{\frac{1}{\alpha}},$$

where the last inequality holds because the function $\zeta(z) = e^z/3 - e^{3z}/(1 + (e^{6z} - 1)/z)$ is non-negative whenever $z \geq 0$. Multiplying both the sides by $e^{-t}$ gives the desired relation.

2. *Case 2: $\alpha < 1$:* In this case, ignoring positive terms in the denominator (since $1 - \alpha > 0$), we get

$$\frac{e^{2t}}{1 + \alpha(e^{6t} - 1)} = \frac{e^{2t}}{1 - \alpha + \alpha e^{6t}} \leq \frac{e^{2t}}{\alpha e^{6t}} = e^{-t} \cdot \frac{e^{-3t}}{\alpha} \leq \frac{1}{3}e^{-t + 1/\alpha},$$

where the second to last inequality follows from the fact that $e^{-3t}$ is a decreasing function of $t$ and thus for $t \geq 1/\alpha$, we have that $e^{-3t} \leq e^{-3/\alpha}$. The last inequality holds because $\frac{1}{\alpha}e^{-\frac{3}{\alpha}} \leq \frac{1}{3}e^{1/\alpha}$ for any $\alpha > 0$.

$\square$

The next lemma shows that the rate function in Lemma 13 is admissible.

**Lemma 15.** *Consider the function $R$ defined as*

$$R(w,t) = \min\{F(w), F(w)e^{-t+\frac{1}{\langle w,w^*\rangle^2}}\}.$$

*Then, $R$ is an admissible rate of convergence for the objective function $F$.*

**Proof of Lemma 15.** Recall that a sufficient conditions for a rate function $R$ to be admissible w.r.t. the objective $F$ is that for any point $w$,

$$\int_{t=0}^{\infty}\Big(\frac{\partial R(w,t)}{\partial t} + \langle \nabla_w R(w,t), \nabla F(w)\rangle\Big)\,\mathrm{d}t \ge 0. \tag{61}$$

Since the function $R$ is not differentiable at $t = 1/\langle w,w^*\rangle^2$, we use the following definition of the partial derivative

$$\frac{\partial R(w,t)}{\partial t} = \begin{cases} 0 & \text{for} \quad t \le 1/\langle w,w^*\rangle^2 \\ -F(w)e^{-t+\frac{1}{\langle w,w^*\rangle^2}} & \text{for} \quad t > 1/\langle w,w^*\rangle^2 \end{cases},$$

and

$$\nabla_w R(w,t) = \begin{cases} \nabla F(w) & \text{for} \quad t \le 1/\langle w,w^*\rangle^2 \\ \Big(\nabla F(w) - 2\frac{F(w)w^*}{\langle w,w^*\rangle^3}\Big)\cdot e^{-t+\frac{1}{\langle w,w^*\rangle^2}} & \text{for} \quad t > 1/\langle w,w^*\rangle^2 \end{cases}.$$

Thus, we get that

$$\int_{t=0}^{\infty}\frac{\partial R(w,t)}{\partial t}\,\mathrm{d}t = -F(w),$$

and

$$\int_{t=0}^{\infty}\langle \nabla_w R(w,t),\nabla F(w)\rangle\,\mathrm{d}t = \int_{t=0}^{\frac{1}{\langle w,w^*\rangle^2}}\langle \nabla_w R(w,t),\nabla F(w)\rangle\,\mathrm{d}t + \int_{\frac{1}{\langle w,w^*\rangle^2}}^{t=\infty}\langle \nabla_w R(w,t),\nabla F(w)\rangle\,\mathrm{d}t$$

$$= \int_{t=0}^{\frac{1}{\langle w,w^*\rangle^2}}\|\nabla F(w)\|^2\,\mathrm{d}t$$

$$\qquad + \int_{\frac{1}{\langle w,w^*\rangle^2}}^{\infty}\Big(\|\nabla F(w)\|^2 - 2F(w)\frac{\langle \nabla F(w),w^*\rangle}{\langle w,w^*\rangle^3}\Big)\cdot e^{-t+\frac{1}{\langle w,w^*\rangle^2}}\,\mathrm{d}t$$

$$= \int_{t=0}^{\frac{1}{\langle w,w^*\rangle^2}}\|\nabla F(w)\|^2\,\mathrm{d}t + \int_{0}^{\infty}\Big(\|\nabla F(w)\|^2 - 2F(w)\frac{\langle \nabla F(w),w^*\rangle}{\langle w,w^*\rangle^3}\Big)\cdot e^{-t}\,\mathrm{d}t$$

$$= \frac{\|\nabla F(w)\|^2}{\langle w,w^*\rangle^2} + \|\nabla F(w)\|^2 - 2F(w)\frac{\langle \nabla F(w),w^*\rangle}{\langle w,w^*\rangle^3}$$

$$= \frac{\|\nabla F(w)\|^2}{\langle w,w^*\rangle^2} + \|\nabla F(w)\|^2 - 6F(w)\frac{(\|w\|^2 - 1)}{\langle w,w^*\rangle^2},$$

where the last line follows from the fact that $\nabla F(w) = 3(\|w\|^2 - 1)w$. Plugging the above in (61), we get that a sufficient condition for $R$ to be an admissible rate of convergence is that

$$\frac{\|\nabla F(w)\|^2}{\langle w,w^*\rangle^2} + \|\nabla F(w)\|^2 - F(w)\Big(\frac{6(\|w\|^2 - 1)}{\langle w,w^*\rangle^2} + 1\Big) \ge 0,$$

or equivalently that

$$\|\nabla F(w)\|^2 + \langle w,w^*\rangle^2\|\nabla F(w)\|^2 - F(w)\big(6\|w\|^2 - 6 + \langle w,w^*\rangle^2\big) \ge 0. \tag{62}$$

We next observe that (62) holds if

$$0 \le \|\nabla F(w)\|^2 - F(w)\big(6\|w\|^2 - 6 + \langle w,w^*\rangle^2\big)$$

$$\stackrel{(i)}{=} 12\|w\|^2 F(w) - 8\big(\|w\|^2 - \langle w, w^*\rangle^2\big) - F(w)\big(6\|w\|^2 - 6 + \langle w, w^*\rangle^2\big)$$

$$= F(w)\big(6\|w\|^2 - \langle w, w^*\rangle^2 + 6\big) - 8\big(\|w\|^2 - \langle w, w^*\rangle^2\big)$$

$$\stackrel{(ii)}{=} \Big(\|w\|^2 - \langle w, w^*\rangle^2 + \frac{3}{4}\big(\|w\|^2 - 1\big)^2\Big)\big(6\|w\|^2 - \langle w, w^*\rangle^2 + 6\big) - 8\big(\|w\|^2 - \langle w, w^*\rangle^2\big), \quad (63)$$

where the $(i)$ and $(ii)$ follow by plugging in the forms of $\|\nabla F(w)\|^2$ and $F(w)$ from Lemma 12. In the following, we argue that the relation (63) holds for any $w$.

Consider the 2d function

$$\Lambda(\alpha, \beta) := \Big(\beta + \frac{3}{4}(\alpha + \beta - 1)^2\Big)(5\alpha + 6\beta + 6) - 8\beta$$

and note that $\Lambda(\alpha, \beta) \geq 0$ whenever $\alpha \geq 0$ and $\beta \geq 0$ (this can be easily checked by plotting the two dimensional function $\Lambda$). Setting $\alpha = \langle w, w^*\rangle^2$ and $\beta = \|w\|^2 - \langle w, w^*\rangle^2$, we note that both $\alpha, \beta \geq 0$ and so (63) follows immediately, which further implies that the relation in (62) holds. Thus, the sufficient conditions for $R$ to be an admissible rate of convergence hold, and the statement of the lemma follows. $\qquad\square$

**Proof of Lemma 1.** We prove the rate of convergence in Lemma 13 and show its admissibility in Lemma 15 above. $\qquad\square$

### D.1.2 Potential function and self-bounding regularity conditions

Consider the function

$$R(w, t) = \min\{F(w), F(w)e^{-t + \frac{1}{\langle w, w^*\rangle^2}}\}.$$

Lemma 13 and Lemma 15 imply that $R$ is an admissible rate of convergence for the objective function $F$. Thus, using Theorem 2 with $g(z) = z$, we get that the function $\Phi$ constructed in the following is an admissible potential function for $F$,

$$\Phi(w) = \int_{t=0}^{\infty} R(w, t)\,dt$$

$$= \int_{t=0}^{\infty} \min\{F(w), F(w)e^{-t + \frac{1}{\langle w, w^*\rangle^2}}\}\,dt$$

$$= \int_{t=0}^{t = \frac{1}{\langle w, w^*\rangle^2}} \min\{F(w), F(w)e^{-t + \frac{1}{\langle w, w^*\rangle^2}}\}\,dt + \int_{t = \frac{1}{\langle w, w^*\rangle^2}}^{\infty} \min\{F(w), F(w)e^{-t + \frac{1}{\langle w, w^*\rangle^2}}\}\,dt$$

$$= \int_{t=0}^{t = \frac{1}{\langle w, w^*\rangle^2}} F(w)\,dt + \int_{t = \frac{1}{\langle w, w^*\rangle^2}}^{\infty} F(w)e^{-t + \frac{1}{\langle w, w^*\rangle^2}}\,dt$$

$$= \frac{F(w)}{\langle w, w^*\rangle^2} + F(w). \tag{64}$$

We first establish the self-bounding regularity conditions for $F$.

**Lemma 16.** *Let $\|w^*\| = 1$. For any point $w$,*

$$\|\nabla F(w)\|^2 \leq 12 F(w)^{3/2} + 12 F(w)$$

*and*

$$\|\nabla^2 F(w)\| \leq 10 + 9\sqrt{F(w)}.$$

**Proof of Lemma 16.** We first bound $\|\nabla F(w)\|^2$. Using Lemma 12-(c), we have that

$$\|\nabla F(w)\|^2 = 12\|w\|^2 F(w) - 8\big(\|w\|^2 - \langle w, w^*\rangle^2\big)$$

$$\leq 12\|w\|^2 F(w)$$

$$\leq 12\big(\big|\|w\|^2 - \|w^*\|^2\big| + \|w^*\|^2\big)F(w)$$

$$\leq 12\big(\sqrt{F(w)} + \|w^*\|^2\big)F(w)$$

$$= 12F(w)^{3/2} + 12F(w),$$

where the first inequality holds because $\|w\|^2 - \langle w, w^* \rangle^2 \geq 0$ whenever $\|w^*\| \leq 1$, the second inequality is an application of the Triangle inequality and the last inequality follows from Lemma 12-(d). The equality in the last line holds because $\|w^*\| = 1$. Note that the function on the right hand size above is positive and monotonically increasing in $F(w)$.

We next bound $\|\nabla^2 F(w)\|$. From the form of $F$ in Lemma 12-(a), we get that

$$\nabla^2 F(w) = 2\mathrm{I} - 2(w^*)(w^*)^\top + 3(\|w\|^2 - 1)\mathrm{I} + 6ww^\top.$$

Thus, using Triangle inequality, we have

$$\|\nabla^2 F(w)\| \leq 2 + 2\|w^*\|^2 + 3(\|w\|^2 - 1) + 6\|w\|^2 = 10 + 9(\|w\|^2 - 1) \leq 10 + 9\sqrt{F(w)},$$

where the equality in the second line holds because $\|w^*\| = 1$ and the last line is due to Lemma 12-(d).
□

We next establish self-bounding regularity conditions for the potential function $\Phi$.

**Lemma 17.** *Let $\|w^*\| = 1$. For any point $w$, The function $\Phi$ defined in (64) satisfies for any point $w$,*

$$\|\nabla \Phi(w)\| \leq 39\Phi(w)^2 + 17,$$

*and*

$$\|\nabla^2 \Phi(w)\| \leq 54\Phi(w)^3 + 215\Phi^2(w) + 23\Phi(w) + 79 \leq 300\Phi(w)^3 + 100.$$

**Proof of Lemma 17.** Before delving into self-bounding regularity conditions for $\Phi$, we first derive an upper bound on $1/\langle w, w^* \rangle^2$. Note that

$$\begin{aligned}
\left| 1 - \langle w, w^* \rangle^2 \right| &\leq \left| 1 - \|w\|^2 \right| + \left| \|w\|^2 - \langle w, w^* \rangle^2 \right| \\
&\leq \sqrt{F(w)} + \frac{3}{4}\left| (\|w\|^2 - 1)^2 - F(w) \right| \\
&\leq \sqrt{F(w)} + \frac{3}{4}\left| \|w\|^2 - 1 \right|^2 + F(w) \\
&\leq \sqrt{F(w)} + 2F(w),
\end{aligned}$$

where the first and the third inequality above follows from Triangle inequality, and the second and the forth inequalities are due to Lemma 12-(a, d). Squaring both the sides, we get that

$$1 + \langle w, w^* \rangle^4 - 2\langle w, w^* \rangle^2 \leq 2F(w) + 8F(w)^2.$$

Ignoring positive terms on the left hand size and dividing both the sides by $\langle w, w^* \rangle^2$, we get that

$$\begin{aligned}
\frac{1}{\langle w, w^* \rangle^2} &\leq 2 + 2\frac{F(w)}{\langle w, w^* \rangle^2} + 8\frac{F(w)^2}{\langle w, w^* \rangle^2} \\
&\leq 2 + 2\Phi(w) + 8F(w)\Phi(w) \\
&\leq 2 + 2\Phi(w) + 8\Phi^2(w) \\
&\leq 3 + 9\Phi^2(w), \tag{65}
\end{aligned}$$

where the inequalities in second and the third line follow from the fact that both $F(w)/\langle w, w^* \rangle^2$ and $F(w)$ are smaller than $\Phi(w)$ (from the definition in (64) and because $F(w) \geq 0$). The last line is due to AM-GM inequality.

We now prove the self-bounding regularity conditions for $w$.

- *Bound on $\|\nabla \Phi(w)\|$.* Note that

$$\nabla \Phi(w) = \frac{\nabla F(w)}{\langle w, w^* \rangle^2} - \frac{2F(w)}{\langle w, w^* \rangle^3}w^* + \nabla F(w).$$

Using Triangle inequality and the fact that $\|w^*\| = 1$, we get

$$\|\nabla \Phi(w)\| \leq \frac{\|\nabla F(w)\|}{\langle w, w^* \rangle^2} + 2\frac{F(w)}{\langle w, w^* \rangle^2} \cdot \frac{1}{|\langle w, w^* \rangle|} + \|\nabla F(w)\|$$

$$\overset{(i)}{\leq} \|\nabla F(w)\| \left( \frac{1}{\langle w, w^* \rangle^2} + 1 \right) + 2\Phi(w) \cdot \frac{1}{|\langle w, w^* \rangle|}$$

$$\overset{(ii)}{\leq} \sqrt{12F(w)^{3/2} + 12F(w)} \left( \frac{1}{\langle w, w^* \rangle^2} + 1 \right) + 2\Phi(w) \cdot \frac{1}{|\langle w, w^* \rangle|}$$

$$\overset{(iii)}{\leq} \sqrt{15F^2(w) + 9} \cdot \left( \frac{1}{\langle w, w^* \rangle^2} + 1 \right) + \Phi^2(w) + \frac{1}{\langle w, w^* \rangle^2}$$

$$\overset{(iv)}{\leq} 4F(w) \left( \frac{1}{\langle w, w^* \rangle^2} + 1 \right) + 3 + \Phi^2(w) + \frac{4}{\langle w, w^* \rangle^2},$$

where $(i)$ holds because $F(w)/\langle w, w^* \rangle^2 \leq \Phi(w)$, $(ii)$ is due to Lemma 16 and $(iii)$ follows from multiple applications of AM-GM inequality. The inequality $(iv)$ is due to subadditivity of square-root and from rearranging the terms. Plugging in the bound in (65) and the definition in (64) in the above, we get that

$$\|\nabla\Phi(w)\| \leq 37\Phi(w)^2 + 4\Phi(w) + 15$$
$$\leq 39\Phi(w)^2 + 17, \tag{66}$$

where the last line holds due to AM-GM inequality.

- *Bound on $\|\nabla^2\Phi(w)\|$.* Note that

$$\nabla^2\Phi(w) = \nabla^2 F(w) \left( \frac{1}{\langle w, w^* \rangle^2} + 1 \right) - 2\frac{\nabla F(w)(w^*)^\top}{\langle w, w^* \rangle^3} - 2\frac{w^*(\nabla F(w))^\top}{\langle w, w^* \rangle^3} + 6\frac{F(w) \cdot (w^*)(w^*)^\top}{\langle w, w^* \rangle^4}. \tag{67}$$

Using Triangle inequality, Cauchy Schwartz inequality and the fact that $\|w^*\| = 1$, we get

$$\|\nabla^2\Phi(w)\| \leq \|\nabla^2 F(w)\| \left( \frac{1}{\langle w, w^* \rangle^2} + 1 \right) + 4\frac{\|\nabla F(w)\|}{|\langle w, w^* \rangle|^3} + 6\frac{F(w)}{\langle w, w^* \rangle^4}.$$

We bound each of the terms separately below:

(a) *Term I:* Using Lemma 16, we get that

$$\|\nabla^2 F(w)\| \left( \frac{1}{\langle w, w^* \rangle^2} + 1 \right) \leq (10 + 9\sqrt{F(w)}) \left( \frac{1}{\langle w, w^* \rangle^2} + 1 \right)$$

$$\leq 10 + \frac{10}{\langle w, w^* \rangle^2} + \frac{9\sqrt{F(w)}}{\langle w, w^* \rangle^2}$$

$$\leq 10 + \frac{10}{\langle w, w^* \rangle^2} + \frac{9}{2\langle w, w^* \rangle^2} + \frac{9}{2}\frac{F(w)}{\langle w, w^* \rangle^2}$$

$$\leq 55 + 135\Phi^2(w) + \frac{9}{2}\Phi(w),$$

where the second line is due to AM-GM inequality and the last line follows from plugging in (64) and (65).

(b) *Term II:* Using the bound from Lemma 16, we get

$$4\frac{\|\nabla F(w)\|}{|\langle w, w^* \rangle|^3} \leq 4\sqrt{12F(w)^{3/2} + 12F(w)} \cdot \frac{1}{|\langle w, w^* \rangle^3|}$$

$$\leq (16F(w) + 12) \cdot \frac{1}{|\langle w, w^* \rangle^3|}$$

$$\leq 16\Phi(w) \cdot \frac{1}{|\langle w, w^* \rangle|}$$

$$\leq 24 + 80\Phi^2(w),$$

where the line line is due to AM-GM inequality and subadditivity of square-root, the third line is due to (64), the forth line again uses AM-GM inequality and the last line follows from plugging in the bound in (65).

(c) *Term III:* Using the fact that $F(w)/\langle w, w^* \rangle^2 \leq \Phi(w)$ from (64), we get that

$$\frac{6F(w)}{\langle w, w^* \rangle^4} \leq 6\Phi(w) \cdot \frac{1}{\langle w, w^* \rangle^2}$$
$$\leq 18\Phi(w) + 54\Phi(w)^3,$$

where the second inequality follows by plugging (65).

Plugging the three bounds above in (67), we get that

$$\|\nabla^2 \Phi(w)\| \leq 54\Phi(w)^3 + 215\Phi^2(w) + 23\Phi(w) + 79.$$

$\square$

### D.1.3   GD for phase retrieval

In the following, we provide the convergence guarantee for GD algorithm. We first define the respective problem dependent quantities and instantiate Theorem 4 to provide an $O(1/T)$ bound for GD. We then provide a refined analysis which improves this bound to $O(e^{-T})$.

**$O(1/T)$ rate by direct application of Theorem 4.**

- Setting $g(z) = z$ implies the potential function

$$\Phi(w) = \frac{F(w)}{\langle w, w^* \rangle^2} + F(w).$$

- Assumption 1 follows from Lemma 16 which implies that

$$\psi(z) = 12z^{3/2} + 12z.$$

- Assumption 2 follows from Lemma 17 which implies that

$$\rho(z) = 300z^3 + 100.$$

- The function $\theta$ is given by

$$\theta(z) = \int_{y=0}^{z} \frac{1}{\rho(y)} \, \mathrm{d}y \leq \frac{z}{100}.$$

- The monotonically increasing function $\zeta$ is defined such that

$$\zeta^{-1}(z) = \int_{y=0}^{z} \frac{g(y)}{\psi(y)} \, \mathrm{d}y = \frac{1}{6} \left( \sqrt{z} - \log(1 + \sqrt{z}) \right) \geq \frac{1}{12} \sqrt{z},$$

which implies that

$$\zeta(z) \leq 144z^2.$$

Note that the function $\frac{\psi(z)}{g(z)} = 12\sqrt{z} + 12$ is clearly a monotonically increasing function of $z$. Thus, plugging the above problem-dependent constants in Theorem 4 implies that setting $\eta$ such that

$$\eta \propto \frac{1}{(1 + \Phi(w_0))(1 + \Phi(w_0)^3)}$$

implies that GD for any $T \geq 1$ has the rate

$$g(F(\widehat{w}_T)) \lesssim \frac{\Phi(w_0) + \Phi(w_0)^8}{T}, \tag{68}$$

where recall that $\Phi(w_0) = \frac{F(w_0)}{\langle w_0, w^* \rangle^2} + F(w_0).$

$O(e^{-T})$ **rate via a refined analysis.** We can further improve over the rate in (68) by a refined analysis for GD. In the following, we will show that GD in fact enjoys a $e^{-O(T-\tau)}$ rate of convergence for GD for all $T \geq \tau$, where $\tau$ depends on $w_0$ and problem dependent parameters specified below.

Before delving into the proof of the above, we first provide the relevant improved version of problem dependent parameters that hold for any $w$ for which $F(w) \leq 1$:

- Assumption 1 follows from Lemma 16 which implies that
$$\psi(z) = 24z.$$

- Assumption 2 follows from Lemma 17 which implies that
$$\rho(z) = 400.$$

- The function $\theta$ is given by
$$\theta(z) = \int_{y=0}^{z} \frac{1}{\rho(y)}\, \mathrm{d}y = \frac{z}{400}. \tag{69}$$

We are now ready to provide the improved convergence rate for GD. Note that using (68), there exists some
$$\tau \leq 20\big(\Phi(w_0) + \Phi(w_0)^8\big) \tag{70}$$

for which $F(w_\tau) \leq 1/20$. Using Lemma 12-(e), we get that such a point $w_\tau$ must satisfy $\langle w_\tau, w^* \rangle^2 \geq 1/4$, which implies that
$$\Phi(w_\tau) = \frac{F(w_\tau)}{\langle w_\tau, w^* \rangle^2} + F(w_\tau) \leq 5F(w_\tau) \leq \frac{1}{4}.$$

In the following, we first show via induction that $\langle w_t, w^* \rangle \geq 1/4$ and $\Phi(w_t) \leq 1/4$ for all $t \geq \tau$. As shown above, the base case for $t = \tau$ holds. For the induction step, consider any $t \geq \tau$ and assume that $\langle w_t, w^* \rangle^2 \geq 1/4$ and $\Phi(w_t) \leq 1/4$; we will show that the same holds for $w_{t+1}$. Starting from (36) in the proof of Theorem 4, we note that
$$\theta(\Phi(w_{t+1})) \leq \theta(\Phi(w_t)) - \frac{\eta}{2\rho(\Phi(w_0))}g(F(w_t))). \tag{71}$$

However, also note that $w_t$ satisfies,
$$F(w_t) \leq \Phi(w_t) = \frac{F(w_t)}{\langle w_t, w^* \rangle^2} + F(w_t) \leq 5F(w_t), \tag{72}$$

where the last inequality holds since $\langle w_t, w^* \rangle^2 \geq 1/4$ by induction hypothesis. Plugging the relation (72) in (71) and using the fact that $g(z) = z$, we get that
$$\theta(\Phi(w_{t+1})) \leq \theta(\Phi(w_t)) - \frac{\eta}{10\rho(\Phi(w_0))}\Phi(w_t),$$

Plugging in the value of $\theta$ and $\rho$ from (69) in the above, we get that
$$\Phi(w_{t+1}) \leq \Phi(w_t) - \frac{\eta}{10}\Phi(w_t)$$
$$= \Phi(w_t)\Big(1 - \frac{\eta}{10}\Big). \tag{73}$$

The above clearly implies that $\Phi(w_{t+1}) \leq \Phi(w_t) \leq 1/4$. Furthermore, from the definition of $\Phi$, we immediately get that $F(w_{t+1}) \leq 1/4$, plugging which in Lemma 12-(e) implies that $\langle w_{t+1}, w^* \rangle^2 \geq 1/4$. This completes the induction step hence showing that $\langle w_t, w^* \rangle \geq 1/4$ and $\Phi(w_t) \leq 1/4$ holds for all $t \geq \tau$.

Now, in order to complete the proof of convergence, note that (73) will hold for all $t \geq \tau$, recursing which implies that
$$\Phi(w_T) \leq \Phi(w_\tau)\Big(1 - \frac{\eta}{10}\Big)^{T-\tau} \leq \Phi(w_\tau)e^{-\eta(T-\tau)/10} \leq \frac{1}{4}e^{-\eta(T-\tau)/10},$$

where the last inequality holds since $\Phi(w_\tau) \leq 1/4$.

Plugging in the value of $\tau$ from (73), we get that for all $T \geq \tau = 20\big(\Phi(w_0) + \Phi(w_0)^8\big)$, GD has convergence rate
$$F(w_T) \leq \Phi(w_T) \leq \frac{1}{4}e^{-\frac{\eta(T-\tau)}{10}}. \tag{74}$$

#### D.1.4 SGD for phase retrieval

We build on the problem dependent quantities introduced in Appendix D.1.3. Suppose SGD is run with stochastic gradient estimates that satisfy Assumption 3 with

$$\chi(z) = \min\{\sqrt{z}, c\},$$

where $c$ is a universal constant. Such a bound is satisfied when the stochastic gradient estimate is computed by using samples from $\mathcal{S}$ where a fresh sample is used for each estimate, i.e. $\nabla f(w; (a, y)) = 4((a^\top w)^2 - y)(a^\top w)w$ (c.f. Candes et al. [11, Lemma 7.4, 7.7]). Using the above, we define the function $\Lambda$ used in Theorem 5 as

$$\Lambda(z) = 24z^{3/2} + 24z + 2\min\{\sqrt{z}, c\}.$$

Fixing any $\bar{w}$ such that $F(\bar{w}) \geq F(w_0)$, set $\kappa = F(\bar{w})/F(w_0)$, and define $B = 24\Phi(\bar{w})^3 + 24\Phi(\bar{w})^2 + 2\min\{\Phi(\bar{w}), c\} \lesssim (1 + \Phi(\bar{w})^3)$. The following guarantee is due to Theorem 5 (in particular the bound in Remark 4). Setting

$$\eta \leq \frac{1}{2\log^2(20T)} \cdot \frac{\Phi(\bar{w}) - \Phi(w_0)}{\sqrt{B\Phi(\bar{w})T}},$$

the point returned by SGD algorithm after $T$ iterations satisfies with probability at least 0.7,

$$g(F(\widehat{w}_T)) \lesssim \rho(\Phi(\bar{w})) \cdot \frac{\Phi(\bar{w})}{\Phi(\bar{w}) - \Phi(w_0)} \cdot \sqrt{B\Phi(\bar{w})} \cdot \frac{1}{\sqrt{T}},$$

where recall that $\Phi(w) = \frac{F(w)}{\langle w, w^* \rangle^2} + F(w)$. Since $g(z) = z$, the above immediately implies a bound on $F(\widehat{w}_T)$.

### D.2 Proof of Lemma 2

The proof of Lemma 2 follows by defining a rate function which holds for every initial point. We then get an admissible potential function by using Theorem 2. The desired self-bounding regularity conditions follow by plugging in the given properties of $\Gamma$ and $h$ in the lemma statement.

**Proof of Lemma 2.** Note that for any initialization $w(0) = w$ for which $h(w) \geq 0$, gradient flow satisfies $F(w(t)) \leq R(w, t)$. Define the function $\widetilde{R}(w, t) = R(w, h(w)t)$. Clearly, for any $w$,

$$F(w(t)) \leq \widetilde{R}(w, t) = R(w, h(w)t).$$

To see the above, note that when $h(w) = 0$, the above relation simply reduces to $F(w(t)) \leq R(w, 0)$ which holds from our assumptions. When $0 < h(w) \leq 1$, we have that $F(w(t)) = R(w, t) \leq R(w, h(w)t)$ which again holds because $R(w, \cdot)$ is monotonically decreasing in $t$ and because $h(w) \leq 1$.

Next, using the premise that $\widetilde{R}$ is admissible rate function w.r.t. $F$, and Theorem 2, we get that the function $\Phi_g$ defined below is an admissible potential function w.r.t. $F$ with $g(z) = z$,

$$\Phi_g(w) = \int_{t=0}^\infty \bar{R}(w, t)\, \mathrm{d}t = \int_{t=0}^\infty R(w, h(w)t)\, \mathrm{d}t = \frac{\Gamma(w)}{h(w)}.$$

In the following, we show that Assumption 2 (self-bounding regularity conditions) hold for the potential function $\Phi_g$. First note that, for any $w$, the assumption $(h(w) - h(w^*))^2 \leq \mu(\Gamma(w))$ implies that

$$\mu(\Gamma(w)) \geq h(w^*)^2 + h(w)^2 - 2h(w)h(w^*)$$
$$\geq h(w^*)^2 - 2h(w)h(w^*),$$

which after rearranging the terms implies that

$$\frac{1}{h(w)} \leq \frac{1}{h(w^*)^2}\left(2h(w^*) + \frac{\mu(\Gamma(w))}{h(w)}\right)$$
$$\leq \frac{1}{h(w^*)^2}\left(2h(w^*) + \mu\left(\frac{\Gamma(w)}{h(w)}\right)\right)$$

$$= \frac{1}{h(w^*)^2} (2h(w^*) + \mu(\Phi_g(w))), \qquad (75)$$

where the second inequality holds because $h(w) \le 1$ and $\mu$ satisfies the property that $k\pi(z) \le \pi(kz)$ for any $k \ge 1$.

We are now ready to establish the self-bounding regularity properties for $\Phi_g$.

$(a)$ $\|\nabla\Phi_g(w)\|$ *satisfies self-bounding regularity.* Using Chain rule and Triangle inequality, we have that

$$\begin{aligned}
\|\nabla\Phi_g(w)\| &\le \frac{\|\nabla\Gamma(w)\|}{h(w)} + \frac{\Gamma(w)}{h(w)^2} \|\nabla h(w)\| \\
&\overset{(i)}{\le} \frac{\lambda(\Gamma(w))}{h(w)} + \Phi_g(w) \frac{\pi(\Gamma(w))}{h(w)} \\
&\overset{(ii)}{\le} \frac{1}{h(w)} \lambda\left(\frac{\Gamma(w)}{h(w)}\right) + \frac{1}{h(w)} \Phi_g(w) \pi\left(\frac{\Gamma(w)}{h(w)}\right) \\
&= \frac{1}{h(w)} \lambda(\Phi_g(w)) + \frac{1}{h(w)} \Phi_g(w) \pi(\Phi_g(w)) \\
&\overset{(iii)}{\le} \left(\frac{2}{h(w^*)} + \frac{\mu(\Phi_g(w))}{h(w^*)^2}\right) \cdot (\lambda(\Phi_g(w)) + \Phi_g(w)\pi(\Phi_g(w)))
\end{aligned}$$

where $(i)$ holds due to the assumption that $\|\nabla\Gamma(w)\| \le \lambda(\Gamma(w))$ and $\|\nabla h(w)\| \le \pi(\Gamma(w))$, $(ii)$ holds because $\lambda$ and $\pi$ are positive, monotonically increasing functions and $h(w) \le 1$. The equality in the next line follows from the definition of $\Phi_g(w)$, and the inequality $(iii)$ follows from plugging in (75).

Note that the function

$$\zeta(z) = \frac{1}{h(w^*)^2} (2h(w^*) + \mu(z)) \cdot (\lambda(z) + z\pi(z))$$

appearing on the right side above is positive, monotonically increasing.

$(b)$ $\|\nabla^2\Phi_g(w)\|$ *satisfies self-bounding regularity.* Using Chain rule and Triangle inequality, we get that

$$\|\nabla^2\Phi_g(w)\| \le \frac{\|\nabla^2\Gamma(w)\|}{h(w)} + 2\frac{\|\nabla\Gamma(w)\nabla h(w)^\top\|}{h(w)^2} + \frac{\Gamma(w)}{h(w)^3} \|\nabla h(w)\|^2 + \frac{\Gamma(w)}{h(w)^2} \|\nabla^2 h(w)\|. \qquad (76)$$

We bound each of the terms in the RHS above separately, as follows:

- For the first term in (76), using the relation $\|\nabla^2\Gamma(w)\| \le \lambda(\Gamma(w))$, we get

$$\begin{aligned}
\frac{\|\nabla^2\Gamma(w)\|}{h(w)} &\le \frac{\lambda(\Gamma(w))}{h(w)} \\
&\le \lambda(\Gamma(w)) \cdot \left(\frac{2}{h(w^*)} + \frac{\mu(\Phi_g(w))}{h(w^*)^2}\right) \\
&\le \lambda(\Phi_g(w)) \cdot \left(\frac{2}{h(w^*)} + \frac{\mu(\Phi_g(w))}{h(w^*)^2}\right),
\end{aligned}$$

where the second inequality is by plugging in (75), and the last line follows from the fact that $h(w) \le [0,1]$ and from the definition of $\Phi_g(w)$. This proves self-bounding regularity conditions for $\nabla\Phi_g(w)$

- For the second term in (76), using Cauchy-Schwarz inequality, we have

$$\begin{aligned}
\frac{2}{h(w)^2} \|\nabla\Gamma(w)\nabla h(w)^\top\| &\le \frac{2}{h(w)^2} \|\nabla\Gamma(w)\| \|\nabla h(w)\| \\
&\le 2\lambda(\Gamma(w)) \cdot \pi(\Gamma(w)) \cdot \left(\frac{2}{h(w^*)} + \frac{\mu(\Phi_g(w))}{h(w^*)^2}\right)^2
\end{aligned}$$

$$\leq 2\lambda(\Phi_g(w)) \cdot \pi(\Phi_g(w)) \cdot \left(\frac{2}{h(w^*)} + \frac{\mu(\Phi_g(w))}{h(w^*)^2}\right)^2$$

where the second inequality holds because $\|\nabla\Gamma(w)\| \leq \lambda(\Gamma(w))$ and $\|\nabla h(w)\| \leq \pi(\Gamma(w))$, and the last inequality follows from the definition of $\Phi_g(w)$ and the fact that $h(w) \leq 1$.

- For the third term in (76), using the relation $\|\nabla h(w)\| \leq \pi(\Gamma(w))$, we get

$$\frac{\Gamma(w)}{h(w)^3}\|\nabla h(w)\|^2 = \frac{\Gamma(w)}{h(w)} \cdot \frac{1}{h(w)^2} \cdot \pi^2(\Gamma(w))$$

$$\leq \Phi_g(w) \cdot \left(\frac{2}{h(w^*)} + \frac{\mu(\Phi_g(w))}{h(w^*)^2}\right)^2 \cdot \pi^2(\Phi_g(w)),$$

where the last line uses the definition of $\Phi_g$, the fact that $\pi$ is positive and monotonically increasing, $h(w) \leq 1$, and the bound in (75).

- For the fourth term in (76), using the relation $\|\nabla^2 h(w)\| \leq \pi(\Gamma(w))$, we get

$$\frac{\Gamma(w)}{h(w)^2}\|\nabla^2 h(w)\| \leq \frac{\Gamma(w)}{h(w)} \cdot \frac{1}{h(w)} \cdot \pi(\Gamma(W))$$

$$\leq \Phi_g(w) \cdot \left(\frac{2}{h(w^*)} + \frac{\mu(\Phi_g(w))}{h(w^*)^2}\right) \cdot \pi(\Phi_g(w))$$

where the last line uses the definition of $\Phi_g$, the fact that $\pi$ is positive and monotonically increasing and the fact that $h(w) \leq 1$, and the bound in (75).

Clearly, each of the bounds above consists of a positive, monotonically increasing function on the right hand side, thus proving self-bounding regularity conditions for $\nabla^2\Phi_g(w)$.

$\square$

### D.3 Matrix Square root

For any symmetric $W \in \mathbb{R}^{d \times d}$, the population loss for matrix square root problem is given by[6]

$$F(W) = \|W^2 - M\|_F^2, \tag{77}$$

where $M$ is a positive-definite matrix. Note that the global minima of the above objective is obtained at $W = \sqrt{M}$.

The following technical lemma establishes some useful properties of $F$.

**Lemma 18.** *The function $F$ given in* (77) *satisfies for any $W$,*

    $(a)$  $\nabla F(W) = 2(2W^3 - MW - WM)$,

    $(b)$  $\|\nabla F(W)\|_F^2 \geq 16\sigma_d(W^2)F(W)$,

*where $\sigma_d(W)$ denotes the minimum singular value of $W$.*

**Proof.**    $(a)$  The relation follows from Chain rule.

    $(b)$  The proof is identical to the proof of Jain et al. [28, Lemma 4.5]. Note that

$$\langle\nabla F(W), \nabla F(W)\rangle = 4\big\langle(W^2 - M)W + W(W^2 - M), (W^2 - M)W + W(W^2 - M)\big\rangle$$

$$\geq 16\sigma_d(W^2)F(W).$$

$\square$

---

[6]Following the convention, we denote matrix valued variables throughout this section using capital Roman aphabet.

### D.3.1 Rate of convergence for gradient flow

We first note the following technical lemma whose proof is identical to the proof of Jain et al. [28, Lemma 4.2] as $\eta \to 0$.

**Lemma 19** (Jain et al. [28, Lemma 4.2]). *For any initial point $W_0$ and $t \geq 0$, the point $W(t)$ on the gradient flow path with $W(0) = W_0$ satisfies*

$$\sigma_d(W(t)^2) \geq \min\left\{\sigma_d(W_0^2), \frac{\sigma_d(M)}{100}\right\}.$$

Before providing a rate of convergence for GF for the matrix square root problem, we first define additional notation. Let $\alpha = \sigma_d(M)/1600$, and define the function

$$\phi(Z) = \frac{-1}{\gamma}\log(\text{tr}(e^{-\gamma Z}) + e^{-16\alpha\gamma}), \tag{78}$$

and the function

$$h(W) = \sigma\big(\phi(W^2) - \alpha\big), \tag{79}$$

where $\sigma$ denotes a smoothened version of the indicator function and is given by

$$\sigma(z) := \begin{cases} 0 & \text{if} \quad z \leq 0 \\ \frac{2}{\alpha^2}z^2 & \text{if} \quad 0 \leq z \leq \alpha/2 \\ -\frac{2}{\alpha^2}z^2 + \frac{4}{\alpha}z - 1 & \text{if} \quad \alpha/2 \leq z \leq \alpha \\ 1 & \text{if} \quad \alpha \leq z \end{cases}. \tag{80}$$

The following technical lemma establishes some useful properties of the function $\phi$ and $h$.

**Lemma 20.** *Let $\gamma > 0$. For any point $W$, we have*

*(a)* $\min\{\sigma_d(W^2), 16\alpha\} - \frac{\log(d+1)}{\gamma} \leq \phi(W^2) \leq \min\{\sigma_d(W^2), 16\alpha\}$.

*(b)* $\nabla_W \phi(W) = \frac{e^{-\gamma W}}{tr(e^{-\gamma W}) + e^{-16\gamma\alpha}}$ *and* $\nabla_W \phi(W^2) = \frac{2e^{-\gamma W^2}W}{tr(e^{-\gamma W^2}) + e^{-16\gamma\alpha}}$.

*(c)* $\big(h(W) - h(\sqrt{M})\big)^2 \leq \frac{2}{\alpha}F(W)$.

*(d)* $\|\nabla h(W)\| \leq \frac{4}{\alpha}\big(F(W)^{1/4} + \sqrt{\|M\|}\big)$.

*(e)* $\|\nabla^2 h(W)\| \leq 16\big(\frac{2}{\alpha^2} + \frac{1}{\alpha}\big)\big(1 + \gamma\|M\| + \gamma\sqrt{F(W)}\big)$.

*(f)* *if* $F(W) \leq \sigma_d(M)^2/4$, *then $W$ must satisfy* $\sigma_d(W^2) \geq 800\alpha$. *Furthermore, if* $\gamma \geq \frac{\log(d+1)}{\gamma}$, *the $W$ satisfies* $h(W) = 1$.

*where* $\alpha = \sigma_d(M)/1600$.

**Proof of Lemma 20.** We prove each part separately below:

*(a)* For the upper bound, note that

$$\phi(W^2) = \frac{-1}{\gamma}\log(\sum_{i=1}^{d} e^{-\gamma\sigma_i(W^2)} + e^{-16\alpha\gamma})$$

$$\leq \frac{-1}{\gamma}\log(\min\big\{e^{-\gamma\sigma_d(W^2)}, e^{-16\alpha\gamma}\big\})$$

$$= \min\{\sigma_d(W^2), 16\alpha\},$$

where the inequality in the second line holds because $-\log(z)$ is a decreasing function of $z$.

For the lower bound, again using monotonicity of the function $-\log(z)$, we get that

$$\phi(W^2) = \frac{-1}{\gamma}\log(\sum_{i=1}^{d} e^{-\gamma\sigma_i(W^2)} + e^{-16\alpha\gamma})$$

$$\geq \frac{-1}{\gamma}\log((d+1)e^{-\gamma\min\{\sigma_d(W^2), 16\alpha\}})$$

$$\geq \min\{\sigma_d(W^2), 16\alpha\} - \frac{\log(d+1)}{\gamma}.$$

($b$) The proof is a straightforward application of the Chain rule for matrix derivatives.

($c$) Since $\sigma$ is $2/\alpha$-Lipschitz, we have that

$$(h(W) - h(\sqrt{M}))^2 = \big(\sigma(\phi(W^2) - \alpha) - \sigma(\phi(M) - \alpha)\big)^2$$

$$\leq \frac{2}{\alpha}\big(\phi(W^2) - \phi(M)\big)^2$$

$$\leq \frac{2}{\alpha} \sup_{\substack{t\in[0,1]\\ Z=Mt+(1-t)W^2}} \|\nabla_Z \phi(Z)\|_F^2 \cdot \|W^2 - M\|_F^2$$

$$= \frac{2}{\alpha} \sup_{\substack{t\in[0,1]\\ Z=Mt+(1-t)W^2}} \left\|\frac{e^{-\gamma Z}}{\operatorname{tr}(e^{-\gamma Z}) + e^{-16\gamma\alpha}}\right\|_F^2 \cdot \|W^2 - M\|_F^2$$

$$\leq \frac{2}{\alpha}\|W^2 - M\|^2 = \frac{2}{\alpha}F(W),$$

where the inequality in the third line above holds due to Fundamental theorem of calculus and using Cauchy-Schwarz. The inequality is due to the fact that the first term in the product is always smaller than 1.

($d$) Using Chain rule for matrix derivatives, we get that

$$\|\nabla h(W)\| = \sigma'(\phi(W^2) - \alpha)\|\nabla_W \phi(W^2)\|$$

$$\leq \frac{2}{\alpha}\|\nabla_W \phi(W^2)\|$$

$$= \frac{2}{\alpha} \cdot \left\|\frac{2e^{-\gamma W^2}W}{\operatorname{tr}(e^{-\gamma W^2}) + e^{-16\gamma\alpha}}\right\|$$

$$\leq \frac{4}{\alpha} \cdot \left\|\frac{e^{-\gamma W^2}}{\operatorname{tr}(e^{-\gamma W^2}) + e^{-16\gamma\alpha}}\right\|\|W\|,$$

where the first inequality is due to the fact that $\sigma'(z) \leq 2/\alpha$, the equality in the third line is from plugging in the form of $\nabla_W \phi(W^2)$, and the last inequality is due to Cauchy-Schwarz. Using that fact that $\left\|\frac{e^{-\gamma W^2}}{\operatorname{tr}(e^{-\gamma W^2}) + e^{-16\gamma\alpha}}\right\| \leq 1$ and that

$$\|W\| = \sqrt{\|W^2\|} \leq \sqrt{\|W^2 - M\| + \|M\|} \leq \sqrt{\|W^2 - M\|_F + \|M\|} = \sqrt{\sqrt{F(W)} + \|M\|}$$

in the above, we get that

$$\|\nabla h(W)\| \leq \frac{4}{\alpha}\Big(F(W)^{1/4} + \sqrt{\|M\|}\Big).$$

($e$) Using Chain rule for matrix derivatives and Triangle Inequality, we get that

$$\|\nabla^2 h(W)\| \leq 4\gamma\big(\sigma''(\phi(W^2) - \alpha) + \sigma'(\phi(W^2) - \alpha)\big)\left\|\frac{e^{-\gamma W^2}W}{\operatorname{tr}(e^{-\gamma W^2}) + e^{-16\gamma\alpha}}\right\|^2$$

$$+ 2\sigma'(\phi(W^2) - \alpha)\left(\left\|\frac{e^{-\gamma W^2}}{\operatorname{tr}(e^{-\gamma W^2}) + e^{-16\gamma\alpha}}\right\| + 2\gamma\left\|\frac{W^2 e^{-\gamma W^2}}{\operatorname{tr}(e^{-\gamma W^2}) + e^{-16\gamma\alpha}}\right\|\right)$$

$$\leq 4\gamma\big(\sigma''(\phi(W^2) - \alpha) + \sigma'(\phi(W^2) - \alpha)\big)\|W^2\| + 2\sigma'(\phi(W^2) - \alpha)\big(1 + 2\gamma\|W^2\|\big)$$

$$\leq 16\Big(\frac{2}{\alpha^2} + \frac{1}{\alpha}\Big)\big(1 + \gamma\|W^2\|\big),$$

where the second inequality above follows from Cauchy-Schwarz inequality, using the fact that $\left\|\frac{e^{-\gamma W^2}}{\operatorname{tr}(e^{-\gamma W^2}) + e^{-16\gamma\alpha}}\right\| \leq 1$ and from the observation that $W$ is symmetric PD. Using the fact that

$$\|W^2\| \leq \|W^2 - M\| + \|M\| \leq \|W^2 - M\|_F + \|M\| = \sqrt{F(W)} + \|M\|$$

in the above, we get that

$$\|\nabla^2 h(W)\| \leq 16\Big(\frac{2}{\alpha^2} + \frac{1}{\alpha}\Big)\big(1 + \gamma\|M\| + \gamma\sqrt{F(W)}\big).$$

$(f)$ We note that

$$\left|\sigma_d(W^2) - \sigma_d(M)\right|^2 \le \|W^2 - M\|^2 \le \|W^2 - M\|_F^2 = F(W).$$

Thus, for any $W$ for which $F(W) \le (\sigma_d(M)/2)^2$, the above implies that

$$\frac{\sigma_d(M)}{2} \le \sigma_d(W^2) \le \frac{3\sigma_d(M)}{2}.$$

The final bound follows by noting that $\sigma_d(M) = 1600\kappa$. Furthermore, if $\gamma \ge \frac{\log(d+1)}{\gamma}$, then we have that

$$\phi(W^2) - \alpha \ge 14\alpha,$$

which implies that $h(W) = 1$.

$\square$

We next provide a rate of convergence for gradient flow on the matrix square root problem, when the initialization is well behaved.

**Lemma 21** (Lemma 3 in the main body). *Consider the objective function $F$ given in (77). Then, for any initial point $W(0) = W_0$ for which $h(W_0) > 0$, where $h$ is given in (79), the point $w(t)$ on its gradient flow path satisfies*

$$F(W(t)) \le \widetilde{R}(W_0, t) := F(W_0)\exp(-16\alpha t).$$

**Proof.** Due to chain rule, we have that

$$
\begin{aligned}
\frac{\mathrm{d}F(W(t))}{\mathrm{d}t} &= \left\langle \nabla F(W(t)), \frac{\mathrm{d}W(t)}{\mathrm{d}t} \right\rangle \\
&= -\|\nabla F(W(t))\|_F^2 &&\left(\text{since } \tfrac{\mathrm{d}W(t)}{\mathrm{d}t} = -\nabla F(W(t))\right) \\
&\le -16\sigma_d(W(t)^2)F(W(t)) &&(\text{using Lemma 18-(b)} \\
&\le -16\min\left\{\sigma_d(W_0^2), \frac{\sigma_d(M)}{100}\right\}F(W(t)). &&(\text{using Lemma 19})
\end{aligned}
$$

Noting that $F(W(t)) > 0$, rearranging both the sides and integrating with respect to $t$, we get that

$$\int_{\tau=0}^t \frac{1}{F(w(\tau))}\,\mathrm{d}F(W(\tau)) \le -16\min\left\{\sigma_d(W_0^2), \frac{\sigma_d(M)}{100}\right\}\int_{\tau=0}^t \mathrm{d}t.$$

The above implies that

$$F(W(t)) \le F(W(0))\exp\left(-16t\min\left\{\sigma_d(W_0^2), \frac{\sigma_d(M)}{100}\right\}\right)$$

$$\le F(W_0)\exp(-16\alpha t),$$

where the second line above holds since

$$\min\left\{\sigma_d(W_0^2), \frac{\sigma_d(M)}{100}\right\} \ge \phi(W_0^2) \ge \alpha,$$

where the first inequality is due to Lemma 20-(a) and the second inequality holds because $\phi(W_0^2) > \alpha$ since $h(W_0) > 0$.

$\square$

Note that the rate in Lemma 21 holds for any $W$ for which $h(W) > 0$. However, we can extend the above to define a rate function that holds for any $W$. Define

$$R(W, t) = \widetilde{R}(W, t \cdot h(W)) = F(W)e^{-16\alpha h(W)},$$

and note that for any point $W_0$, the GF path from $W_0$ satisfies $F(W(t)) \le R(W_0, t)$. The proof is straightforward: when $W$ is such that $h(W) = 0$, the condition reduces to showing that $F(w(t)) \le \widetilde{R}(W_0, 0) = F(W_0)$ which holds for any GF path (Lemma 5). On the other hand, when $W_0$ is such that $0 < h(W_0) \le 1$, we have that $F(W(t)) \le \widetilde{R}(W_0, t) \le \widetilde{R}(W_0, t \cdot h(W)) = R(W, t)$ since $R$ is monotonically decreasing in $W$.

In the following lemma, we show that the function $R$ is in-fact an admissible rate of convergence w.r.t. $F$, albeit under mild conditions on $\gamma$.

**Lemma 22.** *Let $\gamma \geq \log(d+1)/\alpha$. Consider the function $R$ defined as*

$$R(w,t) = F(W)e^{-16\alpha t h(W)},$$

*where $h$ is given in* (79)*. Then, $R$ is an admissible rate of convergence w.r.t. $F$.*

**Proof of Lemma 22.** Recall that a sufficient conditions for a rate function $R$ to be admissible w.r.t. $F$ is that for any point $W$,

$$\int_{t=0}^{\infty} \left( \frac{\partial R(W,t)}{\partial t} + \langle \nabla_W R(W,t), \nabla F(W) \rangle \right) dt \geq 0. \tag{81}$$

We note that

$$\int_{t=0}^{\infty} \frac{\partial R(W,t)}{\partial t}\, dt = R(W,\infty) - R(W,0) = -F(W)\mathbf{1}\{h(W) > 0\},$$

and due to Chain rule,

$$\int_{t=0}^{\infty} \langle \nabla_w R(w,t), \nabla F(W) \rangle\, dt = \frac{\|\nabla F(W)\|^2}{16\alpha h(W)} - F(W)\frac{\langle \nabla h(W), \nabla F(W) \rangle}{16\alpha h(W)^2}.$$

Taking the two terms together and rearranging, the condition in (81) is equivalent to

$$\|\nabla F(W)\|^2 \geq 16\alpha h(W)F(W)\mathbf{1}\{h(W) > 0\} + \frac{F(W)}{16\alpha h(W)^2}\langle \nabla h(W), \nabla F(W) \rangle, \tag{82}$$

Recall that $h(W) = \sigma(\phi(W^2) - \alpha)$. In the following, we show that the above relation holds for any PD matrix $W$, thus showing that $R$ is an admissible rate of convergence w.r.t. $F$. We divide the proof into the following cases:

- *Case 1: when $\phi(W^2) \leq \alpha$.* In this case, both $h(W) = 0$ and $\nabla h(W)/h(W) = 0$ (by definition) and thus the condition in (82) is trivially satisfied.

- *Case 2: when $\phi(W^2) \geq 2\alpha$.* In this case, $h(W) = 1$ but $\nabla h(W)/h(W) = 0$ (by definition) and thus the condition in (82) reduces to showing that $\|\nabla F(W)^2\| \geq 16\alpha F(W)$, which holds due to Lemma 18-(b) and the fact that $h(W) \geq 2\alpha$ implies that $\sigma_d(W) \geq 2\alpha$ (due to Lemma 20-(a)).

- *Case 3: when $\alpha \leq \phi(W^2) \leq 2\alpha$.* We first show that in this case,

$$\alpha \leq \sigma_d(W^2) \leq 16\alpha. \tag{83}$$

The first inequality holds due to Lemma 20-(a) which implies that $\sigma_d(W^2) \geq \phi(W^2) \geq \alpha$. The second inequality can be proved via contradiction. Suppose that $\sigma_d(W) \geq 16\alpha$, then again due to Lemma 20-(a), we must have that for any $\gamma \geq \log(d+1)/\alpha$,

$$\phi(W^2) \geq \min\{\sigma_d(W^2), 16\alpha\} - \frac{\log(d+1)}{\gamma}$$

$$\geq 16\alpha - \frac{\log(d+1)}{\gamma} \geq 15\gamma,$$

which contradicts the fact that $\phi(W^2) \leq 2\alpha$. Thus, (83) holds. We next argue that under (83),

$$\langle \nabla_W h(W), \nabla F(W) \rangle \leq 0. \tag{84}$$

Note that

$$\langle \nabla_W h(W), \nabla F(W) \rangle = \sigma'(\phi(W^2) - \alpha)\langle \nabla_W \phi W^2, \nabla F(W) \rangle$$

$$= \frac{\sigma'(\phi(W^2) - \alpha)}{\mathrm{tr}(e^{-\gamma W^2}) + e^{-4\gamma\alpha}} \langle e^{-\gamma W^2} W, \nabla F(W) \rangle,$$

where there the second equality follows from Lemma 20. Next, observe that $\sigma'(\phi(W^2) - \alpha)$ and $\mathrm{tr}(e^{-\gamma W^2}) + e^{-4\gamma\alpha}$ are both non-negative. Thus, to show (84), it suffices to show that $\langle e^{-\gamma W^2} W, \nabla F(W) \rangle \leq 0$. Note that

$$\langle e^{-\gamma W^2} W, \nabla F(W) \rangle = 2\langle e^{-\gamma W^2} W, 2W^3 - MW - WM \rangle$$

$$= 2\text{tr}(e^{-\gamma W^2} W(2W^3 - MW - WM))$$

$$\overset{(i)}{=} 4\Big(\text{tr}(e^{-\gamma W^2} W^4) - \text{tr}(e^{-\gamma W^2} W^2 M)\Big)$$

$$\overset{(ii)}{\leq} 4\Big(\text{tr}(e^{-\gamma W^2} W^4) - \sigma_d(M)\text{tr}(e^{-\gamma W^2} W^2)\Big)$$

$$= 4\Big(\text{tr}(e^{-\gamma W^2} W^4) - 1600\alpha\,\text{tr}(e^{-\gamma W^2} W^2)\Big),$$

where $(i)$ holds because $\text{tr}(AB) = \text{tr}(BA)$ and because the matrices $e^{-\gamma W^2}$ and $W$ commute. The inequality $(ii)$ follows from the fact that for PD matrices $A, B$, we have $\sigma_d(B)\text{tr}(A) \leq \text{tr}(AB) \leq \sigma_d(B)\text{tr}(A)$ [22, Inequality-(1)]. The last line uses the fact that $\alpha = \sigma_d(M)/1600$. For the ease of notation, let $\beta_i$ denote the $i$-th largest singular value of $W$. Since $W$ is symmetric PD, we note that the term in the RHS above can be further simplified as

$$\text{tr}\big(e^{-\gamma W^2} W^4\big) - 1600\alpha\,\text{tr}(e^{-\gamma W^2} W^2) = \sum_{i=1}^{d}\Big(e^{-\gamma\beta_i^2}\beta_i^2(\beta_i^2 - 1600\alpha)\Big)$$

$$\overset{(iii)}{\leq} \sum_{i\in\mathcal{I}}\Big(e^{-\gamma\beta_i^2}\beta_i^2(\beta_i^2 - 1600\alpha)\Big) + e^{-\gamma\beta_d^2}\beta_d^2(\beta_d^2 - 1600\alpha)$$

$$\overset{(iv)}{\leq} \sum_{i\in\mathcal{I}} e^{-\gamma\beta_i^2}\beta_i^4 - 1584 e^{-\gamma\alpha}\alpha^2,$$

where in $(iii)$, the set $\mathcal{I} := \{1 \leq i \leq d-1 \mid \beta_i^2 \geq 1600\alpha\}$ consists of all the indices upto $d-1$ for the corresponding term in the sum is positive. $(iv)$ follows by ignoring negative term and using (83). For the first term in the RHS above, using the fact that for $\beta_i \geq 1600\alpha$ and $\gamma \geq \log(d)/\alpha$, we have

$$e^{-\gamma\beta_i^2}\beta_i^4 \leq e^{-800\gamma\alpha}\alpha^2$$

which implies that

$$\text{tr}(e^{-\gamma W^2} W^4) - 1600\alpha\,\text{tr}(e^{-\gamma W^2} W^2) \leq (d-1)\alpha^2 e^{-800\gamma\alpha} - 1584 e^{-\gamma\alpha}\alpha^2 \leq 0,$$

where the last inequality holds for any $\gamma \geq \log(d)/\alpha$.

Combining all the above bounds implies that $\langle\nabla h(W), \nabla F(W)\rangle \leq 0$, and thus (82) reduces to showing that $\|\nabla F(W)^2\| \geq 16\alpha h(W) F(W)$, which holds due to Lemma 18-(b) and because $h(W) \leq 1$.

$\square$

### D.3.2 Potential function and self-bounding regularity conditions

We first establish the self-bounding regularity conditions for $F$.

**Lemma 23.** *For any symmetric and positive definite $W$, the function $F$ given in* (77) *satisfies*

$$\|\nabla F(W)\| \leq \|\nabla F(W)\|_F \leq 2F(W)^{3/4} + 2\sqrt{\|M\|F(W)},$$

*and*

$$\|\nabla^2 F(W)\| \leq 6\sqrt{F(W)} + 8\|M\|.$$

**Proof of Lemma 23.** Since $\nabla F(W) = (W^2 - M)W + W(W^2 - M)$, we have

$$\|\nabla F(W)\|_F^2 \leq 2\|(W^2 - M)W\|_F^2 + 2\|W(W^2 - M)\|_F^2$$

$$\leq 4\sigma_{\max}(W)^2\|W^2 - M\|_F^2$$

$$\leq 4\sigma_{\max}(W^2)F(W),$$

where the last line holds because $W$ is symmetric and positive definite which implies that $\sigma_{\max}(W)^2 = \sigma_{\max}(W^2)$, and from the definition of $F(W)$. Using the fact that

$$\sigma_{\max}(W^2) \leq \sigma_{\max}(W^2 - M) + \sigma_{\max}(M) \leq \|W^2 - M\|_F + \|M\| = \sqrt{F(W)} + \|M\|,$$

we get

$$\|\nabla F(W)\|_F^2 \leq 4F(W)^{3/2} + 4\sigma_{\max}(M)F(W),$$

which implies that

$$\|\nabla F(W)\|_F \leq 2F(W)^{3/4} + 2\sqrt{\|M\|F(W)}.$$

For the bound on $\|\nabla^2 F(W)\|$, note that using Chain rule and Triangle inequality, we have

$$\|\nabla^2 F(W)\| \leq 6\|W^2\| + 2\|M\| \leq 6\|W^2 - M\| + 8\|M\| = 6\sqrt{F(W)} + 8\|M\|.$$

$\square$

We define the admissible potential function using Lemma 2. First recall the definition of $h$ that

$$h(W) = \sigma\big(\phi(W^2) - \alpha\big),$$

here $\phi$ is given in (78) and $\sigma$ is given in (80). Next, recall Lemma 21 which shows that for any initial point $W(0) = W_0$ for which $h(W_0) > 0$, the point $w(t)$ on its gradient flow path satisfies

$$F(W(t)) \leq F(W_0)\exp(-16\alpha t) =: R(W_0, t).$$

Clearly, as shown in Lemma 22, the function $R(W, h(W)t)$ is an admissible rate of convergence w.r.t. $F$. We next note that the function $F$ is minimized at the point $W^* = \sqrt{M}$ and establish the following properties:

(a) The function $\Gamma(W) := \int_{t=0}^{\infty} R(W, t)\, dt$ is continuously differentiable, and $\max\{\|\nabla\Gamma(W)\|, \|\nabla^2\Gamma(W)\|\} \leq \lambda(\Gamma(W))$ where $\lambda$ is a positive, monotonically increasing function.

(b) $\max\{\|\nabla h(W)\|, \|\nabla^2 h(W)\|\} \leq \pi(\Gamma(W))$ where $\pi$ is a positive, monotonically increasing function.

(c) $(h(W) - h(W^*))^2 \leq \mu(\Gamma(W))$ where $\mu$ is a positive, monotonically increasing function with the property that $k\mu(z) \leq \mu(kz)$ for any $k \geq 1$.

**Proof of properties (a)-(c) above.**

(a) Note that

$$\Gamma(w) = \int_{t=0}^{\infty} R(w, t)\, dt = \frac{F(W)}{16\alpha}.$$

Thus, following the bound in Lemma 23, we note that

$$\|\nabla\Gamma(W)\| \leq 2\Gamma(W)^{3/4} + 2\sqrt{\|M\|\Gamma(W)},$$

and

$$\|\nabla^2\Gamma(W)\| \leq 6\sqrt{\Gamma(W)} + 8\|M\|.$$

Thus, we can define the function $\lambda$ such that $\lambda(z) = O(z^{3/4} + \|M\| + 1)$, which is clearly positive and monotonically increasing.

(b) From Lemma 20-(d) and (e), we note that

$$\|\nabla h(W)\| \leq \frac{4}{\alpha}\Big(F(W)^{1/4} + \sqrt{\|M\|}\Big).$$

and

$$\|\nabla^2 h(W)\| \leq 16\Big(\frac{2}{\alpha^2} + \frac{1}{\alpha}\Big)\Big(1 + \gamma\|M\| + \gamma\sqrt{F(W)}\Big).$$

Thus, we define the function

$$\pi(z) = \frac{4}{\alpha}\Big((16\alpha z)^{1/4} + \sqrt{\|M\|}\Big) + 16\Big(\frac{2}{\alpha^2} + \frac{1}{\alpha}\Big)\Big(1 + \gamma\|M\| + \gamma\sqrt{16\alpha z}\Big)$$

$$= O\Big(\Big(\frac{1}{\alpha^2} + \frac{1}{\alpha}\Big)\Big(1 + \gamma\|M\| + \gamma\sqrt{16\alpha z}\Big)\Big),$$

where the second line follows from recursive applications of AM-GM inequality. We note that the function $\pi$ above is positive and monotonically increasing.

($c$) From Lemma 20-(c), we note that

$$(h(w) - h(\sqrt{M}))^2 \le \frac{2}{\alpha} F(w) = 32\Gamma(w).$$

Thus, we can define the function $\mu(z) = 32z$ which clearly satisfies the desired properties.

$\square$

Thus, all the required conditions in Lemma 2 are satisfied which implies that the function

$$\Phi(w) = \frac{\Gamma(w)}{h(w)} = \frac{F(w)}{16\alpha\sigma(\phi(W^2) - \alpha)} \tag{85}$$

is an admissible potential function w.r.t. $F$ with $g(z) = z$. Furthermore, following the proof of Lemma 2, we note that the function $\Phi$ satisfies the following self-bounding regularity condition

$$\|\nabla^2 \Phi(w)\| \le \rho(\Phi(w)),$$

where the function $\rho$ is given by

$$\rho(z) = (\lambda(z) + z\pi(z)) \cdot \left(\frac{2}{h(W^*)} + \frac{\mu(z)}{h(W^*)^2}\right) + \left(2\lambda(z) \cdot \pi(z) + z\pi^2(z)\right) \cdot \left(\frac{2}{h(W^*)} + \frac{\mu(z)}{h(W^*)^2}\right)^2.$$

Using the fact that $\lambda(z) = O(z^{3/4} + \|M\| + 1)$, $\mu(z) = 32z$ and $\pi(z) = O\left(\left(\frac{1}{\alpha^2} + \frac{1}{\alpha}\right)\left(1 + \gamma\|M\| + \gamma\sqrt{16\alpha z}\right)\right)$ in the above, and repeatedly applying AM-GM, we get that

$$\rho(z) = O\left((1 + \gamma)^2(1 + \|M\|)^2\left(\frac{2}{h(W^*)} + \frac{1}{h(W^*)^2}\right)^2(1 + z^4)\right). \tag{86}$$

### D.3.3   GD for matrix square root

In the following, we provide the convergence guarantee for GD algorithm. We first define the respective problem dependent quantities and instantiate Theorem 4 to provide a $O(1/T)$ convergence bound for GD. We then provide a refined analysis which improves this bound to $O(e^{-T})$.

**$O(1/T)$ rate by direct application of Theorem 4.**

- Lemma 2 implies the potential function

$$\Phi_g(w) = \frac{F(w)}{16\alpha\sigma(\phi(W^2) - \alpha)}$$

  with $g(z) = z$. See Appendix D.3.2 for more details.

- Assumption 1 follows from Lemma 23 which implies that

$$\psi(z) = 4z^{3/2} + 4\|M\|z.$$

- Assumption 2 follows from (86) which implies that

$$\rho(z) = O\left((1 + \gamma)^2(1 + \|M\|)^2\left(\frac{2}{h(W^*)} + \frac{1}{h(W^*)^2}\right)^2(1 + z^4)\right)$$
$$= L(1 + z^4),$$

  where we defined $L$ to hide the constants and the problem dependent terms.

- The function $\theta$ is given by $\theta(z) = \int_{y=0}^{z} \frac{1}{\rho(y)} \, dy$.

- The function $\zeta$ is defined such that

$$\zeta^{-1}(z) = \int_{y=0}^{z} \frac{g(y)}{\psi(y)} \, dy = \int_{y=0}^{z} \frac{1}{4\sqrt{y} + 4\|m\|} \, dy.$$

We note that $\frac{\psi(z)}{g(z)} = 4\sqrt{z} + 4\|M\|$ is a monotonically increasing function of $z$. Thus, using Theorem 4, we get that setting $\eta$ appropriately, GD converges at the rate of

$$F(\widehat{w}_T) \le \frac{2\theta(\Phi_g(w_0))\psi(\zeta(\Phi_g(w_0)))\rho^2(\Phi_g(w_0))}{g(\zeta(\Phi_g(w_0)))} \cdot \frac{1}{T}$$

$$= \frac{\nu(w_0)}{T}, \tag{87}$$

where the problem dependent constants can be computed by plugging in the definitions provided above, and the function $\nu$ is defined to contain all the problem dependent parameters in the right hand side above.

$O(e^{-T})$ **rate via a refined analysis.** We can further improve over the rate in (87) by a refined analysis for GD. In the following, we will show that GD in fact enjoys a $e^{-O(T-t_0)}$ rate of convergence for GD for all $T \ge t_0$, where $t_0$ depends on $w_0$ and problem dependent parameters specified below.

Before delving into the proof of the above, we first provide the relevant improved version of problem dependent parameters that hold for any $w$ for which $F(w) \le \left(\frac{\sigma_d(M)}{2}\right)^2$:

- We first note that $\sigma\big(\phi(W^2) - \alpha\big) = 1$.
- Thus, Lemma 2 implies the potential function

$$\Phi_g(w) = \frac{F(w)}{16\alpha}$$

  with $g(z) = z$.
- Assumption 1 follows from Lemma 23 which implies that

$$\psi(z) = 8\|M\|z,$$

  since the above bound is only used when $z \le \|W\|/2$.
- Assumption 2 follows from (86) which implies that

$$\rho(z) = O\left((1+\gamma)^3(1+\|M\|)^6\left(\frac{2}{h(W^*)} + \frac{1}{h(W^*)^2}\right)^2\right) =: \bar{L},$$

  since the above bound is only used when $z = \Phi(w) \le 800^2\alpha$.
- The function $\theta$ is given by

$$\theta(z) = \int_{y=0}^{z} \frac{1}{\rho(y)} \, \mathrm{d}y = \frac{z}{\bar{L}} \tag{88}$$

We are now ready to provide the improved convergence rate for GD. Note that using (87), we have that there exists some

$$t_0 \le 8\nu(w_0)/\sigma_d(M)^2 \tag{89}$$

such that $F(w_{t_0}) \le \sigma_d(M)^2/8$. Using Lemma 20-(f), the above implies that $h(w_{t_0}) = 1$. In the following, we will show via induction that $F(w_t) \le \sigma_d(M)^2/8$ and $h(w_t) = 1$ for all $t \ge t_0$. The base case with $t = t_0$ is shown above. For the induction step, consider any $t \ge t_0$ and assume that $F(w_t) \le \sigma_d(M)^2/8$ and $h(w_t) = 1$; we will show that the same holds for $w_{t+1}$. Starting from (36) in the proof of Theorem 4, we note that

$$\theta(\Phi(w_{t+1})) \le \theta(\Phi(w_t)) - \frac{\eta}{2\rho(\Phi(w_0))}g(F(w_t)).$$

However, note that $w_t$ satisfies $F(w_t) \le \sigma_d(M)^2/8$ and $h(w_t) = 1$. Since, each update of GD is of magnitude at most $\eta$, we also have that $F(w_{t+1}) \le \sigma_d(M)^2/4$ and thus $h(w_{t+1}) = 1$. Thus, plugging the forms of $\theta, \rho, \Phi$ and $g$ from (89), we get that

$$F(w_{t+1}) \le F(w_t)\left(1 - \frac{8\alpha\eta}{\bar{L}}\right). \tag{90}$$

The above clearly implies that $F(w_{t+1}) \le F(w_t) \le \sigma_d(M)^2/8$ and thus $h(w_{t+1}) = 1$. This completes the induction step.

Now, in order to complete the proof of convergence, note that (90) will hold for all $t \ge t_0$, recursing which implies that

$$F(w_t) \le F(w_{t_0})\left(1 - \frac{8\alpha\eta}{\bar{L}}\right)^{t-t_0} \le F(w_{t_0})e^{-\frac{8\alpha\eta(t-t_0)}{\bar{L}}} \le \sigma_d(M)^2/8e^{-\frac{8\alpha\eta(t-t_0)}{\bar{L}}}.$$

#### D.3.4 SGD for matrix square root

We build on the problem dependent quantities introduced in Appendix D.3.3. Suppose SGD is run with stochastic gradient estimates that satisfy Assumption 3 with $\chi(z) = \sigma^2$. Such a bound is satisfied in the classical stochastic optimization setting in which $\nabla f_{\mathrm{ms}}(w;z) = 2(W^2 - M)W + 2W(W^2 - M) + \varepsilon_t$ where $\varepsilon_t$ is a sub-Gaussian random variable with mean 0 and variance $\sigma^2$. Using the above, we define the function $\Lambda$ used in Theorem 5 as

$$\Lambda(z) = 12\sqrt{z} + 8\|M\| + \sigma^2.$$

Fix any $\bar{w}$ such that $\Phi(\bar{w}) \geq \Phi(w_0)$ and define $B = \Lambda(\zeta(\Phi(\bar{w})))$. Thus, Theorem 5 (in particular the bound in Remark 4) implies that with probability at least $0.7$, the point $\widehat{w}_T$ returned by SGD algorithm satisfies for any $\kappa > 1$,

$$g(F(\widehat{w}_T)) \lesssim \rho(\Phi(\bar{w})) \cdot \frac{\Phi(\bar{w})}{\Phi(\bar{w}) - \Phi(w_0)} \cdot \sqrt{B\theta(\Phi(\bar{w}))} \cdot \frac{1}{\sqrt{T}}.$$

Since $g(z) = z$, the above immediately implies a bound on $F(\widehat{w}_T)$.

# E Additional examples

## E.1 Kurdyka-Łojasiewicz (KŁ) functions

Kurdyka-Łojasiewicz (KŁ) functions appear in various non-convex learning settings, for instance, generalized linear models [44], low-rank matrix recovery [8], over parameterized neural networks [61, 3], reinforcement learning [2, 43, 60] and optimal control [9, 23]. We recall the following definition of KŁ functions, where we assumed that $F_{\mathrm{kl}}$ is non-negative and $\min_w F_{\mathrm{kl}}(w) = 0$.[7]

**Definition 6** (KŁ functions). *The objective $F_{\mathrm{kl}}$ satisfies Kurdyka-Łojasiewicz (KŁ) property with exponent $\theta \in (0,1)$ and coefficient $\alpha \in \mathbb{R}^+$, if for any point $w$,*

$$\|\nabla F_{\mathrm{kl}}(w)\|^2 \geq \alpha F_{\mathrm{kl}}(w)^{1+\theta}.$$

Note that the above KŁ property generalizes the PŁ property we considered in earlier sections; setting $\theta = 0$ results in PŁ property. We note the following rate of convergence for gradient flow for KŁ functions.

**Lemma 24.** *For any initial point point $w(0) = w_0$, the point $w(t)$ on its gradient flow path satisfies*

$$F_{\mathrm{kl}}(w(t)) \leq R_{\mathrm{kl}}(w_0, t) := \frac{F_{\mathrm{kl}}(w_0)}{(1 + \alpha\theta F_{\mathrm{kl}}(w_0)^\theta \cdot t)^{1/\theta}}.$$

*Furthermore, $R_{\mathrm{kl}}$ is an admissible rate of convergence w.r.t. $F$.*

Plugging the above rate function in Theorem 2 with $g(z) = \alpha z^{1+\theta}$ implies that the function $\Phi_g(w) = F_{\mathrm{kl}}(w)$ is an admissible potential function w.r.t. $F_{\mathrm{kl}}$. We can thus use this potential function in Theorem 4 and Theorem 5 to provide a convergence guarantee for GD and SGD. We note that the following additional assumption that $F_{\mathrm{kl}}$ is $H$-smooth, is sufficient to derive the required self-bounding regularity conditions on $F_{\mathrm{kl}}$ and $\Phi_g$.

**Assumption 4.** *There exists an $H \in \mathbb{R}^+$ such that $\|\nabla^2 F_{\mathrm{kl}}(w)\| \leq H$ for any $w$.*

We now state the convergence bound for GD and SGD algorithm.

**Theorem 8.** *Suppose $F_{\mathrm{kl}}$ is K with exponent $\theta$ and coefficient $\alpha$, and satisfies Assumption 4. Then, for any initial point $w_0$ and $T \geq 1$, setting $\eta$ appropriately,*

(a) *The point $\widehat{w}_T$ returned by GD algorithm satisfies $F_{\mathrm{kl}}(\widehat{w}_T) \lesssim \left(\frac{HF_{\mathrm{kl}}(w_0)}{\alpha}\right)^{1/1+\theta} \cdot \frac{1}{T^{1/(2+2\theta)}}$.*

(b) *The point $\widehat{w}_T$ returned by SGD starting from $w_0$ and using stochastic gradient estimates for which Assumption 3 holds with $\chi(z) = \sigma^2$, satisfies $F_{\mathrm{kl}}(\widehat{w}_T) \lesssim \left(\frac{BH^3 F_{\mathrm{kl}}(w_0)}{\alpha^2 T}\right)^{1/2+2\theta}$ with probability at least $0.7$.*

---

[7]Various other definitions KŁ functions appear in the literature. However all of them are equivalent under the appropriate change of variables.

We first observe that both GD and SGD converge at the rate of at least $O(1/T^{1/2+2\theta})$. Furthermore, $\theta = 0$ corresponds to the function being PŁ , in which case, we can improve the rate for GD (by extending Lemma 6) to be of the form $F_{\mathrm{kl}}(w(t)) \leq F_{\mathrm{kl}}(w_0)e^{-O(t)}$ which recovers the bound in Proposition 1. We also note that the classical stochastic optimization setting in which $\nabla f_{\mathrm{kl}}(w; z) = \nabla F_{\mathrm{kl}}(w) + \varepsilon_t$ where $\varepsilon_t$ is a sub-Gaussian random variable with mean 0 and variance $\sigma^2$ satisfies Assumption 3. As a result we have convergence guarantees for SGD algorithm for this case. Finally, we note that similar to the results in Section 3.1, we have the following geometric equivalence between KŁ functions and rates for GF.

**Proposition 3.** *The following two properties are equivalent for any function $F$:*

(a) *For any $w(0) \in \mathbb{R}^d$ and $t \geq 0$, GF has the admissible rate $F(w(t)) \leq \frac{F(w_0)}{(1+\alpha\theta F(w_0)^\theta \cdot t)^{1/\theta}}$,*

(b) *$F(w)$ satisfies the Kurdyka-Łojasiewicz (PL) property i.e. $\alpha F_{\mathrm{kl}}(w)^{1+\theta} \leq \|\nabla F_{\mathrm{kl}}(w)\|^2$,*

*for any $\alpha \geq 0$ and $\theta \in (0,1)$.*

In the following, we will provide convergence guarantees for KŁ functions that are $H$-smooth (c.f. Assumption 4).

### E.1.1  Rate of convergence for gradient flow

The next lemma provides an admissible rate of convergence for KŁ function.

**Lemma 25.** *Suppose $F$ is KŁ with exponent $\gamma \in (0,1/2)$ (Definition 6). Then, for any initial point point $w(0) = w_0$, the point $w(t)$ on its gradient flow path satisfies*

$$F(w(t)) \leq R(w_0, t) := \frac{F(w_0)}{\left(1 + \alpha\theta F(w_0)^\theta \cdot t\right)^{1/\theta}}.$$

*Furthermore, $R$ is an admissible rate of convergence w.r.t. $F$.*

**Proof of Lemma 25.** Note that

$$\begin{aligned}
\frac{\mathrm{d}F(w(t))}{\mathrm{d}t} &= \langle \nabla F(w(t)), \frac{\mathrm{d}w(t)}{\mathrm{d}t} \rangle \\
&= -\|\nabla F(w(t))\|^2 \\
&\leq -\alpha F(w(t))^{1+\theta}.
\end{aligned}$$

Rearranging the terms above implies the differential equation

$$\frac{\mathrm{d}F(w(t))}{F(w(t))^{1+\theta}} \leq -\alpha\,\mathrm{d}t,$$

solving which for $\theta \in (0,1)$ gives the bound

$$F(w(t)) \leq \frac{F(w(0))}{\left(1 + \alpha\theta t \cdot F(w(0))^\theta\right)^{1/\theta}}$$

The desired statement following by plugging in $w(0) = w_0$ and defining

$$R(w,t) := \frac{F(w)}{\left(1 + \alpha\theta t \cdot F(w)^\theta\right)^{1/\theta}}$$

We next show that the above function $R$ is an admissible rate of convergence w.r.t. $F$. Recall that a sufficient conditions for admissibility of $R$ is that for any point $w$,

$$\int_{t=0}^\infty \left(\frac{\partial R(w,t)}{\partial t} + \langle \nabla R(w,t), \nabla F(w) \rangle \right) \mathrm{d}t \geq 0. \tag{91}$$

Note that

$$\int_{t=0}^\infty \frac{\partial R(w,t)}{\partial t}\,\mathrm{d}t = -F(w),$$

and

$$\int_{t=0}^{\infty} \langle \nabla R(w,t), \nabla F(w) \rangle = \|\nabla F(w)\|^2 \int_{t=0}^{\infty} \frac{1}{(1 + \alpha\theta t \cdot F(w)^\theta)^{\frac{1}{\theta}}} \, dt$$
$$- \alpha\theta F(w)^\theta \|\nabla F(w)\|^2 \int_{t=0}^{\infty} \frac{t}{(1 + \alpha\theta t \cdot F(w)^\theta)^{1+\frac{1}{\theta}}} \, dt$$
$$= \frac{\|\nabla F(w)\|^2}{(1-\theta)\alpha F(w)^\theta} - \frac{\theta\|\nabla F(w)\|^2}{(1-\theta)\alpha F(w)^\theta}$$
$$= \frac{\|\nabla F(w)\|^2}{\alpha F(w)^\theta}$$

Combining the two bounds together implies that a sufficient condition for $R$ to be admissible is that

$$\frac{\|\nabla F(w)\|^2}{\alpha F(w)^\theta} \geq F(w).$$

Since $F$ is KŁ with exponent $\theta$ and coefficient $\alpha$, the above holds true for any $w$, thus implying that $R$ is an admissible rate function. $\qquad\square$

### E.1.2 Potential function and self-bounding regularity conditions

Consider the function

$$R(w,t) := \frac{F(w)}{(1 + \alpha\theta t \cdot F(w)^\theta)^{1/\theta}}$$

Lemma 25 implies that $R$ is an admissible rate of convergence for any KŁ objective function $F$. Thus, using Theorem 2 with $g(z) = \alpha z^{1+\theta}$, we get that the function $\Phi_g$ constructed in the following is an admissible potential function for $F$,

$$\Phi_g(w) = \int_{t=0}^{\infty} g(R(w,t)) \, dt$$
$$= \alpha \int_{t=0}^{\infty} \frac{F(w)^{1+\theta}}{(1 + \alpha\theta t \cdot F(w)^\theta)^{\frac{1}{\theta}+1}} \, dt$$
$$= F(w). \tag{92}$$

Note that we already assumed self-bounding regularity conditions on $F$ in Assumption 4. In the following, we derive self-bounding regularity conditions for the potential $\Phi_g$.

**Lemma 26.** *Suppose that $F$ satisfies Assumption 4. Then, for any point $w$, the potential function $\Phi_g$ in (92) satisfies that*

$$\|\nabla^2 \Phi_g(w)\| \leq \psi(\Phi_g(w)),$$

*where $\psi$ is the positive, monotonically increasing function given in Assumption 4.*

**Proof.** From the definition of $\Phi_g$, we have that $\|\nabla^2 \Phi_g(w)\| = \|\nabla^2 \Phi_g(w)\|$. The desired self-bounding regularity conditions on $\Phi_g$ thus follows from Assumption 4. $\qquad\square$

We next prove Proposition 3.

**Proof of Proposition 3.** The proof of $(b) \Rightarrow (a)$ follows from Lemma 25. For the proof of $(a) \Rightarrow (b)$, we note that plugging the given rate in Theorem 2, we get that the function $\Phi_g(w) = F(w)$ is an admissible potential function w.r.t. $F(w)$ with $g(z) = \alpha z^{1+\theta}$. Thus, from (4), we get that

$$\|\nabla F(w)\|^2 = \langle \nabla \Phi_g(w), F(w) \rangle \geq g(F(w)) = \alpha F(w)^{1+\theta},$$

which implies the desired PŁ property for $F$. $\qquad\square$

### E.1.3 GD for KŁ functions

In the following, we provide the respective problem dependent quantities and instantiate Theorem 4 to provide a convergence bound for GD for KŁ functions.

- We set

$$g(z) = \alpha z^{1+\theta}.$$

- Assumption 1 follows from Lemma 7 and Assumption 4 which implies that

$$\psi(z) = 4Hz.$$

- Assumption 2 follows from Assumption 4 which implies that

$$\rho(z) = H.$$

- The function $\theta$ is given by

$$\theta(z) = \int_{y=0}^{z} \frac{1}{\rho(y)} \, dy = \frac{z}{H}.$$

- The function $\zeta$ is defined such that

$$\zeta^{-1}(z) = \int_{y=0}^{z} \frac{g(y)}{\psi(y)} \, dy = \int_{y=0}^{z} \frac{\alpha y^{\theta}}{4H} \, dy = \frac{\alpha}{4H(1+\theta)} z^{1+\theta},$$

which implies that

$$\zeta(z) = \left( \frac{4H(1+\theta)z}{\alpha} \right)^{1/1+\theta}.$$

Plugging the above problem-dependent constants in Theorem 4 (under the case that $\Phi_g = F$) implies that setting

$$\eta = \sqrt{\frac{\theta(F(w_0))}{\psi(F(w_0))} \cdot \frac{1}{T}} \leq \frac{1}{2H\sqrt{T}},$$

GD has the rate

$$g(F(\widehat{w}_T)) \leq 4\rho(\Phi_g(w_0))\sqrt{\theta(\Phi_g(w_0))\psi(\zeta(\Phi_g(w_0)))} \cdot \frac{1}{\sqrt{T}}.$$

$$\leq \frac{8HF(w)}{\sqrt{T}},$$

Plugging $g(z) = \alpha z^{1+\theta}$ in the above implies that

$$F(\widehat{w}_T) \leq \left( \frac{4HF(w)}{\alpha} \right)^{\frac{1}{1+\theta}} \cdot \frac{1}{T^{1/(2+2\theta)}}.$$

Clearly, the function $\frac{\psi(z)}{g(z)} = \frac{4H}{\alpha z^{\theta}}$ is not a monotonically increasing function of $z$, and thus the improved analysis for GD does not extend to this case.

### E.1.4 SGD for KŁ functions

Suppose Assumption 3 is satisfies with $\chi(z) = \sigma^2$. In addition to the problem dependent quantities in Appendix E.1.3, we define the function $\Lambda$ used in Theorem 5 as

$$\Lambda(z) = 4Hz + 2\sigma^2.$$

Fix any $\bar{w}$ such that $2F(w_0) \leq F(\bar{w}) \leq 4F(w_0)$ and define $B = 16\left( H^{2+\theta} \cdot \frac{F(\bar{w})}{\alpha} \right)^{1/1+\theta} + 2\sigma^2$. Following Theorem 5 (in particular the bound in Remark 4), we note that for any

$$\eta \leq \frac{1}{20\log^2(20T)} \cdot \frac{F(\bar{w}) - F(w_0)}{\sqrt{BHF(\bar{w})T}},$$

the point returned by SGD algorithm after $T$ iterations satisfies with probability at least 0.7,

$$g(F(\widehat{w}_T)) \lesssim H\sqrt{BHF(w_0)} \cdot \frac{1}{\sqrt{T}}.$$

which implies that

$$F(\widehat{w}_T) \lesssim \left(\frac{BH^3F(w_0)}{\alpha^2 T}\right)^{1/2+2\theta}.$$

## E.2 Extending Chatterjee 2022 [13]

If the objective $F$ is such that some potential $\Phi_g$ satisfies the geometric condition in (4) for every $w$, then we have a rate of convergence for GF (Theorem 1). As we saw earlier, for instance, using this machinery one can obtain rates for GF/GD/SGD when $F$ has PŁ property everywhere. However, such global properties, that (4) holds for every $w$ are often too stringent to hold in practice. In order to go beyond global assumption, in Lemma 2 we showed how to extend our tools (by defining corresponding admissible potentials) when such properties (and thus rates for GF) only hold in some region. Convergence under such local properties has also been considered before in other works [16, 20, 30, 45, 54, 28, 42]. However, all of these results usually rely on being able to choose an initialization $w_0$ in the good region, where the corresponding local property holds, and is close enough to the global minima that we wish to converge to. This is not always practical, and to circumvent this issue in a recent work of Chatterjee [13], an assumption that is "local" w.r.t. initial point is provided under which one can show that GF and GD starting from this initialization is guaranteed to converge (at an exponential rate). The interesting property of this condition is that it is local to initial point $w_0$ considered and does not make any global assumption on the objective.

Using the tools in this paper, this type of local property can be easily extended to more general properties than what was considered in Chatterjee [13]. For ease of presentation, we present below the result for $H$-smooth objective $F$ and for GF convergence, the corresponding techniques can be easily extended show GD/SGD convergence when Assumption 1 holds. Given a function $r : \mathbb{R}^d \mapsto \mathbb{R}^+$ and a monotonically increasing positive function $g$, define

$$\alpha_{r,g}(w_0,\kappa) = \inf_{w:\|w-w_0\|_2 \le \kappa, F(w) \neq 0} \frac{\nabla r(w)^\top \nabla F(w)}{g(F(w))} \tag{93}$$

Our main assumption on the initial point $w_0$ is that for some $\kappa > 0$ and some functions $R$ and $g$,

$$\int_0^\infty \sqrt{g^{-1}\left(\frac{r(w_0)}{t\alpha_{r,g}(w_0,\kappa)}\right)} dt \le \frac{\kappa}{H} \tag{94}$$

The next lemma shows that for any initial point $w_0$ that satisfies the local condition above, one has a rate of convergence for GF starting from $w_0$.

**Lemma 27.** *Suppose $w_0$ satisfies (94) for some functions $R$ and $g$, and radius $\kappa = \kappa_0 > 0$. Then, gradient flow starting from $w(0) = w_0$ satisfies for any $t \ge 0$,*

$$F(w(t)) \le g^{-1}\left(\frac{r(w_0)}{T\alpha(w_0,\kappa_0)}\right).$$

To obtain nearly matching rates for the type of condition in Chatterjee [13], one can choose $r(w) = p \cdot F(w)^{1/p}$ and $g(z) = z^{1/p}$. Since $p$ is arbitrary, setting $p = T\alpha(w_0,\kappa_0)/e$ we obtain nearly the same rate and the local condition as Chatterjee [13] (upto constants). The interesting part though, is that this is for only one choice of $g$ and $r$, whereas we can get the convergence for GF when the condition holds for any $g, r$. In Chatterjee [13], examples of overparmeterized deep neural nets are shown to satisfy the assumption (for the specific $r$ and $g$ above). With a wider choice of $g$ and $r$ we can extend these to more general models (eg. neural networks with milder assumptions on the activation function).

### E.2.1 Proofs

Given a function $r : \mathbb{R}^d \mapsto \mathbb{R}^+$ and a monotonically increasing, positive function $g$, define

$$\alpha_{r,g}(w_0,\kappa) = \inf_{w:\|w-w_0\|_2 \le \kappa, F(w) \neq 0} \frac{\nabla r(w)^\top \nabla F(w)}{g(F(w))}$$

Our main assumption on the initial point $w_0$ is that for some $\kappa > 0$ and some functions $R$ and $g$,

$$\int_0^\infty \sqrt{g^{-1}\left(\frac{r(w_0)}{t\alpha_{r,g}(w_0,\kappa)}\right)}\,dt < \frac{\kappa}{\sqrt{2H}}$$

The next lemma shows that for any initial point $w_0$ that satisfies the local condition above, one has a rate of convergence for GF starting from $w_0$.

**Lemma 28.** *Suppose $w_0$ satisfies* (94) *for some functions $R$ and $g$, and radius $\kappa = \kappa_0 > 0$. Then, gradient flow starting from $w(0) = w_0$ satisfies for any $t \geq 0$,*

$$F(w(t)) \leq g^{-1}\left(\frac{r(w_0)}{t\alpha(w_0,\kappa_0)}\right).$$

**Proof of Lemma 28.** From our assumption, let $R$ $g$ and $\kappa > 0$ be given such that

$$\int_0^\infty \sqrt{g^{-1}\left(\frac{r(w_0)}{t\alpha_{r,g}(w_0,\kappa)}\right)}\,dt < \frac{\kappa}{\sqrt{2H}}$$

First note that by the definition of $\alpha_{r,g}(w_0,\kappa)$, we have that for any point $w$ such that $\|w - w_0\|_2 \leq \kappa$,

$$g(F(w)) \leq \frac{\nabla r(w)^\top \nabla F(w)}{\alpha_{r,g}(w_0,\kappa)}$$

This implies that if we take $\Phi(w) = \frac{r(w)}{\alpha_{r,g}(w_0,\kappa)}$ as a potential, then for every point $w$ that is within distance $\kappa$ from $w_0$, $\Phi$ satisfies property (4) w.r.t. $g$ for any point that is within distance $\kappa$ from $w_0$. Now consider the gradient flow path starting at $w_0$ and let $t_0$ be the first time the gradient flow path reaches a distance of $\kappa$ from $w_0$. Till this time, we can apply Theorem 1 and conclude that for any $t < t_0$,

$$g(F(w(t))) \leq \frac{\Phi(w_0)}{t} = \frac{r(w_0)}{t\alpha(w_0,\kappa_0)}$$

Next, we will argue that $t_0 = \infty$. To this end, note that

$$\begin{aligned}
\|w(t_0) - w(0)\|_2 &= \left\|\int_0^{t_0} \nabla F(w(t))dt\right\|_2 \\
&\leq \int_0^{t_0} \|\nabla F(w(t))\|_2\,dt \\
&\leq \int_0^{t_0} \sqrt{2HF(w(t))}dt \\
&\leq \sqrt{2H}\int_0^{t_0} \sqrt{g^{-1}\left(\frac{r(w_0)}{t\alpha(w_0,\kappa_0)}\right)}dt
\end{aligned}$$

Note note that since $t_0$ is the first time we reach distance $\kappa$ from $w_0$, till that point, we have that the entire GF path is within the $\kappa$ radius from $w_0$ and hence, from our condition, $\int_0^\infty \sqrt{g^{-1}\left(\frac{r(w_0)}{t\alpha_{r,g}(w_0,\kappa)}\right)}\,dt < \frac{\kappa}{\sqrt{2H}}$. USing this above, we conclude that

$$\|w(t_0) - w(0)\|_2 \leq \sqrt{2H}\int_0^{t_0} \sqrt{g^{-1}\left(\frac{r(w_0)}{t\alpha(w_0,\kappa_0)}\right)} < \kappa$$

But this is a contradiction since at $t_0$, the distance to $w_0$ should be $\kappa$ by definition of $t_0$. But we have shown that the distance is strictly smaller than $\kappa$. Hence we can conclude that $t_0 = \infty$. Hence we can conclude that for any $t > 0$ in fact,

$$F(w(t)) \leq g^{-1}\left(\frac{r(w_0)}{t\alpha(w_0,\kappa_0)}\right)$$

$\square$