# OpenReview forum: "From Gradient Flow on Population Loss to Learning with Stochastic Gradient Descent"
_NeurIPS.cc/2022/Conference — NeurIPS 2022 Accept_

### Official Review · Reviewer_T8uY · 2022-07-01

**Rating:** 6
**Confidence:** 4
**Soundness:** 3 good
**Presentation:** 3 good
**Contribution:** 3 good

**Summary:**

This paper aims at providing conditions for the convergence of GD/SGD schemes when there is a rate of convergence for Gradient Flow on population loss, and this is done by deriving a potential function and checking the geometric property. Also this potential should satisfy certain properties, called self-bounding regularity conditions, in order to give us the desired convergence for GD/SGD methods. Authors verify that this is indeed applicable not only to convex and PL functions, but also to non-convex problems.

**Questions:**

grammatical and mathematical typos:

line 108 derivative. line 112 extra "in". line 121 shown. line 144 such that. line 215, $F(w(t))\leq\frac{\lVert w_0 \rVert^2}{2t}$. line 226, satisfied. lines 587, 592, it should be $\Phi(w_0)$. line 611, the negative signs in lines 3-4-5 should be removed, as $|r'(z)|=-r'(z)$. line 625, extra "from", also $R(w,t)=F(w)e^{-\lambda t}$. line 630, $\frac{\lVert \nabla F(w) \rVert ^2}{\lambda}$. line 634, 3rd line, missing "=". line 643, it should be $\lVert u \rVert_{3/2}$. line 651, $F(w(T)) - F^*\leq \frac{\Phi(w_0)}{T}\leq \frac{\lVert w_0 \rVert ^2}{2T}$. line 657, "many ensure". line 662, proof. line 700, extra "defined". line 705, $\theta''(z)=\frac{-\rho'(z)}{\rho(z)^2}$. line 712, $u=-\frac{\nabla \Phi(w)}{\rho(\Phi(w))}$. line 732, since "$\theta$" is "monotonic", this implies that "$\Phi(w_{t+1})\leq \Phi(w_t)$".

in line 752 equation (33), 2nd line is wrong. we don't know if $F(w_t)\leq F(w_0)$. Instead you should use $F(w_t) \leq \zeta(\Phi(w_t)) \leq \zeta(\Phi(w_0))$. However final conclusion of equation (34) is correct.

line 772, $\psi(F(w))$ instead of $\lVert \nabla F(w) \rVert^2$. line 783, $\kappa > 1 $. line 792, 3rd line is duplicate. line 796 and afterward, you should replace $\Phi(w_0)$ with $\theta(\Phi(w_0))$. line 809 $"M="$. line 810, $\text{log}^2$ seems to be $\text{log}^3$. line 819, "note that" and "when".  line 826, "$g$" is missing.

In line 947, I couldn't derive second inequality. maybe there should be a larger coefficient.

lines 1018 and 1025, $\lVert M \rVert$ instead of M.



Questions:

regarding line 207-8, can you provide such examples, or mention some papers?

In line 614, $\underset{\epsilon \rightarrow 0}{\text{lim }}  \epsilon ^{-\frac{1}{\epsilon}}$ seems to be infinity, but for your result to hold it should be zero. This also happens in other places like line 181. Please explain.

My most important question is regarding the exponential rate of GD in theorems 6 and 7. In the proofs you say combining Lemma 1 with Theorem 4 will give the result. However Theorem 4 has a $1/T$ rate and $g(z)=z$. Please carefully explain it and also update the paper to contain more details of this proof.

In line 722, please explain (if needed by writing equation) that why 27 is valid if 26 is violated. I think there must be some $"g"$ and $"\psi"$ missing in (27), by looking at line 741.

In Theorem (4) you mention $O(\frac{1}{\sqrt{T}})$ rate of convergence, but in line 741, I can see $1/T$ as well as a constant term. Please explain. it seems that line 741 and 761 only differ by a constant and both of them are $O(1/T)$ except that 741 is suboptimal assuming g is zero at optimum.

In line 798, I couldn't use Markov to get (39). I'd like to see one line of proof.

line 804, where do you set $\kappa$ in the paper?

Line 853 there is a serious issue with derivation of line 5, because we don't know $\theta(\Phi(w_{t+1})) - \theta(\Phi(w_t))\geq 0$, hence equation (50) may not be true. Please revise/explain the proof.

In proof of Lemma 11 needs serious revision. The derivation in lines 890 is incorrect and you should write $\overset{-}{R}(w(s),t)=R(w(s),\sigma(h(w(s)))t)\leq R(w,\sigma(h(w(s)))t +s) \leq R(w,\sigma(h(w(s)))t + \sigma(h(w(s)))s)\leq=R(w,\sigma(h(w(s))) (t+s) )\leq R(w, h(w(s)) (t+s) )= \overset{-}{R}(w,s+t)$.
More importantly, I see an inconsistency in using $\sigma(h(w))$ and $h(w)$. For example you define $\overset{-}{R}(w,t)=R(w,\sigma(h(w))t)$, but after line 892 suddenly there is no sign of $\sigma$ in derivations of $\nabla \Phi_g(w)$ and $\nabla ^2 \Phi_g(w)$. As you use $\sigma$ in lines 306 and 336, I think $\sigma$ is kinda absored in $h$ but I can't wrap my head around this issue. in line 895 equation (51) $\Phi_g(w)$ uses $h(w)$ instead of $\sigma(h(w))$. Also in line 911 the second term is used incorrectly and should be modified.

In line 1028, how do you calculate the bound on hessian? I assume hessian would be complicated.

in line 1048 line 3, do you assume $\text{min }F=0$?

Suggestion:
I urge you to state the result of Chatterjee more rigorously in the Appendix, and explain (mathematically in appendix) how in the lines 374-378 you can recover his result. I also expected a conclusion part at the end of the paper, but I assume you couldn't fit it.

**Limitations:**

As this is a completely theoretic paper, I cannot think of anything related to social impact.

**Strengths And Weaknesses:**

As authors mention, using admissible potential functions for getting a convergence rate for GF is studied before. Identifying the one-to-one correspondence between admissible rate and potential functions that satisfy the geometric property in (4) seems interesting. Propositions 1 and 2 also provide simple examples on how admissible rates and potentials exist in easy-to-prove scenarios.
As I read all of the proofs, there are lots of typos and some derivations, which I will mention in below, are not carried out correctly, and I hope after revision they will not be problematic for the main claims to hold. However, overall the main results are clear and easy to read and providing examples like propositions 1,2 and touching on Phase retrieval and matrix square root problems makes this converse Lyapunov strategy interesting for analysis of GD/SGD using the GF convergence rate. I think this paper aims at connecting GF and GD analysis which is currently a hot research topic in optimization.

---

> ### Author Response · Authors · 2022-08-02
> **Thank you for the feedback, and response to your concerns!**
>
> Thank you for a careful read. We have incorporated your feedback in the rebuttal revision of the paper (uploaded on openreview). Please find answers to your concerns below. We would be happy to follow up more through the openreview portal on any of these if the reviewer wishes.
>
> > regarding lines 207-8, can you provide such examples, or mention some papers?
>
> We provide an example of one such loss function in Theorem 3. Note that for the considered problem, gradient flow converges at a rate of $\frac{1}{T}$ by gradient descent fails to converge; hence gradient flow and gradient descent must take different paths. Furthermore, reference [14] (and the related work section within it) provides more discussion on path-wise convergence of gradient flow and gradient descent. We will add a pointer to these results in the final version of the paper.
>
> > In line 614, $lim_{\epsilon \rightarrow 0} \epsilon^{-1/\epsilon}$ seems to be infinity, but for your result to hold it should be zero. This also happens in other places
>
>  This was a technical mistake, but can be easily corrected to give a similar result. The key idea in the fix is to use a different choice of $g$, specifically, we choose $g(z) = z/\log^2(1/z)$ to obtain a similar result. Please take a look at Appendix B in the rebuttal revision for more details.
>
> > Exponential rate of GD in theorems 6 and 7.
>
> We apologize for the confusion. A direct invocation of Theorem 4 indeed implies a $1/T$ rate of convergence for gradient descent for phase retrieval and matrix square root. However, a simple refinement of the analysis improves this to a $e^{-T}$ rate of convergence. The key idea for this improvement is to relate $\Phi(w_t)$ with $F(w_t)$, and show that once $F(w_t)$ is smaller than a particular threshold, the GD analysis in the proof of Theorem 4 ensures that $F(w_t)$ keeps on decreasing further at a linear rate. Please take a look at Sections D.2.3 and D.4.3 (in the rebuttal revision) for details on this refined analysis. Furthermore, the $e^{-T}$-style bound for GD matches the state-of-the-art bounds for GD (in terms of dependence on $T$) for both phase retrieval and matrix square root. We have also added a discussion on this in the rebuttal revision of the paper.
>
> > In line 722, please explain (if needed by writing equation) why 27 is valid if 26 is violated.
>
> You are correct and this is a typo. There should be a $g$ in the LHS and $\psi$ in the RHS. This has been corrected in the rebuttal version and does not affect the proofs or the theorem statement.
>
> > In Theorem (4) you mention $O(1/T)$ rate of convergence, but in line 741, I can see $1/T$ as well as a constant term
>
> Note that $\eta$ is a free parameter in line 741. Setting $\eta \propto \frac{1}{\sqrt{T}}$ gives the states $\frac{1}{\sqrt{T}}$ rate for GD. Similarly, $\eta$ is a free parameter in line 761, setting which to a constant gives the stated $\frac{1}{T}$ rate.
>
> > In line 798, I couldn't use Markov to get (39)
>
>  The proof has been corrected. This step can be skipped altogether and we can do a direct analysis accumulating the terms $\theta(\Phi(w_t)) - E_t [{\theta(\Phi(w_{t +1}))}]$ till the end, and then using Azuma-Hoeffding's inequality to bound them. Please find detailed proof on page 28 in the rebuttal revision.
>
> > line 804, where do you set κ in the paper?
>
> As mentioned in line 783 (in the original submission), $\kappa$ is a free parameter in Theorem 5. However, to address reviewer's question, we have also specified an example of a meaningful choice of $\kappa$ in Remark 4 in the appendix of the rebuttal revision. In particular, choosing $\kappa = \tfrac{\rho(\Phi(\bar{w}))}{\rho(\Phi(w_0))}$ where $\bar{w}$ is any point for which $\Phi(\bar{w}) \geq \Phi(w)$ suffices. Furthermore, this choice of $\kappa$ also makes all the problem-dependent parameters more interpretable.
>
> > Typo in Line 853 and proof of Lemma 11.
>
> Inequality  (3) needs an absolute value under the square root. The proof is correct as the absolute value appears back in inequality (4). This has been corrected in the rebuttal revision.
>
> > In line 1028, how do you calculate the bound on hessian?
>
>  Please take a look at the attached rebuttal revision for a detailed proof
>
> > in line 1048 line 3, do you assume min F=0?
>
> As mentioned in the first paragraph of our Setup (section 2) , we assume throughout that $min_w F(w) = 0$.
>
> > I urge you to state the result of Chatterjee more rigorously in the Appendix
>
> Thank you very much for the feedback. We will make this section more rigorous and also add a conclusion section to our paper. There are in fact two ways to recover the result. First is using the choice of r and g as mentioned, but alternatively, even if one uses r = F and g(z) = z, then the implied date applied repeat also can recover the exponential rate given in Chaterjee et. al. 2022. We will provide these in detail in the final version of the paper.

---

> > ### Comment · Reviewer_T8uY · 2022-08-05
> > **Comment 1 after rebuttal**
> >
> > Thanks for the answers. There has been a (rather massive) redoing of proofs in appendix (52 pages vs 39 pages in original submission) and I can just check the concerns I had and your answers to them.
> >
> > Regarding C.2 and proof of theorem 4, if you set $\eta$ to be $1/\sqrt{T}$ it means you fix T beforehand and then set $\eta$. it is not constant and not varying step-size. Hence for different T's you need different $eta$'s in the start of GD, and this is not a convergence rate of $1/\sqrt{T}$ for all T. This time-dependent setting for $eta$ also appears in Theorem 5 and hence all the results of GD-SGD in section 5 are affected. Please explain.
> >
> > The added analysis of exponential rate for GD in section 5 results seems interesting.

---

> > > ### Author Response · Authors · 2022-08-05
> > > **Response to Comment 1**
> > >
> > > We thank the reviewer for their time and further comments!
> > >
> > >
> > > > Regarding C.2 and proof of theorem 4, if you set $\eta$ to be $1/\sqrt{T}$ it means you fix T beforehand and then set $\eta$. it is not constant and not varying step-size. Hence for different T's you need different $\eta$'s in the start of GD, and this is not a convergence rate of  $1/\sqrt{T}$ for all T. This time-dependent setting for $\eta$ also appears in Theorem 5 and hence all the results of GD-SGD in section 5 are affected. Please explain.
> > >
> > > Please note that this is a standard practice in optimization. In our results, the choice of $\eta$ depends on the horizon $T$ that we want to run GD / SGD algorithm for, and $\eta$ is kept fixed for all iterations till time $T$. The statement of Theorem 4 and Theorem 5, and the stated results in Section 5, clearly mention that $\eta$ is set appropriately after choosing $T$. In particular, for every choice of $T$, there exists an $\eta$ such that running GD / SGD with step-size $\eta$ for $T$ time obtains the given rate of convergence.
> > >
> > > We also mention that the standard techniques from optimization literature, e.g. doubling trick, can easily break this dependence on knowing $T$ beforehand [Ref 1, below]. Another trick commonly used is to run GD / SGD with the step size $\eta_t = 1/\sqrt{t}$ (instead of $\eta_t = 1/\sqrt{T}$) which obtains identical results (up to constants factors). Since these techniques are well-known, we skipped mentioning this extension in the paper, but we will add a reference to the final version to clarify this.
> > >
> > > [Ref 1] Lecture 11: Adaptation 1, Lecturer:  Anant Sahai/Vidya Muthukumar, EE194/290S: ML for Sequential Decision under Uncertainty, UC Berkeley, link - https://inst.eecs.berkeley.edu/~ee290s/fa18/scribe_notes/EE290S_Lecture_Note_11.pdf

---

> > > > ### Comment · Reviewer_T8uY · 2022-08-06
> > > > **Comment 2**
> > > >
> > > > The explanation for setting $eta$ beforehand is convincing, and thanks for the reference.

---

### Official Review · Reviewer_RAee · 2022-07-10

**Rating:** 6
**Confidence:** 4
**Soundness:** 3 good
**Presentation:** 2 fair
**Contribution:** 3 good

**Summary:**

The authors of the present paper show the following contributions:
- They provide an equivalence between satisfying certain convergence rates and geometric conditions
- They apply their framework to gradient descent and stochastic gradient descent, requiring  more assumptions on the function to be optimised to derive convergence rates.
- They apply their overall framework to matrix square root and phase retrieval where rates of convergence for the gradient flow have been shown in the literature.

**Questions:**

- Theorem 4 shows 1/T or 1/\sqrt{T} rates of convergence for the gradient descent whereas Theorem 6 and 7 show exponential rates if the gradient descent case. I understand some PL inequality seems to be hidden here, but this needs to be precised as these theorems are not an application of the Theorem of the previous section.
- The SGD noise scales like the function value for the phase retrieval example, hence I would expect an exponential rate of convergence for SGD also (Theorem 6). Can the authors comment on this ?
- In terms of the two applications, it would be nice to refer to known rates of convergence for GD and a comparison with what you obtain with your technique.

**Limitations:**

I use this paragraph to conclude as I already discussed the limitations on the two previous boxes. My opinion is that it could be a super nice paper: I like the overall idea to go from gradient flow with rates to geometrical implications on the lost and then to (stochastic) gradient descent rates. However, the implication of such results on concrete examples could be more referenced and the writing improved.

For now, I give a borderline accept but I will gladly improve my score if the authors give me some evidence that they can improve the paper in such a short notice.

**Strengths And Weaknesses:**

### **Strengths**

- The paper appears to be original, introduce a very nice idea from the ODE's literature with converse Lyapunov theorems (ref 25). The link with gradient descent and stochastic gradient descent is nicely introduced and seems to enable direct analysis from rates of convergence for discrete analysis.
- The papers show a large diversity of results, which is rare in nowadays paper in conference.
- The authors really try to apply their machinery to concrete and hard problems in the last section.
- The hypothesis on the stochastic gradient descent (Assumption 3) is a nice and covers a lot of cases.

### **Weaknesses**

- The paper is overall hard to follow: the progression is okay and technical contributions are understandable but it is hard to understand where the authors want to bring us. As an example of this, I think that the paper lacks some comments in the fifth section where they apply their method: there is absolutely no discussion on knowing whether the rates are tight, whether they are new (for gradient and stochastic gradient descent). Another striking example is the subsection 5.3 where Lemma 4 seems to nicely improve on recent Chatterjee condition without application or explication on this result (even the authors write: "*With a wider choice of g and r we can extend these to more general models (eg. neural networks with milder assumptions on the activation function)*".
- Adding to the previous comment, there is no conclusion, and many times, the paper seems to have been written in a hurry, see some minor flaws below.
- For the result to hold, there is a need to know in advance rates of convergence, which limit the applicability of the results.


### **Minor flaws**

- Why talking of implicit regularisation in the introduction (ligns after 24) ?
- In lign 38, the cited papers do not provide convergence analysis and only focus on implicit regularisation
- lign 87, it is not standard that the estimator is not the final iterate, precise that this is your way to see gradient descent. Same lign 96.
- lign 97, $(w)$ is a weird notation for a process. Prefer $w$ or $(w_t)_{t \geqslant 0}$
- lign 108 ,what is the $p$-th derivative ? Isn't it simply the gradient ?
- For all theorem, comment the fact that Admissible rate functions could not exist
- It would be nice to have a commentary on what is $g$ in Corollary 1

---

> ### Author Response · Authors · 2022-08-02
> **Thank you for the feedback, and response to your concerns!**
>
> Thank you for a careful read. We have incorporated your feedback in the rebuttal revision of the paper (uploaded on openreview). Please find answers to your concerns below. We would be happy to follow up more through the openreview portal on any of these if the reviewer wishes.
>
> >The paper is overall hard to follow: the progression is okay and technical contributions are understandable but it is hard to understand where the authors want to bring us.
>
> Thank you for the feedback. We have been working on improving the readability of the paper and the proofs, and the improved draft can be found in the rebuttal revision. We will also add a discussions and conclusions section to the final version of the paper.
>
> >  For the result to hold, there is a need to know in advance rates of convergence, which limit the applicability of the results.
>
> Please note that our key contribution in this paper is precisely on how to use the rate of convergence of gradient flow, to show that SGD learns. In fact, for all of our applications showing the rate of convergence of gradient flow is extremely simple, and the corresponding rate follows from directly solving the associated PDE for gradient flow. For more details, please take a look at Sections D.1.1 and D.3.1 in the rebuttal revision.
>
> > Why talk of implicit regularisation in the introduction (lines after 24)?
>
> We wanted to contrast our work with the implicit regularization literature, which also aims at explaining why SGD learns for complex non-convex problems. Our hypothesis is that SGD learns because gradient flow converges, which contrasts with the implicit regularization conjecture that SGD learns because it finds a solution that corresponds to the regularized empirical risk minimizer with a problem-dependent regularizer. In fact, in all the prior works where SGD learns and is shown to have an implicit regularization,  gradient flow also converges to the global minimizer and the success of SGD can be explained through our framework (by invoking Theorem 2 + 5).
>
>
>
> > Typos in line 38, 97, 108, Corollary 1.
>
> Thank you for pointing out these typos. We have fixed them in the rebuttal revision of the paper.
>
> ### We provide detailed answers to your questions below: ###
>
> > Theorem 4 shows 1/T or 1/\sqrt{T} rates of convergence for the gradient descent whereas Theorem 6 and 7 show exponential rates if the gradient descent case
>
> We apologize for the confusion. A direct invocation of Theorem 4 indeed implies a $1/T$ rate of convergence for gradient descent for phase retrieval and matrix square root. However, a simple refinement of the analysis improves this to a $e^{-T}$ rate of convergence. The key idea for this improvement is to relate $\Phi(w_t)$ with $F(w_t)$, and show that once $F(w_t)$ is smaller than a particular threshold, the GD analysis in the proof of Theorem 4 ensures that $F(w_t)$ keeps on decreasing further at a linear rate. Please take a look at Sections D.1.3 and D.3.3 (in the rebuttal revision of the paper) for details on this refined analysis. Furthermore, the $e^{-T}$-style bound for GD matches the state-of-the-art bounds for GD (in terms of dependence on $T$) for more details.
>
> > -   The SGD noise scales like the function value for the phase retrieval example, hence I would expect an exponential rate of convergence for SGD also (Theorem 6). Can the authors comment on this?
>
> Our focus in the paper was to develop a general framework for showing the convergence of SGD for general non-convex learning problems. Due to the generality of our framework, we were limited to getting a $1/\sqrt{T}$ rate of convergence for SGD (which is optimal under various noise models for phase retrieval). However, we believe that a similar refined analysis to what we did for GD in section D.1.3 by relating back $\Phi(w_t)$ to $F(w_t)$ can be extended to show a $e^{-T}$ rate of convergence for SGD in the noise model that the reviewer is curious about. We will discuss this in the final version of the paper but are happy to follow up more through openreview platform if the reviewer wishes.
>
> > In terms of the two applications, it would be nice to refer to known rates of convergence for GD and a comparison with what you obtain with your technique.
>
> Thank you for pointing this out. The $O(e^{-T})$ rate of convergence for GD matches the state-of-the-art bounds for both phase retrieval and matrix square root (in terms of dependence on $T$).  The rates for GD that we obtained by using Theorem 4 for PL functions and convex functions are also optimal. Furthermore, to the best of our knowledge, ours is the first work to analyze the convergence of SGD under general noise conditions satisfying Assumption 3.  We finally remark that the $1/\sqrt{T}$ rate of SGD for matrix square root problem in the stochastic optimization setting was not known before and can also be easily recovered from our framework. We have added a discussion on this in the rebuttal revision of the paper after Theorem 6 and 7.

---

> > ### Comment · Reviewer_RAee · 2022-08-07
> > **After rebuttal**
> >
> > I thank the reviewers for the detailed answer and the precisions.
> > - I still do not understand the reference to the implicit bias literature and the answer of the authors on this point, but this is not super important.
> > - I understand that the authors present examples where they know rates of convergence for the gradient flow. I was more concerned about applicability for not-known non-convex examples where we know that the gradient flow converges (with no rates) but do not know if GD/SGD does.
> >
> > Overall I am happy to raise my score as this is a very good paper.

---

### Official Review · Reviewer_fx5A · 2022-07-12

**Rating:** 6
**Confidence:** 4
**Soundness:** 4 excellent
**Presentation:** 4 excellent
**Contribution:** 2 fair

**Summary:**

This work presents a novel framework for analyzing SGD via Lyapunov analyses of continuous-time gradient flow. The authors first establish that, loosely speaking, a Lyapunov function must exist if gradient flow converges. Then the authors show convergence of SGD using the same Lyapunov function. Some interesting and illustrative examples are provided, but there are no experimental results, which makes this a theory paper.

**Questions:**

What is the key main result that demonstrates the effectiveness of this Lyapunov analysis framework?

**Limitations:**

The justification of the usefulness of the Lyapunov analysis framework is not sufficiently convincing, since many researchers in the field are already aware that many (if not all) discrete analyses have nice continuous-time counterparts with gradient flow or SDEs.

**Strengths And Weaknesses:**

The level of abstraction regarding Lyapunov functions is not something I have seen in the optimization and SGD literature, and the first result establishing the necessity and sufficiency of Lyapunov functions is interesting, at least to me. However, it is unclear to me what benefit this abstraction brings. The rates established for SGD are not better than any of the prior rates, and, in fact, many of the state-of-the-art rates are established not with the plain SGD but with some minor variation, such as certain varying stepsizes, appropriate batch sizes, or some non-uniform averaging schemes. Such variations probably can be considered under this framework, but this has not been done in this paper yet.

The control-theoretic framework feels like an interesting framework to me, but there's no key result that validates the strength of this viewpoint. The insistence on considering Lyapunov functions of gradient flow may be a key insight that future research on SGD will benefit from, but I don't think this work has sufficiently demonstrated the usefulness of this framework beyond its novelty.

---

> ### Author Response · Authors · 2022-08-02
> **Response to your questions**
>
> We thank the reviewer for their helpful suggestions, which we will be sure to incorporate into the final version. We will be happy to engage in follow-up discussion through openreview if the reviewer wishes!
>
> > The level of abstraction regarding Lyapunov functions is not something I have seen in the optimization and SGD literature, and the first result establishing the necessity and sufficiency of Lyapunov functions is interesting, at least to me. However, it is unclear to me what benefit this abstraction brings. The rates established for SGD are not better than any of the prior rates, and, in fact, many of the state-of-the-art rates are established not with the plain SGD but with some minor variation, such as certain varying stepsizes, appropriate batch sizes, or some non-uniform averaging schemes. Such variations probably can be considered under this framework, but this has not been done in this paper yet.
>
> The key contribution of our work is that for any problem of interest, (a) convergence of gradient flow and (b) extra regularity conditions e.g. smoothness, etc., are sufficient to show that SGD learns for that problem. To the best of our knowledge, this is the first result at such a level of generality. To illustrate the versatility of our framework, we demonstrate how to use our framework for a spectrum of problems ranging from simple problems like PL functions and convex functions to more complex non-convex problems like matrix square root and phase retrieval. Our bounds for GD match the state-of-the-art bounds for GD for both phase retrieval and matrix square root  (in terms of dependence on $T$), as well as for PL functions and convex functions.  Furthermore, to the best of our knowledge, ours is the first work to analyze the convergence of SGD under general noise conditions satisfying Assumption 3. Finally, the rate of SGD for matrix square root problem in the stochastic optimization setting was not known before and can be easily recovered from our framework. We have added this discussion in the rebuttal version of the paper.
>
> >What is the key main result that demonstrates the effectiveness of this Lyapunov analysis framework?
>
> Our key contribution is to provide a general framework that can be used to show the convergence of SGD. We demonstrate the versatility of our framework by showing how to use it for a spectrum of problems ranging from simple problems like PL functions and convex functions to more complex non-convex problems like matrix square root and phase retrieval. Using our framework, we can recover the state-of-the-art convergence bounds for gradient descent for both phase retrieval and matrix square root. Additionally, using our framework, we also provide the first analysis of SGD for both matrix square root and phase retrieval under general noise conditions satisfying Assumption 3.
>
> We believe that the key takeaway from our paper is that for a new problem of interest, one can show convergence of SGD by deriving a convergence rate for gradient flow and checking the respective self-bounding regularity conditions. The convergence of SGD then follows from Theorem 5.
>
> > The justification of the usefulness of the Lyapunov analysis framework is not sufficiently convincing, since many researchers in the field are already aware that many (if not all) discrete analyses have nice continuous-time counterparts with gradient flow or SDEs.
>
> While Lyapunov-based analysis is quite common in the literature to analyze continuous-time and discrete-time methods, we note that the task of coming up with a Lyapunov function for a new problem is often quite cumbersome and requires extensive domain knowledge. Our main contribution in this work is to precisely automate this. Note that for many problems showing the rate of convergence for gradient flow is extremely simple and follows simply by solving the gradient flow PDE. In these cases, using Theorem 2, one can construct a Lyapunov potential and use it for analyzing the discrete-time process (as we do in Theorem 4 and 5).

---

> > ### Comment · Reviewer_fx5A · 2022-08-09
> > **Thank you for your response**
> >
> > Thank you for your response. I do recognize that there is a definite novelty, as I had initially pointed out, and that it is interesting at some level. The non-convex analysis for matrix square root and phase retrieval alleviates my concerns. I have increased my score.

---

### Official Review · Reviewer_bbKp · 2022-07-12

**Rating:** 7
**Confidence:** 3
**Soundness:** 3 good
**Presentation:** 3 good
**Contribution:** 3 good

**Summary:**

This paper provides a new way of developing the convergence analysis of gradient descent (GD) and stochastic gradient descent (SGD) by leveraging the convergence of the gradient flow (GF). The main tool of the paper is to construct a Lyapunov potential which implies a convergence rate of the GF, formalized as a general converse Lyapunov like theorem. With additional self-bounding regularity conditions, the convergence of GF is sufficient to imply the convergence of GD and SGD. This framework is so general that it provides a unified analysis of GD and SGD that match the best convergence rate of convex functions or PL/KL functions, but also for phase retrieval and matrix square root problems.

**Questions:**

Typos:

Duplicate references: [21] & [22]; [41] & [42]

**Limitations:**

The authors have adequately addressed the limitations of their work.

**Strengths And Weaknesses:**

The paper is well-written, highlights their contribution clearly, provides lots of interpretations and simple examples of their general framework. The framework and the way to show convergence of GD and SGD are creative. There is a lot of conceptual value to the community that allows to develop new convergence analysis of GD and SGD in more complex models (e.g. neural networks). I am not familiar with the optimization theories of GF. The mathematical derivation of convergence seems solid.

My evaluation for the work is mostly positive. It may be better for the authors to write a paragraph how this work can be applied in the future. For instance, how SGD convergence analysis for the neural networks can be established under this framework ? More generally, can this framework be extended to other optimization methods, such as second-order methods and other variants of SGD (e.g. Adam, stochastic variance reduced gradient) ?

---

> ### Author Response · Authors · 2022-08-02
> **Response to your comments**
>
> We thank the reviewer for their positive comments and helpful suggestions, which we will be sure to incorporate into the final version.
> We will be happy to enagage in follow up discussion through openreview if the reviewer wishes!
>
>
> > It may be better for the authors to write a paragraph how this work can be applied in the future. For instance, how SGD convergence analysis for the neural networks can be established under this framework ?
>
> In the final version of the paper, we will add a section discussing how to use our framework to show convergence of SGD, whenever gradient flow has a rate for a new problem of interest (e.g. neural networks). The high level idea is that using the rate for gradient flow, one can construct using Theorem 2 a Lyapunov potential that satisfies the linearity property in eqn (4). Then, establishing the required self-bounding regularity conditions implies a learning rate for SGD (Theorem 5). For neural networks, rate of convergence of gradient flow are provided in [14], and we are excited to build on these rates for GF in our future research, to show convergence of SGD for neural networks.
>
> > More generally, can this framework be extended to other optimization methods, such as second-order methods and other variants of SGD (e.g. Adam, stochastic variance reduced gradient) ?
>
> Our framework to go from convergence of a continuous time method to its discrete counterpart can be extended beyond gradient flow and GD, to include algorithms like momentum, acceleration, etc. For a high level intuition, consider a general continuous time process $\frac{\text{d} w(t)}{\text{d} t} = u(w(t))$. The key idea to go from continuous time to discrete time, is to develop a potential that satisfies the linearity property  $\langle \nabla \Phi(w), u(w) \rangle \geq g(F(w))$. Such a potential can be constructed using the techniques similar to the proof of Theorem 2, by integrating the corresponding rate function under the process $u$. This potential can then be used to show the convergence guarantee for the corresponding discrete time process $w_{t + 1} \leftarrow w_t + \eta u(w_t)$, similar to the proof of Theorem 4 and 5. Extending this to second order method, and adam is an exciting direction for future research. We will provide a detailed discussion on this in the final version of the paper.

---

### Meta-Review · Area_Chair_pjJc · 2022-08-26

**Recommendation:** Accept
**Confidence:** Certain

**Metareview:**

This paper provides a new way of developing the convergence analysis of gradient descent (GD) and stochastic gradient descent (SGD) by leveraging the convergence of the gradient flow (GF). This framework is very general and provides new insight that future research on SGD will benefit from. All reviewer have positive feedback and I would like to recommend acceptance.

**Award:**

No

---

### Decision · Program_Chairs · 2022-09-14

Accept